# Adaptive Gradient Normalization and Independent Sampling for (Stochastic) Generalized-Smooth Optimization

**Yufeng Yang**                                                                 *yufeng.yang@tamu.edu*
*Department of Computer Science and Engineering*
*Texas A& M University*
*College Station, TX 77843, USA*

**Erin E. Tripp**                                                                 *etripp@hamilton.edu*
*Mathematics and Statistics Department*
*Hamilton College*
*Clinton, NY 13323, USA*

**Yifan Sun**                                                                 *ysun@cs.stonybrook.edu*
*Department of Computer Science*
*Stony Brook University*
*Stony Brook, NY 11794, USA*

**Shaofeng Zou**                                                                 *zou@asu.edu*
*School of Electrical, Computer and Energy Engineering*
*Arizona State University*
*Tempe, AZ 85287, USA*

**Yi Zhou**                                                                 *yi.zhou@tamu.edu*
*Department of Computer Science and Engineering*
*Texas A&M University*
*College Station, TX 77843, USA*

**Reviewed on OpenReview:** *https://openreview.net/forum?id=KKSQQMlEfw*

## Abstract

Recent studies have shown that many nonconvex machine learning problems satisfy a generalized-smooth condition that extends beyond traditional smooth nonconvex optimization. However, the existing algorithms are not fully adapted to such generalized-smooth nonconvex geometry and encounter significant technical limitations on their convergence analysis. In this work, we first analyze the convergence of adaptively normalized gradient descent under function geometries characterized by generalized-smoothness and generalized PŁ condition, revealing the advantage of adaptive gradient normalization. Our results provide theoretical insights into adaptive normalization across various scenarios. For stochastic generalized-smooth nonconvex optimization, we propose **I**ndependent-**A**daptively **N**ormalized **S**tochastic **G**radient **D**escent algorithm, which leverages adaptive gradient normalization, independent sampling, and gradient clipping to achieve an $\mathcal{O}(\epsilon^{-4})$ sample complexity under relaxed noise assumptions. Experiments[1] on large-scale nonconvex generalized-smooth problems demonstrate the fast convergence of our algorithm.

## 1 Introduction

In modern machine learning, the convergence of gradient-based optimization algorithms has been well studied in the standard smooth nonconvex setting. However, it has been shown recently that the standard $L$-

---

[1]Code available at github.com/ynyang94/Gensmooth-IAN-SGD

smoothness fails to hold in many nonconvex machine learning problems, including distributionally-robust optimization(DRO) (Levy et al., 2020; Jin et al., 2021), meta-learning (Nichol et al., 2018; Chayti & Jaggi, 2024) and language models (Liu et al., 2023; Zhang et al., 2019). Instead, these problems were shown to satisfy a so-called *generalized-smooth* condition (Zhang et al., 2019), in which the smoothness parameter can scale with the gradient norm in the optimization process.

In the existing literature, various works have proposed different algorithms for solving generalized-smooth nonconvex optimization problems. Specifically, Li et al. (2024); Zhang et al. (2019); Chen et al. (2023); Gorbunov et al. (2024); Vankov et al. (2024b); Reisizadeh et al. (2023) have demonstrated that *deterministic* first-order algorithms such as gradient descent, normalized gradient descent and clipped gradient descent can achieve $\mathcal{O}(\epsilon^{-2})$ iteration complexity under mild assumptions. These complexity results match the lower bound obtained by classical first-order methods (Arjevani et al., 2023). In particular, Chen et al. (2023) empirically demonstrated that proper usage of adaptive gradient normalization can substantially accelerate convergence. However, the formal theoretical justification and understanding of adaptive gradient normalization is still lacking for first-order algorithms in generalized-smooth optimization.

On the other hand, some other works, including Li et al. (2024); Zhang et al. (2019; 2020a) studied first-order *stochastic* algorithms in generalized-smooth nonconvex optimization. Specifically, one line of works (Li et al., 2024) focused on the classic stochastic gradient descent (SGD) algorithm (Ghadimi & Lan, 2013). However, in generalized-smooth setting, the convergence analysis of SGD either relies on adopting very large batch size or involves large constants (Li et al., 2024). These theoretical bottlenecks may be a key factor limiting the practical performance of SGD. Empirical observations from Chen et al. (2023) indicate that SGD often converges slowly due to ill-conditioned smoothness parameters when the gradient norm is large. Another line of works (Zhang et al., 2019; 2020a; Reisizadeh et al., 2023; Koloskova et al., 2023) focused on clipped SGD, which leverages gradient normalization and clipping to handle the generalized-smooth geometry. Although clipped SGD has demonstrated superior performance in solving large-scale problems, the existing theoretical analysis has several limitations. First, to establish convergence guarantees in the stochastic setting, existing studies often rely on strong assumptions, such as the stochastic approximation error being almost surely bounded or the use of extremely large batch sizes. Second, the existing designs of clipped SGD adopt the standard stochastic gradient normalization scheme, which is not fully adapted to the function geometry characterized by the generalized-smooth condition.

Having observed the algorithmic and theoretical limitations discussed above, we aim to advance the algorithm design and analysis for generalized-smooth optimization through investigating the following two fundamental and complementary questions.

- *Q1: In deterministic generalized-smooth optimization, how does adaptive gradient normalization affect the convergence rate of first-order algorithm, e.g., under Polyak-Łojasiewicz type conditions?*

- *Q2: In stochastic generalized-smooth optimization, can we design a novel algorithm that guarantees convergence under relaxed noise assumptions while relying on only a small number of samples for computing each stochastic gradient?*

In this work, we provide comprehensive answers to both questions by developing new algorithms and convergence analysis in generalized-smooth nonconvex optimization. We summarize our contributions as follows.

## 1.1 Our Contributions

To understand the advantage of using adaptive gradient normalization, we first study the convergence rate of adaptive normalized gradient descent (AN-GD) in deterministic generalized-smooth optimization under the generalized Polyak-Łojasiewicz (PŁ) condition over a broad spectrum of gradient normalization parameters. Our results reveal the interplay among learning rate, gradient normalization parameter and function geometry parameter, and characterize their impact on the type of convergence rate. In particular, our results reveal the advantage of using adaptive gradient normalization and provide theoretical guidance on choosing proper gradient normalization parameter to improve convergence rate.

We further propose a novel Independent-Adaptively Normalized Stochastic Gradient Descent (IAN-SGD) algorithm tailored for stochastic generalized-smooth nonconvex optimization. Specifically, IAN-SGD leverages normalized gradient updates with independent sampling and gradient clipping to reduce bias and enhance algorithm stability. Consequently, we are able to establish convergence of IAN-SGD with $\mathcal{O}(\epsilon^{-4})$ sample complexity under a relaxed assumption on the approximation error of stochastic gradient and constant-level batch size. This makes the algorithm well-suited for solving large-scale problems.

We compare the numerical performance of our IAN-SGD algorithm with other state-of-the-art stochastic algorithms in applications of nonconvex phase retrieval, distributionally-robust optimization and training deep neural networks, all of which are generalized-smooth nonconvex problems. Our results demonstrate the efficiency of IAN-SGD in solving generalized-smooth nonconvex problems.

## 2 Related Work

**Generalized-Smoothness.** The concept of generalized-smoothness was first introduced by Zhang et al. (2019) with the $(L_0, L_1)$-smooth condition, which allows a function to either have an affine-bounded Hessian norm or be locally $L$-smooth within a specific region. This definition was extended by Chen et al. (2023), who proposed the $\mathcal{L}^*_{asym}(\alpha)$ and $\mathcal{L}^*_{sym}(\alpha)$ conditions, controlling gradient changes globally with both a constant term and a gradient-dependent term associated with power $\alpha$, thus applying more broadly. Later, Li et al. (2024) introduced $\ell$-smoothness, which use a non-decreasing sub-quadratic polynomial to control gradient differences. Mishkin et al. (2024) proposed directional smoothness, which preserves $L$-smoothness along specific directions.

**Algorithms for Generalized-Smooth Optimization.** Motivated by achieving comparable lower bounds presented in Arjevani et al. (2023) under standard assumptions, algorithms for solving generalized-smooth problems can be categorized into two main series. The first series focuses on gradient descent methods with constant learning rate. Li et al. (2024) proved that GD, SGD converge with $\mathcal{O}(\epsilon^{-2})$ and $\mathcal{O}(\epsilon^{-4})$ complexity under generalized-smoothness. To ensure convergence, Li et al. (2024) adopted assumptions such as gradient upper-bound and bounded variance assumption.

Another series of work focuses on adaptive methods. In the nonconvex deterministic settings, Zhang et al. (2019; 2020a); Gorbunov et al. (2024); Vankov et al. (2024b) showed that clipped GD can achieve an iteration complexity of $\mathcal{O}(\epsilon^{-2})$ under mild assumptions. Later, Chen et al. (2023) proposed $\beta$-GD that achieves $\mathcal{O}(\epsilon^{-2})$ iteration complexity. Specifically, in the convex setting, Vankov et al. (2024b) analyzed clipped and normalized gradient descent under mild assumptions. Gorbunov et al. (2024) studied smoothed gradient clipping, gradient descent with Polyak step-size rule, and triangle method by varying the learning rates. Both works achieve the best-known $\mathcal{O}(\epsilon^{-1})$ convergence rate for generalized-smooth convex optimization. In the nonconvex stochastic setting, Zhang et al. (2020b) proved that clipped SGD achieves an $\mathcal{O}(\epsilon^{-4})$ sample complexity under the standard $L$-smoothness and bounded variance assumptions. Also, Zhang et al. (2019) established the same convergence rate for clipped SGD under generalized smoothness and almost surely bounded noise assumption. Later, Reisizadeh et al. (2023); Koloskova et al. (2023) analyzed the convergence of clipped SGD under the bounded variance assumption. Notably, Koloskova et al. (2023) pointed out that, with bounded variance assumption, clipped SGD converges to $\epsilon$-stationary point with sample complexity $\mathcal{O}(\epsilon^{-5})$ when using $\Omega(1)$ samples per iteration. To improve convergence sample complexity, Reisizadeh et al. (2023) employed a large batch size $\mathcal{O}(\epsilon^{-2})$ to suppress the impact of the variance term and achieve $\mathcal{O}(\epsilon^{-4})$ complexity. We summarize these results under mild assumptions in Table 1.

A number of works also investigated the performance of diverse stochastic algorithms—extending beyond SGD and clipped SGD under the generalized-smooth condition. Wang et al. (2023); Faw et al. (2023); Hong & Lin (2024) studied AdaGrad (Duchi et al., 2011b) under generalized-smooth and affine variance assumption with different learning rate schemes. They all attain $\tilde{\mathcal{O}}(1/\sqrt{T})$ convergence rate under mild conditions. Xie et al. (2024a) studied trust-region methods convergence under generalized-smoothness. For stochastic acceleration methods under the generalized-smoothness condition, Zhang et al. (2020a) proposed a general clipping framework by leveraging momentum clipping and they achieve $\mathcal{O}(\epsilon^{-4})$ sample complexity under almost sure bounded noise assumption; Jin et al. (2021) studied normalized SGD with momentum under parameter-dependent learning rates schemes, which achieves $\mathcal{O}(\epsilon^{-4})$ sample complexity under generalized-

smooth and affine variance assumption; Hübler et al. (2024) studied normalized SGD with momentum (Cutkosky & Mehta, 2020) under parameter-agnostic learning rates schemes, which establishes $\tilde{\mathcal{O}}(\epsilon^{-4})$ convergence rate under bounded variance assumption. Chen et al. (2023); Reisizadeh et al. (2023) demonstrated that the SPIDER algorithm (Fang et al., 2018) can reach the optimal $\mathcal{O}(\epsilon^{-3})$ sample complexity under affine variance assumption for sum-type functions by employing a large batch size. Furthermore, Zhang et al. (2024b); Wang et al. (2024a;b); Li et al. (2023) explored the convergence of RMSprop (Hinton et al., 2012) and Adam (Kingma, 2014) under different noise assumptions, including coordinate-wise affine variance, expected affine variance, bounded variance and sub-gaussian distribution. The acceleration of sign-SGD was explored by Jiang et al. (2024); Sun et al. (2023b), and its generalized-smooth variants were subsequently analyzed by Crawshaw et al. (2022); Sun et al. (2023a), where they both obtain $\mathcal{O}(\epsilon^{-4})$ under almost-sure bound and bounded variance assumptions, respectively.

| Method | Smoothness[1] | Noise assumption[2] | Algorithm[3] | Batch Size[4] | Convergence[5] | Complexity |
|---|---|---|---|---|---|---|
| Zhang et al. (2019) | GS | a.s. bound | Clip-SGD | $\Omega(1)$ | $\epsilon$-stationary | $\mathcal{O}(\epsilon^{-4})$ |
| Zhang et al. (2020b) | LS | var bound | Clip-SGD | $\Omega(1)$ | $\epsilon$-stationary | $\mathcal{O}(\epsilon^{-4})$ |
| Zhang et al. (2020a) | GS | a.s. bound | General-Clip | $\Omega(1)$ | $\epsilon$-stationary | $\mathcal{O}(\epsilon^{-4})$ |
| Li et al. (2024) | GS | var bound | SGD | $\Omega(1)$ | $\epsilon$-stationary | $\mathcal{O}(\epsilon^{-4})$ |
| Reisizadeh et al. (2023) | GS | var bound | Clip-SGD | $\Omega(\epsilon^{-2})$ | $\epsilon$-stationary | $\mathcal{O}(\epsilon^{-4})$ |
| Koloskova et al. (2023) | GS | var bound | Clip-SGD | $\Omega(1)$ | $\epsilon$-stationary | $\mathcal{O}(\epsilon^{-5})$ |
| Our Method | GS | a.s. affine | IAN-SGD | $\Omega(1)$ | $\epsilon$-stationary | $\mathcal{O}(\epsilon^{-4})$ |

Table 1: Comparison of existing clipping-based algorithms. Denote the stochastic gradient and clipping threshold as $\nabla f_\xi(w_t)$ and $c$, respectively. Explanation of abbreviations: (GS) refers to the generalized-smoothness condition, and (LS) refers to the standard $L$-smoothness condition. (a.s. bound) means that for every sample, the stochastic gradient bias is almost surely bounded, i.e., $\|\nabla f_\xi(w_t) - \nabla F(w_t)\| \leq \tau_2$; (var bound) denotes the bounded variance condition, i.e., $\mathbb{E}_\xi[\|\nabla f_\xi(w_t) - \nabla F(w_t)\|^2] \leq \tau_2^2$; (a.s. affine) implies that for every sample, the stochastic gradient bias satisfies the affine bound $\|\nabla f_\xi(w_t) - \nabla F(w_t)\| \leq \tau_1\|\nabla F(w_t)\| + \tau_2$ almost surely, for some $0 \leq \tau_1 < 1$ and $\tau_2 > 0$. The term *Clip-SGD* refers to algorithms that use stochastic gradient clipping of the form $w_{t+1} = w_t - \gamma \min\{1, \frac{c}{\|\nabla f_\xi(w_t)\|}\}\nabla f_\xi(w_t)$; *SGD* refers to stochastic gradient descent; *General-Clip* denotes algorithms that incorporate momentum during clipping and updates $w_{t+1} = w_t - (\nu \min\{c, \frac{\gamma}{\|m_t\|}\}m_t + (1-\nu)\min\{c, \frac{\gamma}{\|\nabla f_\xi(w_t)\|}\}\nabla f_\xi(w_t))$, where $m_{t+1} = (1-\vartheta)m_t + \vartheta\nabla f_\xi(w_t)$. The label $\epsilon$-stationary indicates that the algorithm can converge to an arbitrary target error. For Koloskova et al. (2023), we report their convergence result under learning rate $\gamma = \epsilon^2/\left(\tau_2^2(L_0 + cL_1)\right)$ with $c = \tau_2/\epsilon$, where their analysis guarantees convergence to an $\epsilon$-stationary point.

**Machine Learning Applications.** Generalized-smoothness has been studied under various machine learning frameworks. Levy et al. (2020); Jin et al. (2021) studied the dual formulation of regularized DRO problems, where the loss function objective satisfies generalized-smoothness. Chayti & Jaggi (2024) identified their meta-learning objective's smoothness constant increases with the norm of the meta-gradient. Gong et al. (2024b); Hao et al. (2024); Gong et al. (2024a); Liu et al. (2022b) explored algorithms for bi-level optimization and federated learning within the context of generalized-smoothness. Zhang et al. (2024a) developed algorithms for multi-task learning problem where the objective is generalized-smooth. Xie et al. (2024b) studied online mirror descent when the objective is generalized-smooth. Xian et al. (2024) studied min-max optimization algorithms' convergence behavior under generalized-smooth condition. There is a concurrent work (Vankov et al., 2024a) using independent sampling with clipped SGD framework to solve variational inequality problem (SVI). Based on this idea, they also propose stochastic Korpelevich method for clipped SGD. Under generalized-smooth condition, they proved almost-sure convergence in terms of distance to solution set tailored for solving stochastic SVI problems.

## 3 Notations

Throughout the work, we denote $D$ as a set of training examples, $\xi$ as an example sampled from the training dataset $D$, $f(\cdot)$ as the objective function, $f_\xi(\cdot)$ as the objective function associated with data sample $\xi$, $F(w) = \mathbb{E}_{\xi\sim\mathbb{P}}[f_\xi(w)]$ as the expected loss over all samples following distribution $\mathbb{P}$, and $\|\cdot\|$ as $\ell_2$-norm over

Euclidean space. Let $w \in \mathbf{R}^d$ denote the parameters of function $f$ (i.e., the weights for linear model and neural networks), $f^*$ denote the infimum of $f$. $L_0, L_1$ and $\alpha$ are positive constants used to characterize the generalized-smoothness condition (see Assumption 1). $\rho, \mu$ are positive constant used to describe generalized PŁ-condition (see Assumption 2). $\tau_1, \tau_2$ characterize the stochastic gradient noise (see Assumption 4). $\gamma, \beta$ denote the learning rate and normalization power used in adaptive normalization, respectively. In the Distributionally Robust Optimization (DRO) experiments (see Section 7), we define $L(w, \eta)$ with parameter $w$ and dual variable $\eta$, as the dual objective derived from a regularized $\phi$-divergence primal DRO problem.

## 4 Generalized-Smooth nonconvex Optimization

We first introduce generalized-smooth optimization problems. Consider the following optimization problem.

$$\min_{w \in \mathbf{R}^d} f(w), \tag{1}$$

where $f : \mathbf{R}^d \to \mathbf{R}$ denotes a nonconvex and differentiable function and $w$ corresponds to the model parameters. We assume that function $f$ satisfies the following generalized-smooth condition.

**Assumption 1 (Generalized-smooth)** *The objective function $f$ satisfies the following conditions.*

1. *$f$ is differentiable and bounded below, i.e., $f^* := \inf_{w \in \mathbf{R}^d} f(w) > -\infty$;*

2. *There exists constants $L_0, L_1 > 0$ and $\alpha \in [0, 1]$ such that for any $w, w' \in \mathbf{R}^d$, it holds that*

$$\left\|\nabla f(w) - \nabla f(w')\right\| \leq \left(L_0 + L_1\left\|\nabla f(w')\right\|^\alpha\right)\left\|w - w'\right\|. \tag{2}$$

The generalized-smooth condition in Assumption 1 is a generalization of the standard smooth condition, which corresponds to the special case of $\alpha = 0$. It allows the smoothness parameter to scale with the gradient norm polynomially, and therefore Assumption 1 can model functions with highly irregular geometry. Moreover, following the standard proof, it is easy to show that generalized-smooth functions satisfy the following descent lemma. See Proof 1 in Appendix B.

**Lemma 1** *Under Assumption 1, function $f$ satisfies, for any $w, w' \in \mathbf{R}^d$,*

$$f(w) \leq f(w') + \langle \nabla f(w'), w - w' \rangle + \frac{1}{2}(L_0 + L_1\left\|\nabla f(w')\right\|^\alpha)\left\|w - w'\right\|^2. \tag{3}$$

We note that there are several variants of generalized-smooth conditions proposed by the previous works (Zhang et al., 2019; Jin et al., 2021; Chen et al., 2023). Below, we briefly discuss the relationship between the generalized-smooth condition in Assumption 1 and these existing notions.

**Remark 1 ($(L_0, L_1)$-generalized-smooth condition)** *The descent lemma of $(L_0, L_1)$-generalized-smooth condition proposed in Zhang et al. (2019) is given as*

$$f(w) \leq f(w') + \langle \nabla f(w'), w - w' \rangle + \frac{1}{2}(4L_0 + 5L_1\|\nabla f(w')\|)\|w - w'\|^2,$$

*which is the same as Lemma 1 with $\alpha = 1$ up to differences in the constant coefficients.*

**Remark 2 (Symmetric generalized-smooth condition)** *Chen et al. (2023) introduced a symmetric version of generalized-smooth condition, by replacing $\|\nabla f(w')\|^\alpha$ in equation 2 with $\max_{w_\theta} \|f(w_\theta)\|^\alpha$, where $w_\theta = \theta w' + (1-\theta)w$. We notice that the asymmetric generalized-smooth condition adopted in our Assumption 1 can also imply the following similar symmetric generalized-smooth condition,*

$$\left\|\nabla f(w) - \nabla f(w')\right\| \leq \left(L_0 + L_1\left(\frac{\left\|\nabla f(w)\right\|^\alpha + \left\|\nabla f(w')\right\|^\alpha}{2}\right)\right)\left\|w - w'\right\|. \tag{4}$$

In Jin et al. (2021), it has been shown that certain regularized nonconvex distributionally robust optimization (DRO) problems satisfy the generalized-smooth condition in equation 2 with $L_0 = L$, $L_1 = \frac{2M(G+1)^2}{\lambda}$, $\alpha = 1$. Moreover, the classic nonconvex phase retrieval problem (Drenth, 2007; Miao et al., 1999) can be shown to satisfy the above symmetric generalized-smooth condition in equation 4 (see Proof 2 in Appendix C).

In the following sections, we first consider deterministic generalized-smooth optimization and study the impact of adaptive gradient normalization on the convergence rate of gradient methods. Then, we consider stochastic generalized-smooth optimization and propose a novel independent sampling scheme for improving the convergence guarantee of stochastic gradient methods under relaxed noise assumption.

## 5 Adaptive Gradient Normalization for Deterministic Generalized-Smooth Optimization

In deterministic generalized-smooth optimization, several previous works have empirically demonstrated the faster convergence of normalized gradient descent-type algorithms (e.g., clipped GD) over the standard gradient descent algorithm in various machine learning applications (Zhang et al., 2019; Chen et al., 2023). On the other hand, theoretically, these algorithms were only shown to achieve the same iteration complexity $\mathcal{O}(\epsilon^{-2})$ as the gradient descent algorithm in generalized-smooth optimization. In this section, to further advance the theoretical understanding and explain the inconsistency between theory and practice, we explore the advantage of adapting gradient normalization to the special Polyak-Łojasiewicz-type (PŁ) geometry in generalized-smooth optimization. We aim to show that gradient normalization, when properly adapted to the underlying PŁ geometry, can help accelerate the convergence rate in generalized-smooth optimization.

Specifically, we consider the class of generalized-smooth problems that satisfy the following generalized PŁ geometry.

**Assumption 2 (Generalized Polyak-Łojasiewicz Geometry)** *There exists constants $\mu \in \mathbf{R}_+$ and $0 < \rho \leq 2$ such that $f(\cdot)$ satisfies, for all $w \in \mathbf{R}^d$,*

$$\left\| \nabla f(w) \right\|^\rho \geq 2\mu(f(w) - f^*). \tag{5}$$

The above generalized PŁ condition is inspired by the Kurdyka-Łojasiewicz (KŁ)-exponent condition proposed in Li & Pong (2018). When $\rho > 1$, equation 5 reduces to the KŁ-exponent condition. When $\rho = 2$, equation 5 reduces to the standard PŁ condition. Moreover, some recent works have shown that PŁ-type geometries widely exist in the loss landscape of phase retrieval (Zhou et al., 2016) and over-parametrized deep neural networks (Liu et al., 2022a; Scaman et al., 2022), and we hope that our analysis based on Assumption 2 will allow researchers to rethink the relationship between adaptive normalization and loss landscape geometry.

Here, we consider the adaptively normalized gradient descent (AN-GD) algorithm proposed by Chen et al. (2023) for generalized-smooth nonconvex optimization. The algorithm normalizes the gradient update as follows

$$(\text{AN-GD}) \quad w_{t+1} = w_t - \gamma \frac{\nabla f(w_t)}{\|\nabla f(w_t)\|^\beta}, \tag{6}$$

where $\gamma > 0$ denotes the learning rate and $\beta$ is a normalization scaling parameter that allows us to adapt the normalization scale of the gradient norm to the underlying function geometry. Intuitively, when the gradient norm is large, a smaller $\beta$ would make the normalized gradient update more aggressive; when the gradient norm is small, normalization can slow down gradient vanishing and improve numerical stability.

Chen et al. (2023) studied AN-GD in generalized-smooth nonconvex optimization, showing that it achieves the standard $\mathcal{O}(\epsilon^{-2})$ iteration complexity lower bound. In the following theorem, we obtain the convergence rate of AN-GD under the generalized PŁ condition. See Proof 4 in Appendix E.

**Theorem 1 (Convergence of AN-GD)** *Let Assumptions 1 and 2 hold. Denote $\Delta_t := f(w_t) - f^*$ as the function value gap. Choose target error $0 < \epsilon \leq \min\{1, \frac{1}{2\mu}\}$ and define learning rate $\gamma = \frac{(2\mu\epsilon)^{\beta/\rho}}{8(L_0+L_1)+1} \leq \frac{2}{\mu}$ for some $\beta \in [\alpha, 1]$. Then, the following statements hold,*

- *If $\beta < 2 - \rho$, in order to achieve $\Delta_T \leq \epsilon$, the total number of iterations must satisfy*

$$T \geq \max\left\{\frac{8\rho(8(L_0 + L_1) + 1)}{(2 - \beta - \rho)(2\mu)^{2/\rho}\epsilon^{(2-\rho)/\rho}}, \frac{1}{(2^{(2-\beta-\rho)/(2-\beta)} - 1)\Delta_0^{(\rho+\beta-2)/\rho}\epsilon^{(2-\beta-\rho)/\rho}}\right\} = \Omega\left((\frac{1}{\epsilon})^{\frac{2-\rho}{\rho}}\right). \quad (7)$$

- *If $\beta = 2 - \rho$, in order to achieve $\Delta_T \leq \epsilon$, the total number of iterations must satisfy*

$$T \geq \frac{2^{1-\beta/\rho} \cdot (8(L_0 + L_1) + 1)}{\mu \cdot (\mu\epsilon)^{\beta/\rho}} \cdot \log(\frac{\Delta_0}{\epsilon}) = \Omega\left((\frac{1}{\epsilon})^{\frac{\beta}{\rho}} \cdot \log(\frac{\Delta_0}{\epsilon})\right). \quad (8)$$

- *If $1 \geq \beta > 2 - \rho$, and for some $T_0$ satisfying $\Delta_{T_0} \leq \mathcal{O}\left((\gamma\mu^{(2-\beta)/(\rho+\beta-2)})^{\frac{\rho}{\rho+\beta-2}}\right)$, then the total number of iterations after $T_0$ must satisfy*

$$T \gtrsim \Omega\left(\log\left((\frac{1}{\epsilon})^{\frac{\beta}{\rho+\beta-2}}\right) + \log\left(\log\left(\frac{(2\mu)^{2/(\rho+\beta-2)}}{(32(L_0 + L_1) + 4)^{\rho/(\rho+\beta-2)}\epsilon}\right)\right)\right) = \Omega\left(\log\left((\frac{1}{\epsilon})^{\frac{\beta}{\rho+\beta-2}}\right)\right). \quad (9)$$

Theorem 1 characterizes the convergence rates of AN-GD under different choices of $\rho$ and $\beta$. In particular, when $\alpha = \beta = 0$ and $\rho = 2$, Theorem 1 reduces to the linear convergence rate achieved by gradient descent under the standard PŁ and $L$-smooth condition. Theorem 1 also guides the choice of gradient normalization parameter $\beta$ under different geometric conditions. For example, if there exists $\beta \in [\alpha, 1]$ such that $\rho + \beta > 2$, AN-GD can boost its convergence rate from polynomial to local linear convergence. It also indicates when $\rho$ is very small such that $\rho + \beta < 2$, the effects of $\beta$ can be marginal. However, for $\rho + \beta \leq 2$, the iteration complexities depend more on the specific values of $\rho$ and $\beta$. For example, when $\rho = 1$ and consider two different choices of gradient normalization parameters $\beta_1 = \frac{2}{3}, \beta_2 = 1$, AN-GD achieves the iteration complexities $\mathcal{O}(\epsilon^{-1})$ and $\tilde{\mathcal{O}}(\epsilon^{-1})$, respectively. This result matches the empirical observation made in (Chen et al., 2023) that choosing a smaller $\beta \in [\alpha, 1]$ can improve the convergence rate.

We note that several concurrent works also studied normalized gradient descent methods in deterministic convex settings. In Gorbunov et al. (2024), the authors consider various update rules such as $w_{t+1} = w_t - \gamma \frac{\nabla f(w_t)}{L_0 + L_1 \|\nabla f(w_t)\|}$ and $w_{t+1} = w_t - \gamma \frac{f(w_t) - f(w^*)}{\|\nabla f(w_t)\|} \nabla f(w_t)$. For generalized smooth convex optimization, they obtain a two-phase convergence result, similar to the case when $1 \geq \beta > 2 - \rho$ in Theorem 1, where the second phase achieves complexity bound $\mathcal{O}(\epsilon^{-1})$. They further employ the triangle method to derive a global $\mathcal{O}(\epsilon^{-1})$ complexity bound. In Vankov et al. (2024b), the authors propose a related framework by choosing $\gamma = \frac{1}{\sqrt{T}\|\nabla f(w_t)\|}$, which aligns with our AN-GD when $\beta = 1$. They also obtain $\mathcal{O}(\epsilon^{-1})$ complexity for generalized smoothness convex optimization.

### 5.1 Comparison between AN-GD and GD.

In Li et al. (2024), it has been shown that GD achieves $\mathcal{O}(\log(\epsilon^{-1}))$, $\mathcal{O}(\epsilon^{-1})$, $\mathcal{O}(\epsilon^{-2})$ complexity bounds for generalized-smooth strongly convex, convex and non-convex optimization, respectively. Compared with Chen et al. (2023), both AN-GD and GD achieve $\mathcal{O}(\epsilon^{-2})$ complexity bound for generalized non-convex optimization, and hence are optimal deterministic methods. On the other hand, the following Proposition 1 further obtains the convergence rate of gradient descent (GD) under the same Assumptions 1 and 2. See Proof 5 in Appendix F.

**Proposition 1 (Convergence of GD)** *Let Assumptions 1 and 2 hold. Assume there exists a positive constant $G$ such that $\|\nabla f(x_t)\| \leq G, \forall t \in T$, When $\rho = \alpha = 1$ and setting $\gamma \leq \min\{\frac{1}{L_0}, \frac{1}{2L_1 G}\}$, gradient descent converges to an $\epsilon$-stationary point within $T \geq \Omega(\frac{G}{\epsilon})$ iterations.*

From Theorem 1, under the setting $\rho = \alpha = 1$ and with $\beta = 1$, AN-GD converges to a $\epsilon$-stationary point in $\tilde{\mathcal{O}}(\epsilon^{-1})$ iterations. However, the convergence guarantee for GD requires an additional assumption that the gradient norm is upper bounded by a constant $G$. This constant typically depends on the function value gap $\Delta_0$ and $\|x_0 - x^*\|$, which can be large in generalized smooth optimization settings. Such large value can be the key reason for the slower empirical convergence of GD observed in Chen et al. (2023).

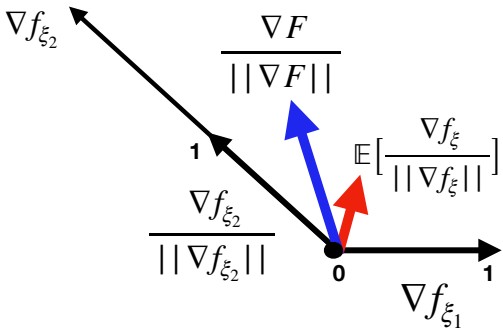

Figure 1: Comparison between normalized full gradient (blue) and expected normalized stochastic gradient (red). Here, $\xi_1$ and $\xi_2$ are sampled uniformly at random.

# 6 Independently-and-Adaptively Normalized SGD for Stochastic Generalized-Smooth Optimization

In this section, we study stochastic generalized-smooth optimization problems, where we denote $f_\xi$ as the loss function associated with the data sample $\xi$, and we assume that the following expected loss function $F(\cdot)$ satisfies the generalized-smooth condition in Assumption 1.

$$\min_{w \in \mathbf{R}^d} F(w) = \mathbb{E}_{\xi \sim \mathbb{P}}\big[f_\xi(w)\big]. \tag{10}$$

Having discussed the superior theoretical performance of AN-GD in the previous section, we aim to leverage adaptive gradient normalization to further develop an adaptively normalized algorithm tailored for stochastic generalized-smooth optimization.

## 6.1 Normalized SGD and Its Limitations

To solve the stochastic generalized-smooth problem in equation 10, one straightforward approach is to replace the full batch gradient in the AN-GD update rule, equation 6 with the stochastic gradient $\nabla f_\xi(w_t)$, resulting in the following adaptively normalized SGD (AN-SGD) algorithm.

$$(\text{AN-SGD}) \quad w_{t+1} = w_t - \gamma \frac{\nabla f_\xi(w_t)}{\|\nabla f_\xi(w_t)\|^\beta}. \tag{11}$$

AN-SGD-type algorithms have attracted a lot of attention recently for solving stochastic generalized-smooth problems (Zhang et al., 2019; 2020a; Liu et al., 2022b; Jin et al., 2021). In particular, previous works have shown that, when choosing $\beta = 1$ and using gradient clipping or momentum acceleration, AN-SGD's variations (e.g., NSGD with momentum (Cutkosky & Mehta, 2020; Jin et al., 2021), clipped SGD (Zhang et al., 2019; 2020a)) can achieve a sample complexity of $\mathcal{O}(\epsilon^{-4})$. This result matches the convergence order of the standard SGD (Ghadimi & Lan, 2013) for solving classic stochastic smooth problems. However, AN-SGD has several limitations as summarized below.

- *Biased gradient estimator:* The normalized stochastic gradient used in equation 11 is biased, i.e., $\mathbb{E}\big[\frac{\nabla f_\xi(w_t)}{\|\nabla f_\xi(w_t)\|^\beta}\big] \neq \frac{\nabla F(w_t)}{\|\nabla F(w_t)\|^\beta}$. This is due to the dependence between $\nabla f_\xi(w_t)$ and $\|\nabla f_\xi(w_t)\|^\beta$. In particular, the bias can be huge if the stochastic gradients are diverse, as illustrated in Figure 1.

- *Strong assumptions:* To control the estimation bias and establish theoretical convergence guarantee for ANSGD-type algorithms in generalized-smooth nonconvex optimization, the existing studies need to adopt strong assumptions. For example, Zhang et al. (2019; 2020a) and Liu et al. (2022b) assume that the stochastic approximation error $\|\nabla f_\xi(w) - \nabla F(w)\|$ is bounded by a constant almost surely. In practical applications, this constant can be a large numerical number if certain sample $\xi$ happen to be an outlier. On the other hand, Chen et al. (2023); Reisizadeh et al. (2023) require AN-SGD type of algorithms such as SPIDER (Fang et al., 2018) to have $\Omega(\epsilon^{-2})$ batch size to ensure fast convergence. It essentially makes it a full-batch algorithm and is impractical in practice.

## 6.2 Independently-and-Adaptively Normalized SGD

To overcome the aforementioned limitations, we propose the following independently-and-adaptively normalized stochastic gradient (IAN-SG) estimator

$$\text{(IAN-SG estimator)} \quad \frac{\nabla f_\xi(w)}{\|\nabla f_{\xi'}(w)\|^\beta}, \tag{12}$$

where $\xi$ and $\xi'$ are samples drawn *independently* from the underlying data distribution. Intuitively, the independence between $\xi$ and $\xi'$ decorrelates the denominator from the numerator, making it an unbiased stochastic gradient estimator (up to a scaling factor). Specifically, we have that

$$\mathbb{E}_{\xi,\xi'}\left[\frac{\nabla f_\xi(w)}{\|\nabla f_{\xi'}(w)\|^\beta}\right] = \mathbb{E}_{\xi'}\left[\frac{\mathbb{E}_\xi[\nabla f_\xi(w)]}{\|\nabla f_{\xi'}(w)\|^\beta}\right] \propto \nabla F(w). \tag{13}$$

Moreover, as we show later under mild assumptions, the scaling factor $\mathbb{E}[\|\nabla f_{\xi'}(w)\|^{-\beta}]$ can be roughly bounded by the full gradient norm and hence resembling the full-batch AN-GD update. Based on this idea, we formally propose the following independently-and-adaptively normalized SGD (IAN-SGD) algorithm (see equation 14), where $A$, $\Gamma$, $\delta$ are positive constants, $\nabla f_\xi(w_t)$, $\nabla f_{\xi'}(w_t)$ corresponds to the stochastic gradient associated with two independent samples $\xi$, $\xi'$ sampled at time $t$.

$$\text{(IAN-SGD):} \quad w_{t+1} = w_t - \gamma\frac{\nabla f_\xi(w_t)}{h_t^\beta},$$
$$\text{where} \quad h_t = \max\left\{1, \Gamma \cdot \left(A\|\nabla f_{\xi'}(w_t)\| + \delta\right)\right\}. \tag{14}$$

The above IAN-SGD algorithm adopts a clipping strategy for the normalization term $h_t$. Intuitively, when gradient converges in the later optimization phase, the term $h_t = 1$ and IAN-SGD reduces to SGD. This is consistent with the fact that the generalized-smooth condition reduces to the standard smooth condition when gradient approaches to 0. Moreover, imposing a constant lower bound $\delta$ on $h_t$ in equation 14 helps develop the theoretical convergence analysis and avoid numerical instability in practice. As we show in the ablation study presented in the appendix, the convergence speed of IAN-SGD is insensitive to the choice of $\delta$. We also note that IAN-SGD requires querying two batches of samples in every iteration. However, we show later in the Theorem 2 and experiments that these batch sizes can be chosen at a constant level in both theory and practice.

## 6.3 Convergence Analysis of IAN-SGD

We adopt the following bias assumption on the stochastic gradient $\nabla f_\xi(w)$.

**Assumption 3 (Unbiased stochastic gradient)** *The stochastic gradient $\nabla f_\xi(w)$ is unbiased, i.e.,* $\mathbb{E}_{\xi\sim\mathbb{P}}[\nabla f_\xi(w)] = \nabla F(w)$ *for all* $w \in \mathbf{R}^d$.

**Assumption 4 (Affine approximation Error)** *There exists* $0 \le \tau_1 < 1, \tau_2 > 0$ *such that for any* $w \in \mathbf{R}^d$,

$$\left\|\nabla f_\xi(w) - \nabla F(w)\right\| \le \tau_1\left\|\nabla F(w)\right\| + \tau_2 \quad a.s. \quad \forall \xi \sim \mathbb{P}. \tag{15}$$

Assumption 4 allows the approximation error of stochastic gradient scaling with the full gradient norm on a per-sample basis, while only requiring the error to remain bounded at stationary points. Compared to other noise assumptions, Assumption 4 is much weaker than the almost sure error bound (i.e., $\tau_1 = 0$) adopted in Zhang et al. (2019; 2020a); Liu et al. (2022b) but stronger than expected error bound (i.e., bounded variance assumption $\mathbb{E}_\xi\|\nabla f(w) - \nabla F(w)\|^2 \le \tau_2^2$) adopted in Koloskova et al. (2023); Reisizadeh et al. (2023). Our motivation for studying the convergence of IAN-SGD under this noise assumption is to strike a balance between achieving standard convergence rates (i.e., $\mathcal{O}(\epsilon^{-4})$) under relaxed noise conditions, while keeping the batch size at the $\Omega(1)$-level. As later we show in Theorem 2, IAN-SGD achieves $\mathcal{O}(\epsilon^{-4})$ sample complexity under Assumption 4.

**Remark 3** *Existing work (Koloskova et al., 2023) has shown that when adopting $\Omega(1)$ sample per iteration, clipped SGD attains $\mathcal{O}(\epsilon^{-5})$ sample complexities under generalized-smooth and bounded variance assumptions. In order to achieve the standard $\mathcal{O}(\epsilon^{-4})$ sample complexity under such assumption, Reisizadeh et al. (2023) demonstrated that clipped SGD requires $\Omega(\epsilon^{-2})$ samples per iteration. Through this way, the constant arising from the variance of the approximation error can be controlled at the $\mathcal{O}(\epsilon^2)$-level. This, in turn, relaxes the burden on iteration complexity, requiring only $\mathcal{O}(\epsilon^{-2})$ iterations for averaging during convergence proof. We leave technical discussions on optimizing the batch size while maintaining $\mathcal{O}(\epsilon^{-4})$ total sample complexity of IAN-SGD for future work.*

Based on these assumptions, we first obtain the following upper bounds regarding $\|\nabla F(w_t)\|$, $\|\nabla f_\xi(w_t)\|$. See Proof 6 in Appendix G.

**Lemma 2** *Let Assumptions 3 and 4 hold. The gradient $\|\nabla F(w_t)\|$ and stochastic gradient $\nabla f_\xi(w_t)$ satisfy*

$$\|\nabla F(w_t)\| \leq \frac{1}{1-\tau_1}\|\nabla f_\xi(w_t)\| + \frac{\tau_2}{1-\tau_1}, \, and$$

$$\|\nabla f_\xi(w_t)\| \leq (1+\tau_1)\|\nabla F(w_t)\| + \tau_2. \tag{16}$$

Lemma 2 indicates that the full gradient norm can be upper bounded by the stochastic gradient plus a constant. This result suggests that for $h_t$ in equation 14, one can choose $A = \frac{1}{\tau_1-1}$ and $\delta = \frac{\tau_2}{1-\tau_1}$, and it is very useful in our convergence analysis to effectively bound the stochastic gradient norm in $h_t$. We obtain the following convergence result of IAN-SGD. See Proof 8 in Appendix I

**Theorem 2 (Convergence of IAN-SGD)** *Let Assumptions 1, 3 and 4 hold. For the IAN-SGD algorithm, choose learning rate $\gamma = \min\{\frac{1}{4L_0(2\tau_1^2+1)}, \frac{1}{4L_1(2\tau_1^2+1)}, \frac{1}{\sqrt{T}}, \frac{1}{8L_1(2\tau_1^2+1)(2\tau_2/(1-\tau_1))^\beta}\}$, and $A = \frac{1}{1-\tau_1}$ $\delta = \frac{\tau_2}{1-\tau_1}$, $\Gamma = (4L_1\gamma(2\tau_1^2+1))^{\frac{1}{\beta}}$. Denote $\Lambda := F(w_0) - F^* + \frac{1}{2}(L_0 + L_1)(1 + 4\tau_2^2)^2$. Then, with probability at least $\frac{1}{2}$, IAN-SGD produces a sequence satisfying $\min_{t\leq T}\|\nabla F(w_t)\| \leq \epsilon$ if the total number of iteration $T$ satisfies*

$$T \geq \Lambda \max\left\{\frac{256\Lambda}{\epsilon^4}, \frac{64L_1(2\tau_1^2+1)(2+2\tau_1)^\beta}{(1-\tau_1)^\beta\epsilon^{2-\beta}}, (2\tau_1^2+1)\cdot\frac{64(L_0+L_1)+128L_1(2\tau_2/(1-\tau_1))^\beta}{\epsilon^2}\right\}. \tag{17}$$

**Remark 4** *Existing work (Zhang et al., 2019) studied the iteration complexity of clipped SGD (i.e., $\beta = 1$ and $\xi' = \xi$, without independence sampling) in the special bounded noise setting $\tau_1 = 0$, and obtained the complexity*

$$T \geq \Lambda \max\left\{\frac{4\Lambda}{\epsilon^4}, \frac{128L_1}{\epsilon}, \frac{80L_0+512L_1\tau_2}{\epsilon^2}\right\}, \tag{18}$$

*where $\Lambda = \left(F(x_0) - F^* + (5L_0 + 2L_1\tau_2)\tau_2^2 + 9\tau_2L_0^2/L_1\right)$. As a comparison, the above Theorem 2 establishes a comparable complexity result under the more general noise model ($\tau_1 \geq 0$) in Assumption 4, by leveraging the adaptive gradient normalization and independent sampling of IAN-SGD. In particular, in the special setting of $\tau_1 = 0, \beta = 1$, these two complexity results match up to absolute constant coefficients.*

*Another existing work Zhang et al. (2020a) analyzed a generalized clipping framework that combines momentum acceleration and gradient clipping, namely $w_{t+1} = w_t - \left(\nu\min\left\{c, \frac{\gamma}{\|m_t\|}\right\}m_t + (1-\nu)\min\left\{c, \frac{\gamma}{\|\nabla f_\xi(w_t)\|}\right\}\nabla f_\xi(w_t)\right)$, under a bounded noise setting (i.e., $\tau_1 = 0$). They establish a complexity of $\mathcal{O}\left((F(w_0) - F^*)\tau_2^2 L_0/\epsilon^4\right)$ when $\gamma/c = \Theta(\tau_2)$, where $m_t$ denotes the momentum term and $c$ is the clipping threshold. This framework enables them to construct a recursion based directly on $\|\nabla F(w_t)\|$, whereas our analysis is built around recursive bounds involving $\|\nabla F(w_t)\|^2$ or $\|\nabla F(w_t)\|^{2-\beta}$ at each step. Consequently, while both approaches yield the same order of convergence in terms of $\epsilon$, the complexity expressions slightly differ in the numerators due to different recursions.*

Compared to the existing studies (Zhang et al., 2019; 2020b; Reisizadeh et al., 2023; Zhang et al., 2020a; Koloskova et al., 2023) on gradient clipping algorithms, Theorem 2 establishes convergence guarantees without the bounded approximation error assumption and the use of extremely large batch sizes that depend on

the target error $\epsilon$. Through numerical experiments in Section 7 and ablation study in A.2 later, we show that it suffices to query a small number of independent samples for IAN-SGD in practice.

### 6.3.1 Proof outline and novelty

The independent sampling strategy adopted by IAN-SGD naturally decouples stochastic gradient from gradient norm normalization, making it easier to achieve the desired optimization progress in generalized-smooth optimization under relaxed conditions. By the descent lemma, we have that

$$
\begin{aligned}
&\mathbb{E}_{\xi,\xi'}\big[F(w_{t+1}) - F(w_t)|w_t\big] \\
&\overset{(i)}{\leq} \mathbb{E}_{\xi'}\Big[ -\frac{\gamma\|\nabla F(w_t)\|^2}{h_t^\beta} + \frac{\gamma^2(L_0 + L_1\|\nabla F(w_t)\|^\alpha)}{2h_t^{2\beta}}\mathbb{E}_\xi\big[\|\nabla f_\xi(w_t)\|^2\big]|w_t\Big] \\
&\overset{(ii)}{\leq} \mathbb{E}_{\xi'}\Big[\Big(\frac{\gamma}{h_t^\beta}\big(-1 + \gamma(2\tau_1^2+1)\frac{L_0 + L_1\|\nabla F(w_t)\|^\alpha}{h_t^\beta}\big)\Big)\|\nabla F(w_t)\|^2 + \gamma^2\frac{2\tau_2^2(L_0 + L_1\|\nabla F(w_t)\|^\alpha)}{h_t^{2\beta}}|w_t\Big],
\end{aligned} \quad (19)
$$

where (i) follows from the descent lemma (conditioned on $w_t$) and the independence between $\xi$ and $\xi'$; (ii) leverages Assumption 4 to bound the second moment $\mathbb{E}_\xi[\|\nabla f_\xi(w_t)\|^2]$ by $(4\tau_1^2 + 2)\|\nabla F(w_t)\|^2 + 4\tau_2^2$. Then, for the first term in equation 19, we leverage the clipping structure of $h_t$ to bound the coefficient $\gamma(2\tau_1^2+1)(L_0 + L_1\|\nabla F(w)\|^\alpha)/h_t^\beta$ by $\frac{1}{2}$. For the second term in equation 19, we again leverage the clipping structure of $h_t$ and consider two complementary cases: when $\|\nabla F(w_t)\| \leq \sqrt{1 + 4\tau_2^2/(2\tau_1^2 + 1)}$, this term can be upper bounded by $\frac{1}{2}\gamma^2(L_0 + L_1)(1 + 4\tau_2^2)^2$; when $\|\nabla F(w_t)\| > \sqrt{1 + 4\tau_2^2/(2\tau_1^2 + 1)}$, this term can be upper bounded by $\frac{\gamma}{4h_t^\beta}\|\nabla F(w_t)\|^2$. Summing them up gives the desired bound. We refer to Lemma 5 in the appendix for more details. Substituting these bounds into equation 19 and rearranging the terms yields that

$$
\mathbb{E}_{\xi'}\big[\frac{\gamma}{4h_t^\beta}\|\nabla F(w_t)\|^2\big] \leq \mathbb{E}_{\xi,\xi'}\big[F(w_t) - F(w_{t+1})\big] + \frac{1}{2}(L_0 + L_1)\gamma^2(1 + 4\tau_2^2)^2.
$$

Furthermore, by leveraging the clipping structure of $h_t^\beta$ and Assumption 4, the left hand side of above inequality can be lower bounded as $\frac{\gamma\|\nabla F(w_t)\|^2}{h_t^\beta} \geq \min\{\gamma\|\nabla F(w_t)\|^2, \frac{\|\nabla F(w_t)\|^{2-\beta}}{4((2\tau_2+2)/1-\tau_1)^\beta L_1(2\tau_1^2+1)}\}$. Finally, telescoping the above inequalities over $t$ and taking expectation leads to the desired bound in equation 17.

As a comparison, in the previous works on clipped SGD (Zhang et al., 2019; 2020a), their stochastic gradient and normalization term $h_t$ depend on the same mini-batch of samples, and therefore cannot be treated separately in the analysis. For example, their analysis proposed the following decomposition.

$$
\mathbb{E}_\xi\Big[\frac{\|\nabla f_\xi(w_t)\|^2}{h_t^{2\beta}}\Big] = \mathbb{E}_\xi\Big[\big(\|\nabla F(w_t)\|^2 + \|\nabla f_\xi(w_t) - \nabla F(w_t)\|^2 + 2\langle\nabla F(w_t), \nabla f_\xi(w_t) - \nabla F(w_t)\rangle\big)/h_t^{2\beta}\Big].
$$

Hence their analysis needs to assume a constant upper bound for the approximation error $\|\nabla f_\xi(w_t) - \nabla F(w_t)\|$ in order to obtain a comparable bound to ours.

### 6.3.2 IAN-SGD under generalized PŁ condition

Under Assumption 2, we obtain the following recursion for IAN-SGD. See Appendix J for Proof details.

**Lemma 3 (Descent inequality of IAN-SGD under generalized PŁ condition)** *Let Assumptions 1, 2, 3 and 4 hold. For the IAN-SGD algorithm, choose target error $0 < \epsilon \leq \min\{1, 1/2\mu\}$ and learning rate $\gamma = (2\mu\epsilon)^{\frac{4-2\beta}{\rho}} \cdot \min\{\frac{1}{4L_0(2\tau_1^2+1)}, \frac{1}{4L_1((2\tau_1+2)/(1-\tau_1))(2\tau_1^2+1)}, \frac{1}{8L_1(2\tau_1^2+1)(2\tau_2/(1-\tau_1))^\beta}, \frac{L_1(2\tau_1^2+1)}{16((L_0+L_1)(1+4\tau_2^2)+L_1(\tau_1^2+1/2))^2}\}$, and $A = \frac{1}{1-\tau_1}$ $\delta = \frac{\tau_2}{1-\tau_1}$, $\Gamma = (4L_1\gamma(2\tau_1^2+1))^{\frac{1}{\beta}}$. Then, we have the following descent inequality*

$$
\mathbb{E}_{w_{t+1}}[\Delta_{t+1}] \leq \mathbb{E}_{w_t}[\Delta_t - \frac{\gamma^{3/2}(L_1(2\tau_1^2+1))^{1/2}(2\mu\Delta_t)^{(2-\beta)/\rho}}{4} + \frac{\gamma^{3/2}(L_1(2\tau_1^2+1))^{1/2}(2\mu\epsilon)^{(2-\beta)/\rho}}{8}]. \quad (20)
$$

We notice that equation 20 is almost the same as the recursion of AN-GD obtained in Appendix D. The main difference between equation 20 and equation 35 is the usage of $\gamma^{\frac{3}{2}}$ instead of $\gamma$ to characterize the recursion.

As a result, the convergence rate of IAN-SGD is expected to be much slower than AN-GD, i.e., IAN-SGD converge with rate $\tilde{\mathcal{O}}(\epsilon^{-\frac{3\beta}{\rho}})$ when $\rho + \beta = 2$; IAN-SGD converge with rate $\mathcal{O}(\epsilon^{-\frac{3(2-\rho)}{\rho}})$ when $\rho + \beta < 2$. We omit the discussions of proof as its convergence analysis exactly follows the proof presented in Appendix E.

## 7   Experiments

We conduct numerical experiments to compare IAN-SGD with other state-of-the-art stochastic algorithms, including the standard SGD (Ghadimi & Lan, 2013), normalized SGD (NSGD), clipped SGD (Zhang et al., 2019), SPIDER (Fang et al., 2018), normalized SGD with momentum (Cutkosky & Mehta, 2020)(NSGDm) etc. Compared with previous works that study clipped SGD (Zhang et al., 2019; 2020a;b; Koloskova et al., 2023; Reisizadeh et al., 2023), the main change of IAN-SGD lies in its usage of independent sampling and adaptive normalization. To empirically validate the effect of adaptive normalization in accordance with Theorem 1, our experiments focus on nonconvex phase retrieval, distributionally robust optimization (DRO), and deep neural network training. Prior works (Zhou et al., 2016; Jin et al., 2021; Liu et al., 2022a; Scaman et al., 2022) have shown that phase retrieval and deep neural networks satisfy generalized PŁ conditions under mild assumptions. Accordingly, we evaluate the convergence of all algorithms in terms of total sample complexity (for phase retrieval and DRO) and number of epochs (for deep neural networks). Furthermore, to align the DRO formulation with our theoretical assumptions, we focus on regression tasks and adopt the $\chi^2$-divergence to model distributional uncertainty. All experiments were conducted on a PC computer equipped with a 24-core CPU, 32GB of RAM, and a single NVIDIA RTX4090 GPU, running Python 3.8.

### 7.1   Nonconvex Phase Retrieval

The phase retrieval problem arises in optics, signal processing, and quantum mechanics (Drenth, 2007). The goal is to recover a signal from measurements where only the intensity is known, and the phase information is missing or difficult to measure. Specially, denote the underlying object as $w \in \mathbf{R}^d$. Suppose we take $m$ intensity measurements $y_r = |a_r^T w|^2 + n_r$ for $r = 1 \cdots m$, where $a_r$ denotes the measurement vector and $n_r$ is the additive noise. We aim to reconstruct $w$ by solving the following regression problem,

$$\min_{w \in \mathbf{R}^d} f(w) = \frac{1}{2m} \sum_{r=1}^{m} \left( y_r - |a_r^T w|^2 \right)^2. \tag{21}$$

Such nonconvex function satisfies generalized-smooth with parameter $\alpha = \frac{2}{3}$. In this experiment, we generate the initialization $w_0 \sim \mathcal{N}(1,6)$ and the underlying true object $w^* \sim \mathcal{N}(0, 0.5)$ with dimension $d = 100$. We take $m = 3k$ measurements with $a_r \sim \mathcal{N}(0, 0.5)$ and $n_r \sim \mathcal{N}(0, 16)$. We implement all of the stochastic algorithms in original form as described in previous literature. We set batch size $|B| = 64$, and for IAN-SGD, we choose a small independent batch size $|B'| = 4$. We fine-tuned learning rate for all algorithms, i.e., $\gamma = 5e^{-5}$ for SGD, $\gamma = 0.2$ for NSGD and NSGD with momentum, $\gamma = 0.6$ for clipped SGD, and $\gamma = 0.25$ for SPIDER and IAN-SGD. We set the maximal gradient clipping constant as 20, $\delta = 1e^{-3}$ for both clipped SGD and IAN-SGD. And we set normalization parameter $\beta = \frac{2}{3}$. Figure 2 (left) shows the comparison of objective value versus sample complexity. It can be observed that IAN-SGD consistently converges faster than other baseline algorithms.

### 7.2   Distributionally-Robust Optimization

Distributionally-robust optimization (DRO) is a popular approach to enhance robustness against data distribution shift. We consider the regularized DRO problem $\min_{w \in \mathbf{R}^d} f(w) = \sup_{\mathbb{Q}} \left\{ \mathbb{E}_{\xi \sim \mathbb{Q}}[\ell_\xi(w)] - \lambda \phi(\mathbb{P}; \mathbb{Q}) \right\}$, where $\mathbb{Q}$, $\mathbb{P}$ represents the underlying distribution and the nominal distribution respectively. $\lambda$ denotes a regularization hyper-parameter and $\phi$ denotes $\phi$-divergence metric. Under mild technical assumptions, Jin et al. (2021) showed that such a problem has the following equivalent dual formulation

$$\min_{w \in \mathbf{R}^d, \eta \in \mathbf{R}} L(w, \eta) = \lambda \mathbb{E}_{\xi \sim P} \phi^* \left( \frac{\ell_\xi(w) - \eta}{\lambda} \right) + \eta, \tag{22}$$

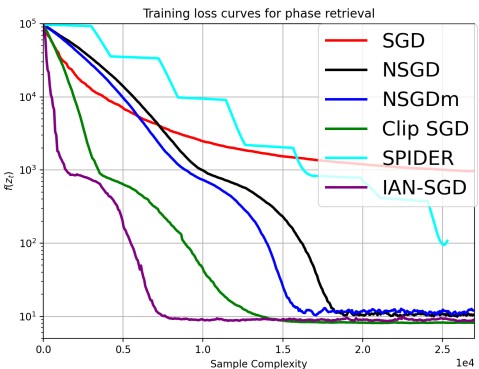 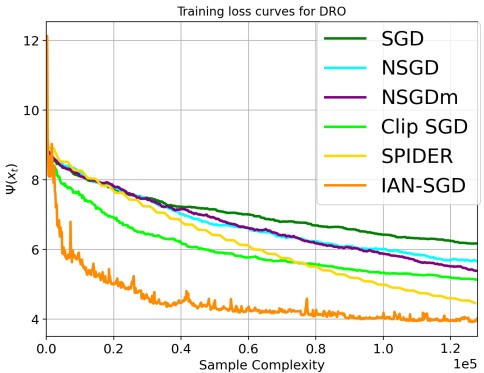

Figure 2: Experimental Results on Phase Retrieval and DRO

where $\phi^*$ denotes the convex conjugate function of $\phi$. In particular, such dual objective function satisfies the generalized-smooth condition 2 with parameter $\alpha = 1$ (Jin et al., 2021; Chen et al., 2023). In this experiment, we use the Life Expectancy data (Arshi, 2017) for regression task. For objective equation 22, we set $\lambda = 0.01$, select $\phi^*(t) = \frac{1}{4}(t+2)_+^2 - 1$, which is the convex conjugate of $\chi^2$-divergence. We adopt the regularized loss $\ell_\xi(w) = \frac{1}{2}(y_\xi - x_\xi^\top w)^2 + 0.1 \sum_{j=1}^{34} \ln(1 + |w^{(j)}|)$, where $x_\xi, y_\xi$ represent input and output data, respectively. And we optimize the primal and dual variable of $L(w, \eta)$ in parallel following conclusion drawn from Jin et al. (2021).

For the stochastic momentum moving average parameter used for acceleration methods, we set it as 0.1 and 0.25 for NSGD with momentum and SPIDER respectively. For stochastic algorithms without usage of multiple mini-batches, i.e., SGD, NSGD, NSGD with momentum and clipped SGD, we set their batch sizes as $|B| = 128$. For SPIDER, we set $|B| = 128$ and $|B'| = 2313$, where the algorithm will conduct a full-gradient computation after every 15 iterations. For IAN-SGD, we set the batch size for two batch samples as $|B| = 128$ and $|B'| = 8$. We fine-tuned the learning rate for each algorithm, i.e., $\gamma = 4e^{-5}$ for SGD, $\gamma = 5e^{-3}$ for NSGD, NSGD with momentum and SPIDER, $\gamma = 0.11$ for clipped SGD and IAN-SGD. We set the $\delta = 1e^{-1}$, maximal gradient clipping constant as 30, 25 for clipped SGD and IAN-SGD respectively. And we set normalization parameter $\beta = \frac{2}{3}$. Figure 2 (right) shows the comparison of objective value versus sample complexity. It can be observed that objective value optimized by IAN-SGD consistently converges faster than other baselines algorithms.

## 7.3 Deep Neural Networks

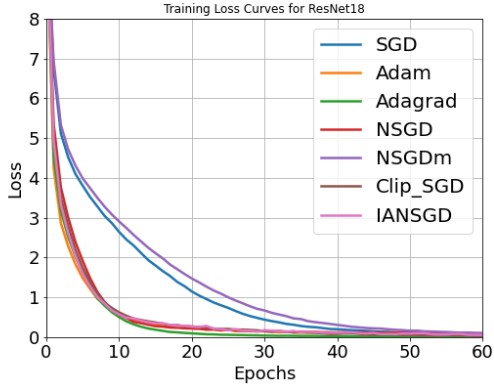 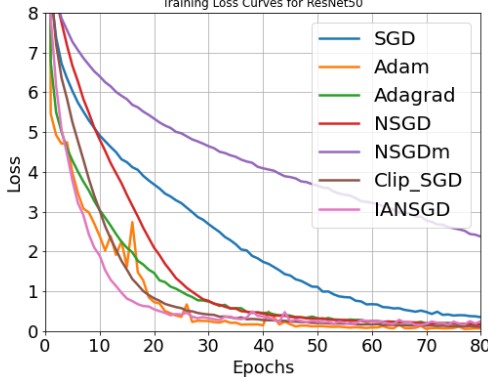

Figure 3: Experimental Result on training ResNet18, ResNet50.

According to Zhang et al. (2019), generalized-smooth condition 2 has been observed to hold in deep neural networks. To further demonstrate the effectiveness of IAN-SGD algorithm, we conduct experiments for training deep neural networks. Specially, we train ResNet18 and ResNet50 (He et al., 2016) on CIFAR10 Dataset (Krizhevsky, 2009) from scratch. We resize images as $32 \times 32$ and normalize images with standard derivation equals to 0.5 on each dimension. At the beginning of each algorithm, we fix the random seed and initialize model parameters using the Kaiming initialization (He et al., 2015). We compare our algorithm with several baseline methods, including SGD (Robbins & Monro, 1951), Adam (Kingma, 2014), Adagrad (Duchi et al., 2011a), NSGD, NSGD with momentum (Cutkosky & Mehta, 2020) and clipped SGD (Zhang et al., 2019).

For SGD, Adam and Adagrad, we utilize PyTorch built-in optimizer (Paszke et al., 2019) to implement training pipelines. We implement training pipelines for NSGD, NSGD with momentum, clipped SGD and IAN-SGD. The normalization constant is computed through all model parameters at each iteration. The detailed algorithm settings are as follows. For batch size, all algorithms use $B = 128$, and $B' = 32$ for IAN-SGD. For stochastic momentum moving average parameter, we set it as 0.9, 0.99 for Adam, and 0.25 for normalized SGD with momentum. For clipping threshold used in clipped SGD and IAN-SGD, we set them as 2 and $\delta = 1e^{-1}$. The normalization power used for IAN-SGD is $\beta = \frac{2}{3}$. We fine-tuned learning rate for all algorithms, i.e., $\gamma = 1e^{-3}$ for SGD, Adam and Adagrad, $\gamma = 1e^{-1}$ for NSGD and NSGD with momentum, $\gamma = 2e^{-1}$ for clipped SGD and IAN-SGD. We trained ResNet18, ResNet50 on CIFAR10 dataset for $60, 80$ epochs and plot the training loss in Figure 3. Figure 3 (left) shows the training loss of ResNet18, Figure3 (right) shows the training loss of ResNet50. As we can see from these figures, the (pink) loss curve optimized by IAN-SGD indicates fast convergence rate comparable with several baselines, including SGD, NSGD NSGDm, clipped SGD, which demonstrate the effectiveness of IAN-SGD framework.

Based on our experiments on non-convex phase retrieval, distributionally robust optimization (DRO), deep neural networks, and an ablation study A.1, we found that IAN-SGD performs best when the adaptive normalization parameter is set to $\beta = \frac{2}{3}$. When $\beta$ is too small, i.e., $\beta \leq \frac{1}{2}$, IAN-SGD struggles to converge; when $\beta$ is close to 1, its performance resembles that of clipped SGD. To achieve fast convergence, the learning rate for IAN-SGD should lie between $1e^{-1}$ and $2e^{-1}$. Users should carefully fine-tune this parameter, as overly large learning rates may lead to divergence.

Regarding the numerical stabilization term $\delta$, our ablation study (see Appendix A.3) shows that IAN-SGD is empirically insensitive to its value; small value such as $1e^{-3}$ is sufficient to ensure convergence. Finally, to balance stable performance and fast convergence, ablation results (see Appendix A.2) suggest that user should set batch size $B \geq 16$ and the independent sample batch size $|B'|$ should be at least one-quarter of batch size $|B|$.

# 8 Conclusion

In this work, we provide theoretical insights on how normalization interplays with function geometry, and their overall effects on convergence. We then propose independent normalized stochastic gradient descent for stochastic setting, achieving same sample complexity under relaxed assumptions. Our results extend the existing boundary of first-order nonconvex optimization and may inspire new developments in this direction. In the future, it is interesting to explore if the popular acceleration method such as stochastic momentum and variance reduction can be combined with independent sampling and normalization to improve the sample complexity.

# 9 Acknowledgements

We are grateful to the anonymous reviewers and Dr. Ziyi Chen for insightful comments that led to several improvements in the presentation and technical details. This research was supported in part by the Air Force Research Laboratory Information Directorate (AFRL/RI), through AFRL/RI Information Institute®, contract number FA8750-20-3-1003. The work of Yufeng Yang and Yi Zhou was supported by the National Science Foundation under grants DMS-2134223, ECCS-2237830. The work of Shaofeng Zou was supported

in part by National Science Foundation under Grants CCF-2438429 and ECCS-2438392(CAREER). The work of Yifan Sun was supported in part by the ONR grant W911NF-22-1-0292.

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

**Appendix Table of Contents**

# A Ablation Study

In order to have a comprehensive understanding of the performance of IAN-SGD in practical problems, we conduct ablation studies regarding on important components of IAN-SGD, including adaptive normalization $\beta$, batch size $|B'|$ of independent samples and numerical stabilizer $\delta$.

## A.1 Effects of $\beta$

To justify the advantage of using adaptive normalization, we conduct the following two experiments.

In the first experiment, we choose a unified normalization parameter $\beta = \frac{2}{3}$ for all the normalized methods, i.e., NSGD, NSGD with momentum, clipped SGD, SPIDER and IAN-SGD. To guarantee convergence, we fine-tuned the learning rate accordingly, i.e., $\gamma = 0.03$ for NSGD and NSGD with momentum, $\gamma = 0.05$ for SPIDER, $\gamma = 0.17$ for both clipped SGD and IAN-SGD. To make a fair comparison, we keep other parameters unchanged. Figure 4 (left) shows the comparison of objective value versus sample complexity for the Phase Retrieval problem. It can be observed that, by adjusting $\beta = \frac{2}{3}$, the objective value optimized by all algorithms decreases much faster compared with Figure 2. This indicates that adaptive normalization can accelerate convergence. Moreover, compared with other normalization methods, although IAN-SGD requires additional sampling at each iteration, the training loss still decreases faster than NSGD SGD with momentum and SPIDER.

Similarly, for DRO, we unify the normalization parameter of all the normalized methods, i.e., NSGD, NSGD with momentum, clipped SGD, SPIDER and IAN-SGD with $\beta = \frac{2}{3}$. To guarantee algorithm convergence, we fine-tuned learning rate accordingly. For NSGD, NSGDm and SPIDER, we keep learning rate unchanged; For clipped SGD, and IAN-SGD, we set $\gamma = 0.08$. To make a fair comparison, we keep other parameters unchanged. Figure 4 (right) shows the comparison of objective value versus sample complexity. It can be observed that, by setting $\beta = \frac{2}{3}$, the objective value optimized by all normalization methods decreases faster than Figure 2. This verifies the effectiveness of adaptive normalization. Although IAN-SGD requires additional sampling, it converges faster than other normalized methods.

In summary, the results in Figure 4 indicate that independent sampling and adaptive normalization doesn't increase the sample complexity to find a stationary point, which justifies IAN-SGD framework's effectiveness when dealing with nonconvex geometry characterized by generalized-smooth condition.

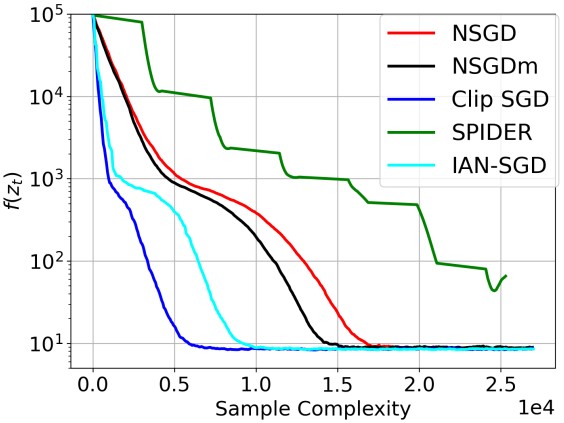 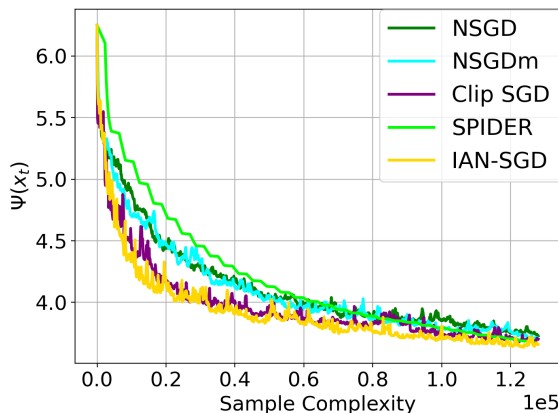

Figure 4: Advantage of using adaptive normalization on normalized first-order algorithms

In the second experiment, we vary $\beta$ for IAN-SGD and keep other parameters unchanged. Figure 5 shows the convergence result with different $\beta$ for Phase Retrieval and DRO. It can be observed that decreasing $\beta$ accelerates convergence in general. However, small $\beta$ can make convergence unstable. In the DRO experiment

shown in Figure 5 (right), we observed that when $\beta = \frac{3}{5}$, the objective value curve fluctuates a lot. If $\beta$ is smaller than $\frac{3}{5}$, the algorithm fails to converge. This phenomenon matches the implications of Theorem 1, where $\beta$ must satisfy $\beta \in [\alpha, 1]$ to guarantee convergence.

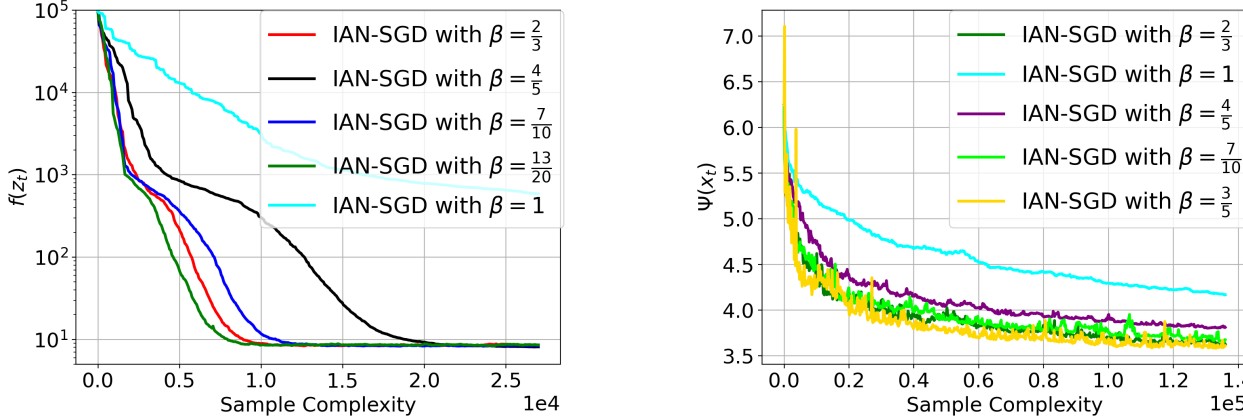

Figure 5: Effects of adaptive normalization on convergence of IAN-SGD

## A.2 Effects of Independent Samples' Batch Size

In this section, we study the effects of batch size for independent samples used in IAN-SGD. For Phase retrieval, we keep $|B| = 64$ and other parameters same as section A.1, and we vary independent batch sizes to be $|B'| = \{4, 8, 16, 32, 64\}$. Similarly, for DRO, we keep $|B| = 128$ and other parameters same as section A.1, and we vary batch size of independent samples $|B'| = \{16, 32, 64, 128\}$.

The following Figure 6 shows the convergence of IAN-SGD under different batch size choices for Phase Retrieval (left) and DRO (right). It can be observed that for Phase retrieval and DRO problem, small batch

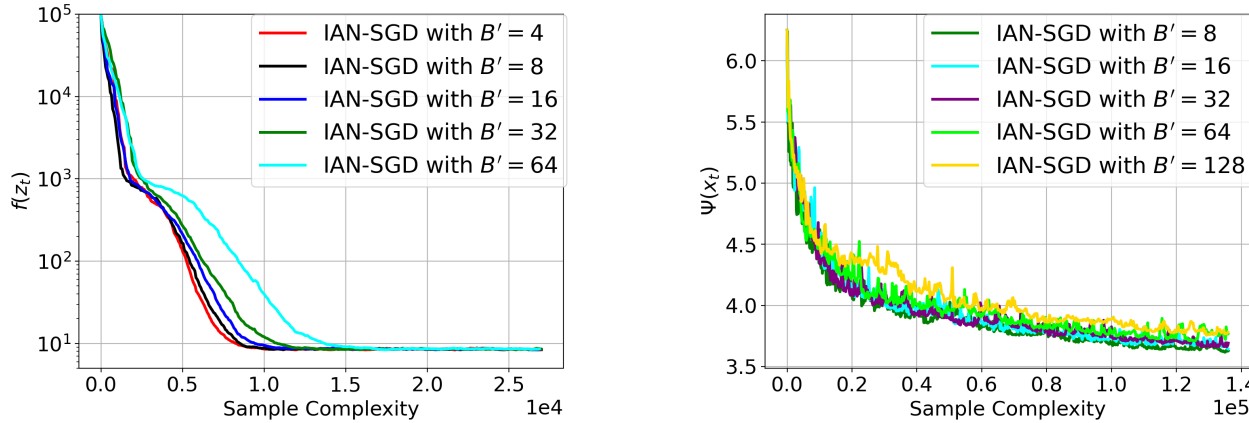

Figure 6: Effects of independent samples' batch size on convergence

sizes such as $|B'| = \{4, 8\}$ are sufficient to guarantee convergence, whereas large batch sizes of $|B'|$ do not significantly increase the sample complexity, which verifies the effectiveness of independent sampling.

### A.3 Effects of $\delta$

In Theorem 2, we set $\delta = \frac{\tau_2}{1-\tau_1}$ based on Assumption 4, which assumes an affine bound for approximation error $\|\nabla f_\xi(w) - \nabla F(w)\|$. This assumption relaxes the strong assumption used in Zhang et al. (2019); Liu et al. (2022b); Zhang et al. (2020a), where they assume that the approximation error $\|\nabla f_\xi(w) - \nabla F(w)\|$ is upper bounded by a constant. The major weakness is that if certain samples have outlier gradients, such constant upper bound can be very large. Thus, to verify the effectiveness of Assumption 4, we expect convergence of IAN-SGD under a wide range of $\delta$, especially for small $\delta$.

To study the effect of $\delta$, we keep other parameters the same as mentioned in Section A.1 and only vary $\delta$. For Phase Retrieval, we vary $\delta = \{1e^{-8}, 1e^{-3}, 1e^{-1}, 1, 10\}$ and Figure 7 (left) shows the corresponding convergence result. For DRO, we vary $\delta = \{1e^{-8}, 1e^{-3}, 1e^{-1}, 1, 10\}$ and Figure 7 (right) shows the convergence result. We observe that IAN-SGD convergence is robust to the choice of $\delta$. But $\delta = 1e^{-3}, 1e^{-1}$ demonstrate faster convergence than others for Phase retrieval and DRO experiments. This result indicates that a small $\delta$ is sufficient to guarantee convergence, which verifies the effectiveness of assumption 4 and IAN-SGD when dealing with stochastic nonconvex generalized-smooth geometry.

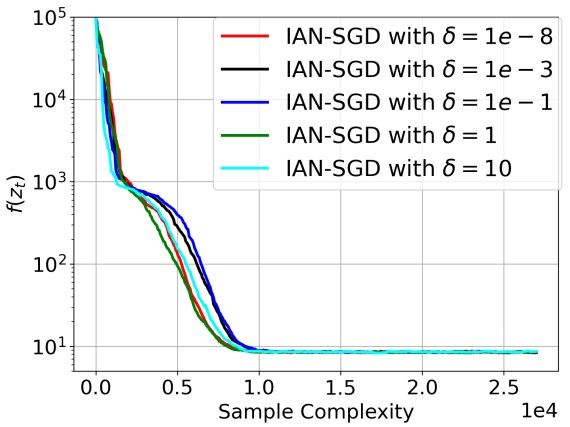 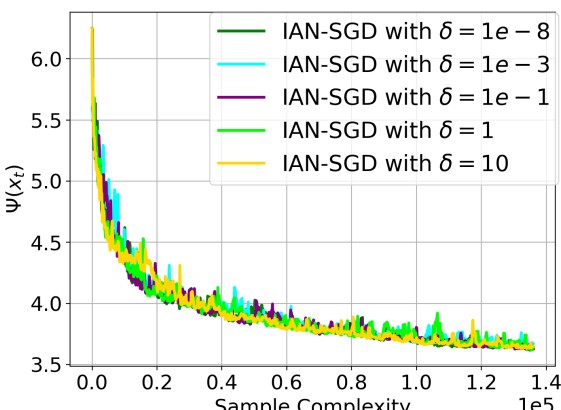

Figure 7: Effects of $\delta$ on convergence

### A.3.1 Effects of $\tau_1$, $\tau_2$ on convergence

In Theorem 2, we set $A = \frac{1}{1-\tau_1}$ and $\delta = \frac{\tau_2}{1-\tau_1}$, making the sample complexity dependent on both $\tau_1$ and $\tau_2$. To further investigate the effect of the gradient noise assumption, we perform experiments that allow flexible choices of $\tau_1$ and $\tau_2$. Specifically, we control the stochastic gradient noise by computing gradients over mini-batch samples with $B = B' = 16$ at each iteration. We then evaluate IAN-SGD under various noise settings with $\tau_1 \in \{0, 0.5, 0.8\}$ and $\tau_2 \in \{10^{-8}, 10^{-3}, 10^{-1}, 1, 10\}$. As a result, the scaling factor becomes $A \in \{1, 2, 5\}$ and we define $\delta = A \cdot \{10^{-8}, 10^{-3}, 10^{-1}, 1, 10\}$ accordingly. For phase retrieval, we use a unified learning rate $\gamma = 1e^{-1}$, and for DRO, we set $\gamma = 5e^{-3}$. Other parameters, such as the clipping threshold and $\beta$, remain unchanged. Figure 8 plots the results, where the first row shows IAN-SGD's convergence for phase retrieval, and the second row shows IAN-SGD's convergence for DRO.

As we can see from these figures, IAN-SGD converges slightly slower as $\tau_1$ increases for both the phase retrieval and DRO problems. This observation aligns with Theorem 2, which shows that when $\tau_1 > 0$, the resulting complexity is slightly worse than in the case $\tau_1 = 0$. Regarding $\tau_2$, although the final convergence intervals remain approximately the same, we find that smaller values lead to less stable convergence for phase retrieval. For DRO, the variation in $\tau_2$ does not significantly affect convergence when learning rate is unified. This might be because for the DRO experiment, the model initialization and data processing have reduced the noise of the stochastic gradients, enabling convergence with flexible choices of $\tau_2$.

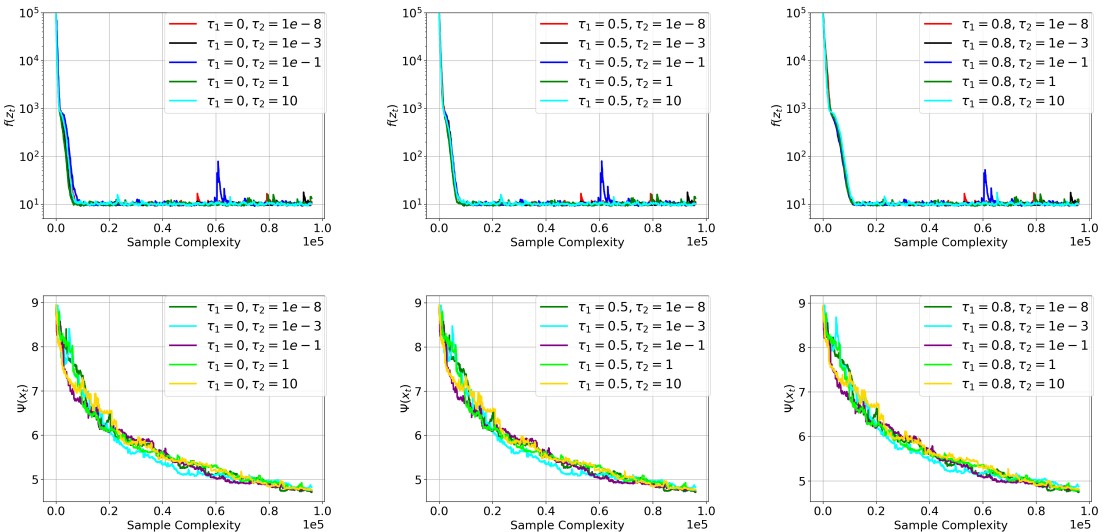

Figure 8: Effects of $\tau_1, \tau_2$ for IAN-SGD convergence

## A.4 Ablation Study on Extreme Small Batch Size $|B|, |B'|$

In this subsection, we study the effects of batch size in IAN-SGD. Specially, we want to justify if IAN-SGD works with $B, B' = \Omega(1)$ in practice. For phase retrieval, we set $B = B'$ and keep other parameters the same as in section A.1. For comparison, we vary batch sizes to be $\{1, 2, 4, 8, 16, 32\}$. To guarantee algorithm converging to same level of magnitude, we fine-tune learning rates accordingly, i.e., for each batch size, we set $\gamma = \{5e^{-4}, 5e^{-3}, 2e^{-2}, 4e^{-2}, 0.10, 0.10\}$, respectively. Similarly, for DRO, we set $|B| = |B'|$ and keep other parameters the same as section A.1. For comparison, we vary batch size to be $\{1, 2, 4, 8, 16, 32\}$, and we fine-tuned learning rates accordingly, i.e., for each batch size, we set $\gamma = \{5e^{-4}, 1e^{-3}, 1e^{-3}, 5e^{-3}, 5e^{-3}, 1e^{-2}\}$, respectively. Figure 6 plots the convergence of IAN-SGD under different batch size choices for Phase Retrieval (left) and DRO (right). We conclude IAN-SGD converges under constant-level batch size $B, B' = \Omega(1)$.

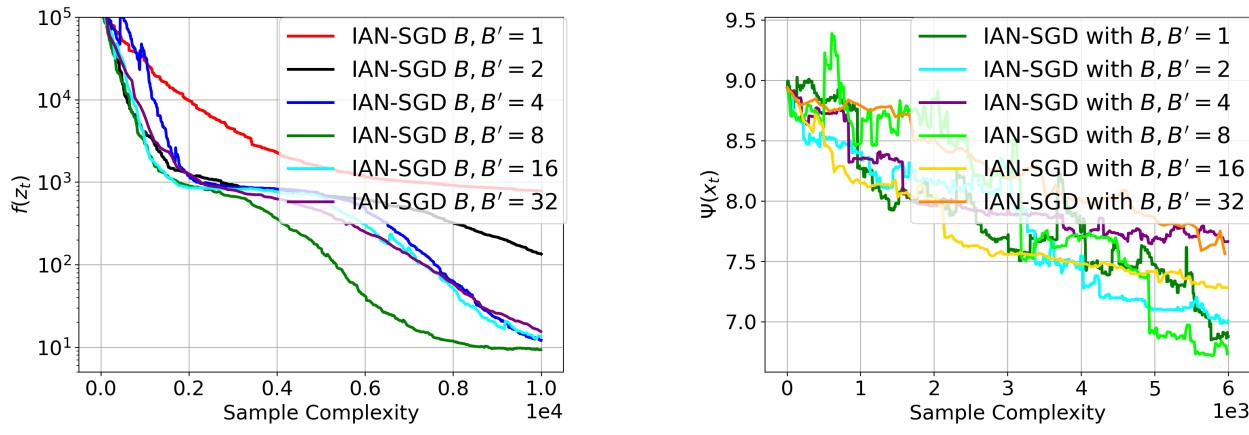

Figure 9: Effects of batch size on convergence for IAN-SGD

Next, we compare IAN-SGD with other baseline methods under small batch size. To make a trade-off between convergence and stability, we set $|B| = 2$ for SGD, NSGD with momentum, Clip-SGD and SPIDER; For SPIDER algorithm (Fang et al., 2018), we set epoch size to be 20. For phase retrieval, we fine-tuned learning

rate for all algorithms, i.e., $\gamma = 1e^{-5}$ for SGD, $\gamma = 1e^{-2}$ for NSGD, NSGD with momentum, $\gamma = 5e^{-3}$ for clipped SGD and IAN-SGD, $\gamma = 5e^{-2}$ for SPIDER. Similarly, for DRO, we set $|B| = 2$ for SGD, NSGD with momentum, clipped SGD and SPIDER. We fine-tuned learning rate for all algorithms, i.e., $\gamma = 1e^{-6}$ for SGD, $\gamma = 1e^{-3}$ for NSGD, NSGD with momentum, clipped SGD and IAN-SGD, $\gamma = 5e^{-3}$ for SPIDER. We keep other parameters the same as in Section A.1. The first row of Figure 10 plots the comparison results of Phase Retrieval (left) and DRO (right). We find in both problems, normalized methods such as NSGD, SPIDER, clipped SGD are slow to converge under small batch size.

To further justify the effectiveness of IAN-SGD under small batch size, we also test IAN-SGD and other baseline methods using small batch size to train ResNet (He et al., 2016) over CIFAR10 (Krizhevsky, 2009). For all algorithms, we unify the batch size $B = 32$. We fine-tuned learning for all algorithms, i.e., $\gamma = 1e^{-5}$ for SGD, $\gamma = 1e^{-4}$ for Adam, $\gamma = 1e^{-3}$ for Adagrad, $\gamma = 5e^{-2}$ for NSGD and NSGD with momentum, $\gamma = 8e - 2$ for clipped SGD and IAN-SGD, the second row of Figure 10 plots the corresponding results for training ResNet18 (left) and ResNet50 (right). As a result, we found IAN-SGD still outperforms than most baselines. This verifies the effectiveness of using IAN-SGD with small batch size in practice.

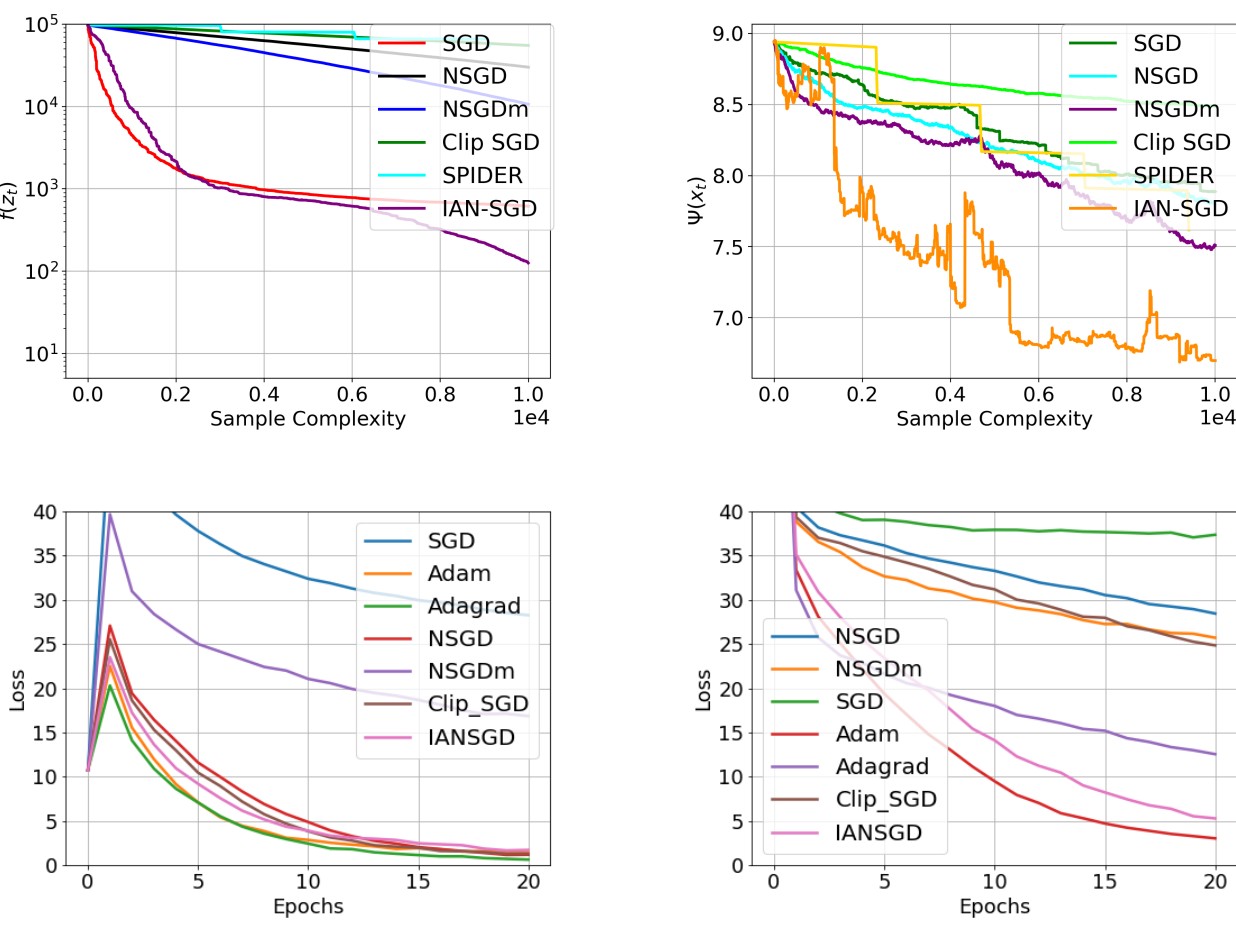

Figure 10: Comparison of IAN-SGD with other baselines under small batch size

# B   Proof of Descent Lemma 1

**Lemma 1** *Under Assumption 1, function $f$ satisfies, for any $w, w' \in \mathbf{R}^d$,*

$$f(w) \leq f(w') + \langle \nabla f(w'), w - w' \rangle + \frac{1}{2}(L_0 + L_1 \|\nabla f(w')\|^{\alpha})\|w - w'\|^2. \tag{3}$$

**Proof 1** *Denote $w_\theta = \theta w' + (1 - \theta)w$. By the fundamental theorem of calculus we have $f(w') - f(w) = \int_0^1 \langle \nabla f(w_\theta), w' - w \rangle \mathrm{d}\theta$, which implies that*

$$f(w') - f(w) - \langle \nabla f(w), w' - w \rangle = \int_0^1 \langle \nabla f(w_\theta), w' - w \rangle \mathrm{d}\theta - \int_0^1 \langle \nabla f(w), w' - w \rangle \mathrm{d}\theta,$$

*Substituting the generalized-smooth condition in Assumption 1 into the above equation, we have*

$$
\begin{aligned}
&f(w') - f(w) - \langle \nabla f(w), w' - w \rangle \\
&= \int_0^1 \langle \nabla f(w_\theta), w' - w \rangle \mathrm{d}\theta - \int_0^1 \langle \nabla f(w), w' - w \rangle \mathrm{d}\theta \\
&= \int_0^1 \langle \nabla f(w_\theta) - \nabla f(w), w' - w \rangle \mathrm{d}\theta \\
&\overset{(i)}{\leq} \int_0^1 \|\nabla f(w_\theta) - \nabla f(w)\| \|w' - w\| \mathrm{d}\theta \\
&\overset{(ii)}{\leq} \int_0^1 \theta(L_0 + L_1 \|\nabla f(w)\|^{\alpha})\|w' - w\|^2 \mathrm{d}\theta \\
&= \frac{1}{2}(L_0 + L_1 \|\nabla f(w)\|^{\alpha})\|w' - w\|^2,
\end{aligned}
\tag{23}
$$

*where (i) applies Cauchy-schwarz inequality on $\langle \nabla f(w_\theta) - \nabla f(w), w_\theta - w \rangle$, and (ii) is due to Assumption 1, $\|\nabla f(w_\theta) - \nabla f(w)\| \leq (L_0 + L_1 \|\nabla f(w)\|^{\alpha})\|w_\theta - w\| = \theta(L_0 + L_1 \|\nabla f(w)\|^{\alpha})\|w' - w\|$. Re-organizing the above inequality gives the desired result.*

# C   Proof for Nonconvex Phase Retrieval

The proof of generalized-smooth property for nonconvex Phase retrieval problem is similar as proof in Chen et al. (2023) with minor changes. We present the proof details here for completeness.

**Proof 2** *The objective function of phase retrieval problem can be expressed in the following form*

$$F(w) = \frac{2}{m} \sum_{\xi \in D} f_\xi(w) = \frac{1}{2m} \sum_{\xi \in D} \left(y_\xi - \left|a_\xi^\top w\right|^2\right)^2, \tag{24}$$

*where $f_\xi(w) = \frac{1}{4}(y_\xi - |a_\xi^T w|^2)^2$.*

*We prove that for every sample $\xi$, $\nabla f_\xi(w) = -(y_\xi - |a_\xi^\top w|^2)(a_\xi a_\xi^\top)w$ satisfies equation 4 with $\alpha = \frac{2}{3}$. And thus $F(w)$ satisfies finite-sum versions of equation 4, namely*

$$\frac{2}{m} \sum_{\xi \in D} \|f_\xi(w) - f_\xi(w')\| \leq \|w - w'\| \cdot \left(L_0 + \frac{L_1}{m} \sum_{\zeta \in D} \frac{\|\nabla f_\xi(w)\|^\alpha + \|\nabla f_\xi(w')\|^\alpha}{2}\right) \tag{25}$$

We construct a lower bound of $\|\nabla f_\xi(w)\|$ given $w$ and $\xi$ as following

$$
\begin{aligned}
\left\|\nabla f_\xi(w)\right\|^{\frac{2}{3}} &= \left\|\left(|a_\xi^\top w|^2 - y_\xi\right)\left(a_\xi a_\xi^\top\right)w\right\|^{\frac{2}{3}} \\
&= \left\|\left(|a_\xi^\top w|^2 \cdot a_\xi^\top w - y_\xi a_\xi^\top w\right)a_\xi\right\|^{\frac{2}{3}} \\
&\overset{(i)}{=} \left\||a_\xi^\top w|^3 - y_\xi|a_\xi^\top w|\right\|^{\frac{2}{3}}\|a_\xi\|^{\frac{2}{3}} \\
&\overset{(ii)}{\geq} \left(|a_\xi^\top w|^2 - \left|y_\xi|a_\xi^\top w|\right|^{\frac{2}{3}}\right)\|a_\xi\|^{\frac{2}{3}} \\
&\overset{(iii)}{\geq} \frac{2}{3}\left(|a_\xi^\top w|^2 - |y_\xi|\right)\|a_\xi\|^{\frac{2}{3}},
\end{aligned}
\tag{26}
$$

where (i) extracts scalar $\left||a_\xi^\top w|^3 - y_\xi|a_\xi^\top w|\right|^{\frac{2}{3}}$ out; (ii) uses inequality $|a-b|^{\frac{2}{3}} \geq |a|^{\frac{2}{3}} - |b|^{\frac{2}{3}}$ holding for any $a, b \in \mathbf{R}$ (sub-additive property of function $f(x) = x^{2/3}$); (iii) applies young's inequality $|y_\xi|^{\frac{2}{3}}|a_\xi^\top w|^{\frac{2}{3}} \leq \frac{1}{3}\|a_\xi^\top w\|^2 + \frac{2}{3}|y_\xi|$.

Then, for any $w, w' \in \mathbf{R}^d$, we have

$$
\begin{aligned}
&\left\|\nabla f_\xi(w') - \nabla f_\xi(w)\right\| \\
=&\left\|\left(|a_\xi^\top w'|^2 - y_\xi\right)\left(a_\xi a_\xi^\top\right)w' - \left(|a_\xi^\top w|^2 - y_\xi\right)\left(a_\xi a_\xi^\top\right)w\right\| \\
=&\frac{1}{2}\left\|2\left(|a_\xi^\top w'|^2 - y_\xi\right)\left(a_\xi a_\xi^\top\right)w' - 2\left(|a_\xi^\top w|^2 - y_\xi\right)\left(a_\xi a_\xi^\top\right)w + \left(|a_\xi^\top w'|^2 - y_\xi\right)\left(a_\xi a_\xi^\top\right)w - \left(|a_\xi^\top w|^2 - y_\xi\right)\left(a_\xi a_\xi^\top\right)w'\right. \\
&\left. - \left(|a_\xi^\top w'|^2 - y_\xi\right)\left(a_\xi a_\xi^\top\right)w + \left(|a_\xi^\top w|^2 - y_\xi\right)\left(a_\xi a_\xi^\top\right)w'\right\| \\
=&\frac{1}{2}\left\|\left(|a_\xi^\top w'|^2 - y_\xi\right)\left(a_\xi a_\xi^\top\right)w' - \left(|a_\xi^\top w'|^2 - y_\xi\right)\left(a_\xi a_\xi^\top\right)w + \left(|a_\xi^\top w|^2 - y_\xi\right)\left(a_\xi a_\xi^\top\right)w' - \left(|a_\xi^\top w|^2 - y_\xi\right)\left(a_\xi a_\xi^\top\right)w\right. \\
&\left. + \left(|a_\xi^\top w'|^2 - y_\xi\right)\left(a_\xi a_\xi^\top\right)w' + \left(|a_\xi^\top w'|^2 - y_\xi\right)\left(a_\xi a_\xi^\top\right)w - \left(|a_\xi^\top w|^2 - y_\xi\right)\left(a_\xi a_\xi^\top\right)w' - \left(|a_\xi^\top w|^2 - y_\xi\right)\left(a_\xi a_\xi^\top\right)w\right\| \\
=&\frac{1}{2}\left\|\left(|a_\xi^\top w'|^2 + |a_\xi^\top w|^2 - 2y_\xi\right)\left(a_\xi a_\xi^\top\right)(w'-w) + \left(|a_\xi^\top w'|^2 - |a_\xi^\top w|^2\right)\left(a_\xi a_\xi^\top\right)(w'+w)\right\| \\
\overset{(i)}{\leq}&\frac{1}{2}\left\|\left(|a_\xi^\top w'|^2 + |a_\xi^\top w|^2 - 2y_\xi\right)\left(a_\xi a_\xi^\top\right)(w'-w)\right\| + \frac{1}{2}\left\|\left(|a_\xi^\top w'|^2 - |a_\xi^\top w|^2\right)\left(a_\xi a_\xi^\top\right)(w'+w)\right\| \\
=&\frac{1}{2}\left\|\left(|a_\xi^\top w'|^2 + |a_\xi^\top w|^2 - 2y_\xi\right)\left(a_\xi a_\xi^\top\right)(w'-w)\right\| + \frac{1}{2}\left\|\left(|a_\xi^\top w'|^2 - |a_\xi^\top w|^2\right)a_\xi\left(a_\xi^\top w' + a_\xi^\top w\right)\right\| \\
\overset{(ii)}{\leq}&\frac{1}{2}\left\|\left(|a_\xi^\top w'|^2 + |a_\xi^\top w|^2 - 2y_\xi\right)\left(a_\xi a_\xi^\top\right)(w'-w)\right\| + \frac{1}{2}\left\|\left(|a_\xi^\top w'|^2 - |a_\xi^\top w|^2\right)a_\xi\left(|a_\xi^\top w'| + |a_\xi^\top w|\right)\right\| \\
\overset{(iii)}{\leq}&\frac{1}{2}\|a_\xi\|^2\left(|a_\xi^\top w'|^2 + |a_\xi^\top w|^2 + 2|y_\xi|\right)\|w'-w\| + \frac{1}{2}\|a_\xi\|^2\left(|a_\xi^\top w'| + |a_\xi^\top w|\right)^2\|w'-w\| \\
\overset{(iv)}{\leq}&\frac{1}{2}\|w'-w\|\|a_\xi\|^2\left(3|a_\xi^\top w'|^2 + 3|a_\xi^\top w|^2 + 2|y_\xi|\right) \\
\leq&\frac{1}{2}\|w'-w\|\|a_\xi\|^{\frac{4}{3}}\|a_\xi\|^{\frac{2}{3}} \cdot \left(3|a_\xi^\top w'|^2 + 3|a_\xi^\top w|^2 - 3|y_\xi| - 3|y_\xi| + 8|y_\xi|\right) \\
\overset{(v)}{\leq}&\|w'-w\|\left(\frac{9}{4}a_{\max}^{\frac{4}{3}}\left\|\nabla f_\xi(w')\right\|^{\frac{2}{3}} + \frac{9}{4}a_{\max}^{\frac{4}{3}}\left\|\nabla f_\xi(w)\right\|^{\frac{2}{3}} + 4y_{\max}a_{\max}^2\right),
\end{aligned}
\tag{27}
$$

where (i) uses triangular inequality; (ii) uses the fact $a_\xi^T w \leq |a_\xi^T w|$; (iii) uses the fact $\|a_\xi a_\xi^\top\| = \|a_\xi\|^2$ and equation 28; (iv) uses $(|a_\xi^\top w'| + |a_\xi^\top w|)^2 \leq 2|a_\xi^\top w'|^2 + 2|a_\xi^\top w|^2$; (v) uses equation 26 and denotes $y_{\max} = \max\|y_\xi\|$ and $a_{\max} = \max\|a_\xi\|$.

The inequality used in (iii) is constructed as following

$$
\begin{aligned}
\left||a_\xi^\top w'|^2 - |a_\xi^\top z|^2\right| &= \left(|a_\xi^\top w'| + |a_\xi^\top w|\right)\left(|a_\xi^\top w'| - |a_\xi^\top w|\right) \\
&\leq \left(|a_\xi^\top w'| + |a_\xi^\top w|\right)\|a_\xi^\top(w'-w)\| \\
&\leq \|a_\xi^\top\|\left(|a_\xi^\top w'| + |a_\xi^\top w|\right)\|w'-w\|.
\end{aligned}
\tag{28}
$$

*Thus, $f_\xi(w) = \frac{1}{4}(y_\xi - |a_\xi^T w|^2)^2$ satisfies equation 4 with $\alpha = \frac{2}{3}$, $L_0 = 4y_{\max}a_{\max}^2$, and $L_1 = \frac{9}{2}a_{\max}^{\frac{4}{3}}$. Thus, $F(w)$ satisfies equation 25 with $\alpha = \frac{2}{3}$, $L_0 = 8y_{\max}a_{\max}^2$, and $L_1 = 9a_{\max}^{\frac{4}{3}}$.*

## D Proof of Descent Lemma under Generalized PŁ condition

**Lemma 5 in** Chen et al. (2023). For any $x \geq 0$, $C \in [0,1]$, $\Delta > 0$, and $0 \leq w \leq w'$ such that $\Delta \geq w' - w$, we have the following inequality hold

$$Cx^w \leq x^{w'} + C^{\frac{w'}{\Delta}}. \tag{29}$$

The proof details for this lemma can be found at Chen et al. (2023), Lemma 5 at Appendix.

**Lemma 4 (Descent Lemma under Generalized PL condition)** *Let Assumption 1 and 2 hold. Apply AN-GD with $\beta \in [\alpha, 1]$. Choose target error $0 < \epsilon \leq \min\{1, 1/2\mu\}$. Let step size $\gamma = \frac{(2\mu\epsilon)^{\beta/\rho}}{8(L_0+L_1)+1}$. Denote $\Delta_t = f(w_t) - f^*$, then we have*

$$\Delta_{t+1} \leq \Delta_t - \frac{\gamma(2\mu)^{\frac{2-\beta}{\rho}}}{2}\Delta_t^{\frac{2-\beta}{\rho}} + \frac{\gamma}{4}(2\mu\epsilon)^{\frac{2-\beta}{\rho}}, \quad \forall t. \tag{30}$$

**Proof 3** *We consider two complementary cases $\alpha > 0$ and $\alpha = 0$.*

*When $\alpha > 0$, we have $\beta \geq \alpha > 0$, and by the descent lemma 1, we have that*

$$
\begin{aligned}
&f(w_{t+1}) - f(w_t)\\
&\overset{(i)}{\leq} \nabla f(w_t)^\top (w_{t+1} - w_t) + \frac{1}{2}\big(L_0 + L_1\|\nabla f(w_t)\|^\alpha\big)\|w_{t+1} - w_t\|^2\\
&\overset{(ii)}{=} -\gamma\|\nabla f(w_t)\|^{2-\beta} + \frac{\gamma}{4}\big(2L_0\gamma \cdot \|\nabla f(w_t)\|^{2-2\beta} + 2L_1\gamma \cdot \|\nabla f(w_t)\|^{2+\alpha-2\beta}\big)\\
&\overset{(iii)}{\leq} -\gamma\|\nabla f(w_t)\|^{2-\beta} + \frac{\gamma}{4}\big(2\|\nabla f(w_t)\|^{2-\beta} + (2L_0\gamma)^{\frac{2}{\beta}-1} + (2L_1\gamma)^{\frac{2}{\beta}-1}\big)\\
&\overset{(iv)}{\leq} -\frac{\gamma}{2}\|\nabla f(w_t)\|^{2-\beta} + \frac{\gamma^{\frac{2}{\beta}}}{4}\big(2L_0 + 2L_1\big)^{\frac{2}{\beta}-1}\\
&\overset{(v)}{\leq} -\frac{\gamma}{2}\|\nabla f(w_t)\|^{2-\beta} + \frac{(2\mu\epsilon)^{\frac{2}{\rho}}}{(8(L_0+L_1)+1)^{\frac{2}{\beta}}}(\frac{1}{4})(8(L_0+L_1)+1)^{\frac{2}{\beta}-1}\\
&\overset{(vi)}{\leq} -\frac{\gamma}{2}\|\nabla f(w_t)\|^{2-\beta} + \frac{1}{4}\frac{(2\mu\epsilon)^{\frac{\beta}{\rho}}(2\mu\epsilon)^{\frac{2-\beta}{\rho}}}{8(L_0+L_1)+1}\\
&= -\frac{\gamma}{2}\|\nabla f(w_t)\|^{2-\beta} + \frac{\gamma}{4}(2\mu\epsilon)^{\frac{2-\beta}{\rho}},
\end{aligned}
\tag{31}
$$

*where (i) follows from the descent lemma 1; (ii) follows from the AN-GD update rule by replacing $w_{t+1} - w_t$ with $-\gamma\frac{\nabla f(w_t)}{\|\nabla f(w_t)\|^\beta}$; (iii) follows from the technical lemma 5 above proved in Chen et al. (2023) by letting $\omega' = 2 - \beta$, $\Delta = \beta$ and substituting equation 29 to $2L_0\gamma\|\nabla f(w_t)\|^{2-2\beta}$ and $2L_1\gamma\|\nabla f(w_t)\|^{2+\alpha-2\beta}$ (Let $x = \|\nabla f(w_t)\|$ and $C = 2L_0\gamma$ or $2L_1\gamma$, $C \in [0,1]$ because $\epsilon \leq 1/2\mu$); (iv) follows from the fact that $a^\tau + b^\tau \leq (a+b)^\tau$ with $\tau = 2/\beta - 1 \geq 1$ and $a, b \geq 0$, (v) follows from the step size rule $\gamma = (2\mu\epsilon)^{\beta/\rho}/(8(L_0+L_1)+1)$; (vi) follows from the facts that $2L_0 + 2L_1 \leq 8(L_0+L_1)+1$ and $(2L_0+2L_1)^{\frac{2}{\beta}-1} \leq (8(L_0+L_1)+1)^{\frac{2}{\beta}-1}$ when $\beta > 0$.*

When $\alpha = 0$, the same argument applies if $\beta > 0$. Next, we consider the case $\alpha = \beta = 0$, and we have $\gamma = \frac{1}{8(L_0 + L_1) + 1}$. We obtain that

$$
\begin{aligned}
&f(w_{t+1}) - f(w_t) \\
&\overset{(i)}{\leq} \nabla f(w_t)^\top (w_{t+1} - w_t) + \frac{1}{2}(L_0 + L_1)\|w_{t+1} - w_t\|^2 \\
&\overset{(ii)}{=} -\gamma\|\nabla f(w_t)\|^2 + \frac{\gamma}{4}\left(2L_0\gamma \cdot \|\nabla f(w_t)\|^2 + 2L_1\gamma \cdot \|\nabla f(w_t)\|^2\right) \\
&= -\gamma\|\nabla f(w_t)\|^2 + \frac{\gamma^2}{4}(2L_0 + 2L_1)\|\nabla f(w_t)\|^2 \\
&\overset{(iii)}{\leq} -\gamma\|\nabla f(w_t)\|^2 + \frac{\gamma}{4}\frac{1}{8(L_0 + L_1) + 1}(8(L_0 + L_1) + 1)\|\nabla f(w_t)\|^2 \\
&\overset{(iv)}{\leq} -\gamma\|\nabla f(w_t)\|^2 + \frac{\gamma}{2}\|\nabla f(w_t)\|^2 \\
&\overset{(v)}{\leq} -\gamma\|\nabla f(w_t)\|^2 + \frac{\gamma}{2}\|\nabla f(w_t)\|^2 + \frac{\gamma}{4}(2\mu\epsilon)^{\frac{2}{\rho}} \\
&= -\frac{\gamma}{2}\|\nabla f(w_t)\|^2 + \frac{\gamma}{4}(2\mu\epsilon)^{\frac{2}{\rho}},
\end{aligned}
\tag{32}
$$

where (i) follows from the descent lemma 1 with $\alpha = 0$; (ii) follows from the update rule of AN-GD by replacing $w_{t+1} - w_t$ with $-\gamma\nabla f(w_t)$ (Normalization term vanishes due to $\beta = 0$); (iii) follows the fact that $2L_0 + 2L_1 \leq 8(L_0 + L_1) + 1$; (iv) follows from the fact $\frac{\gamma}{4} \leq \frac{\gamma}{2}$ and (v) holds by adding additional positive number $\frac{\gamma}{4}(2\mu\epsilon)^{\frac{2}{\rho}}$.

Thus, in conclusion, for any $\beta \in [\alpha, 1]$, we always have the following descent lemma

$$
f(w_{t+1}) - f(w_t) \leq -\frac{\gamma}{2}\|\nabla f(w_t)\|^{2-\beta} + \frac{\gamma}{4}(2\mu\epsilon)^{\frac{2-\beta}{\rho}}
\tag{33}
$$

under the step size rule $\gamma = \frac{(2\mu\epsilon)^{\beta/\rho}}{8(L_0 + L_1) + 1}$. Moreover, by the generalized PŁ-condition in Definition 5, we have

$$
\|\nabla f(w)\| \geq (2\mu)^{\frac{1}{\rho}}\left(f(w) - f^*\right)^{\frac{1}{\rho}},
$$

which implies that

$$
\|\nabla f(w_t)\|^{2-\beta} \geq (2\mu)^{\frac{2-\beta}{\rho}}\left(f(w_t) - f^*\right)^{\frac{2-\beta}{\rho}} = (2\mu)^{\frac{2-\beta}{\rho}}\Delta_t^{\frac{2-\beta}{\rho}}.
\tag{34}
$$

Substituting equation 34 into equation 33, we have

$$
f(w_{t+1}) - f(w_t) \leq -\frac{\gamma}{2}(2\mu)^{\frac{2-\beta}{\rho}}\left(f(w_t) - f^*\right)^{\frac{2-\beta}{\rho}} + \frac{\gamma}{4}(2\mu\epsilon)^{\frac{2-\beta}{\rho}}.
$$

Subtracting $f^*$ on both sides, we have

$$
f(w_{t+1}) - f^* \leq f(w_t) - f^* - \frac{\gamma}{2}(2\mu)^{\frac{2-\beta}{\rho}}(f(w) - f^*)^{\frac{2-\beta}{\rho}} + \frac{\gamma}{4}(2\mu\epsilon)^{\frac{2-\beta}{\rho}}.
$$

Denote $\Delta_t = f(w_t) - f^*$, the above inequality can be written as

$$
\Delta_{t+1} \leq \Delta_t - \frac{\gamma(2\mu)^{\frac{2-\beta}{\rho}}}{2}\Delta_t^{\frac{2-\beta}{\rho}} + \frac{\gamma}{4}(2\mu\epsilon)^{\frac{2-\beta}{\rho}}, \quad \forall t.
\tag{35}
$$

# E    Proof of Theorem 1

**Theorem 1 (Convergence of AN-GD)** *Let Assumptions 1 and 2 hold. Denote $\Delta_t := f(w_t) - f^*$ as the function value gap. Choose target error $0 < \epsilon \leq \min\{1, \frac{1}{2\mu}\}$ and define learning rate $\gamma = \frac{(2\mu\epsilon)^{\beta/\rho}}{8(L_0 + L_1) + 1} \leq \frac{2}{\mu}$ for some $\beta \in [\alpha, 1]$. Then, the following statements hold,*

- If $\beta < 2 - \rho$, in order to achieve $\Delta_T \leq \epsilon$, the total number of iterations must satisfy

$$T \geq \max \left\{ \frac{8\rho(8(L_0 + L_1) + 1)}{(2 - \beta - \rho)(2\mu)^{2/\rho}\epsilon^{(2-\rho)/\rho}}, \frac{1}{(2^{(2-\beta-\rho)/(2-\beta)} - 1)\Delta_0^{(\rho+\beta-2)/\rho}\epsilon^{(2-\beta-\rho)/\rho}} \right\} = \Omega\left((\frac{1}{\epsilon})^{\frac{2-\rho}{\rho}}\right). \quad (7)$$

- If $\beta = 2 - \rho$, in order to achieve $\Delta_T \leq \epsilon$, the total number of iterations must satisfy

$$T \geq \frac{2^{1-\beta/\rho} \cdot (8(L_0 + L_1) + 1)}{\mu \cdot (\mu\epsilon)^{\beta/\rho}} \cdot \log(\frac{\Delta_0}{\epsilon}) = \Omega\left((\frac{1}{\epsilon})^{\frac{\beta}{\rho}} \cdot \log(\frac{\Delta_0}{\epsilon})\right). \quad (8)$$

- If $1 \geq \beta > 2 - \rho$, and for some $T_0$ satisfying $\Delta_{T_0} \leq \mathcal{O}\left((\gamma\mu^{(2-\beta)/(\rho+\beta-2)})^{\frac{\rho}{\rho+\beta-2}}\right)$, then the total number of iterations after $T_0$ must satisfy

$$T \gtrsim \Omega\left( \log\left((\frac{1}{\epsilon})^{\frac{\beta}{\rho+\beta-2}}\right) + \log\left(\log\left(\frac{(2\mu)^{2/(\rho+\beta-2)}}{(32(L_0 + L_1) + 4)^{\rho/(\rho+\beta-2)}\epsilon}\right)\right)\right) = \Omega\left(\log\left((\frac{1}{\epsilon})^{\frac{\beta}{\rho+\beta-2}}\right)\right). \quad (9)$$

**Proof 4** *Given target error $0 < \epsilon \leq \min\{1, 1/2\mu\}$, define stopping time as the first time that $\Delta_t$ satisfies $\Delta_t \leq \epsilon$, i.e., $T = \inf\{t | \Delta_t \leq \epsilon\}$. In the following proof, we provide an estimate of such stopping time $T$.*

*For any $t$ satisfies $\Delta_t > \epsilon$, we must have $\Delta_t^{(2-\beta)/\rho} > \epsilon^{(2-\beta)/\rho}$ since $\frac{2-\beta}{\rho} > 0$. This implies that $\frac{\gamma}{4}(2\mu\epsilon)^{(2-\beta)/\rho} < \frac{\gamma}{4}(2\mu)^{(2-\beta)/\rho}\Delta_t^{(2-\beta)/\rho}$. Thus, equation 35 further reduces to the following descent inequality*

$$\Delta_{t+1} \leq \Delta_t - \frac{\gamma(2\mu)^{\frac{2-\beta}{\rho}}}{4}\Delta_t^{\frac{2-\beta}{\rho}}. \quad (36)$$

*For above equation 36, we provide convergence rate based on the value of $\rho$ and $\beta$.*
**Case I:** $\beta < 2 - \rho$
*This is equivalent to $\frac{2-\beta}{\rho} > 1$. For simplicity of notation, denote $\theta = \frac{2-\beta}{\rho}$. When $\theta > 1$, we have following inequalities*

$$\begin{aligned}
\Delta_{t+1} &\leq \Delta_t \\
\Delta_{t+1}^{\theta} &\leq \Delta_t^{\theta} \\
\Delta_{t+1}^{-\theta} &\geq \Delta_t^{-\theta}.
\end{aligned} \quad (37)$$

*Now define an auxiliary function $\Phi(t) = \frac{1}{\theta-1}t^{1-\theta}$, where its derivative is $\Phi'(t) = -t^{-\theta}$ We divide the last inequality of equation 37 into two different cases for analysis. One is the case when $\Delta_t^{-\theta} \leq \Delta_{t+1}^{-\theta} \leq 2\Delta_t^{-\theta}$, another is the case when $\Delta_{t+1}^{-\theta} \geq 2\Delta_t^{-\theta}$.*

*When $\Delta_t^{-\theta} \leq \Delta_{t+1}^{-\theta} \leq 2\Delta_t^{-\theta}$, we have*

$$\begin{aligned}
\Phi(\Delta_{t+1}) - \Phi(\Delta_t) &= \int_{\Delta_t}^{\Delta_{t+1}} \Phi'(t)\mathrm{d}t = \int_{\Delta_{t+1}}^{\Delta_t} t^{-\theta}\mathrm{d}t \\
&\overset{(i)}{\geq} (\Delta_t - \Delta_{t+1})\Delta_t^{-\theta} \\
&\overset{(ii)}{\geq} (\Delta_t - \Delta_{t+1})\frac{\Delta_{t+1}^{-\theta}}{2} \\
&\overset{(iii)}{\geq} \frac{\gamma(2\mu)^{\theta}}{4}\Delta_t^{\theta}\frac{\Delta_{t+1}^{-\theta}}{2} \\
&\overset{(iv)}{\geq} \frac{\gamma(2\mu)^{\theta}}{4}\Delta_{t+1}^{\theta}\frac{\Delta_{t+1}^{-\theta}}{2} = \frac{\gamma(2\mu)^{\theta}}{8},
\end{aligned}$$

*where $(i)$ applies mean value theorem such that $\Phi(\Delta_{t+1}) - \Phi(\Delta_t) = |\Delta_{t+1} - \Delta_t||\Phi'(\xi)|$ and $\Delta_{t+1}^{-\theta} \geq |\Phi'(\xi)| \geq \Delta_t^{-\theta}$ holds for any $\xi \in [\Delta_{t+1}, \Delta_t]$; $(ii)$ uses the fact $\Delta_{t+1}^{-\theta} \leq 2\Delta_t^{-\theta}$; $(iii)$ utilizes recursion $\Delta_t - \Delta_{t+1} \geq \frac{\gamma(2\mu)^{\theta}}{4}\Delta_t^{\theta}$; $(iv)$ uses the fact that $\Delta_t^{\theta} > \Delta_{t+1}^{\theta}$ for all $\theta > 0$.*

When $\Delta_{t+1}^{-\theta} > 2\Delta_t^{-\theta}$, it holds that $\Delta_{t+1}^{1-\theta} = (\Delta_{t+1}^{-\theta})^{\frac{1-\theta}{-\theta}} > 2^{\frac{1-\theta}{-\theta}}\Delta_t^{1-\theta}$. Then, we have

$$
\begin{aligned}
\Phi(\Delta_{t+1}) - \Phi(\Delta_t) &= \frac{1}{\theta-1}(\Delta_{t+1}^{1-\theta} - \Delta_t^{1-\theta}) \\
&\overset{(i)}{\geq} \frac{1}{\theta-1}\left((2)^{\frac{\theta-1}{\theta}} - 1\right)\Delta_t^{1-\theta} \\
&\overset{(ii)}{\geq} \frac{1}{\theta-1}\left((2)^{\frac{\theta-1}{\theta}} - 1\right)\Delta_0^{1-\theta},
\end{aligned}
$$

where (i) is due to the recursion $\Delta_{t+1}^{1-\theta} = (\Delta_{t+1}^{-\theta})^{\frac{1-\theta}{-\theta}} > 2^{\frac{1-\theta}{-\theta}}\Delta_t^{1-\theta}$; (ii) is due to the fact the sequence $\{\Delta_t\}_{t=1}^{T}$ generated by equation 36 is non-increasing. Substitute $\theta = \frac{2-\beta}{\rho}$ in and denote

$$
C = \min\left\{\frac{\gamma(2\mu)^{\frac{2-\beta}{\rho}}}{8}, \frac{\rho}{2-\beta-\rho}(2^{\frac{2-\beta-\rho}{2-\beta}} - 1)\Delta_0^{\frac{\beta+\rho-2}{\rho}}\right\}.
$$

We conclude for all $t$, we have

$$
\Phi(\Delta_t) \geq \Phi(\Delta_t) - \Phi(\Delta_0) = \sum_{i=0}^{t-1}\Phi(\Delta_{i+1}) - \Phi(\Delta_i) \geq Ct,
$$

Substituting the expression $\Phi(\Delta_t) = \frac{1}{\theta-1}\Delta_t^{1-\theta}$ into the above inequality, we have

$$
\Delta_t \leq \left(\frac{\rho}{(2-\beta-\rho)\cdot Ct}\right)^{\frac{\rho}{2-\rho-\beta}}, \tag{38}
$$

In order to obtain $\Delta_T \leq \epsilon$, when $C = \frac{\rho}{2-\beta-\rho}(2^{(2-\beta-\rho)/(2-\beta)} - 1)\Delta_0^{(\beta+\rho-2)/\rho}$, $T$ should satisfy

$$
T = \frac{1}{(2^{(2-\beta-\rho)/(2-\beta)} - 1)\Delta_0^{(\rho+\beta-2)/\rho}\epsilon^{(2-\beta-\rho)/\rho}}; \tag{39}
$$

When $C = \frac{\gamma(2\mu)^{2-\beta/\rho}}{8} = \frac{(2\mu)^{2/\rho}\cdot\epsilon^{\beta/\rho}}{8(8(L_0+L_1)+1)}$, $T$ should satisfy

$$
T = \frac{8\rho(8(L_0+L_1)+1)}{(2-\beta-\rho)(2\mu)^{2/\rho}\epsilon^{(2-\rho)/\rho}}. \tag{40}
$$

This concludes when $\rho + \beta < 2$, $T$ should satisfy

$$
T \geq \max\left\{\frac{8\rho(8(L_0+L_1)+1)}{(2-\beta-\rho)(2\mu)^{2/\rho}\epsilon^{(2-\rho)/\rho}}, \frac{1}{(2^{(2-\beta-\rho)/(2-\beta)} - 1)\Delta_0^{(\rho+\beta-2)/\rho}\epsilon^{(2-\beta-\rho)/\rho}}\right\} = \Omega\left(\left(\frac{1}{\epsilon}\right)^{\frac{2-\rho}{\rho}}\right). \tag{41}
$$

**Case II:** $\beta = 2 - \rho$

This is equivalent to $\frac{2-\beta}{\rho} = 1$, equation 36 reduces to

$$
\Delta_{t+1} \leq \Delta_t - \frac{\gamma\mu}{2}\Delta_t = (1 - \frac{\gamma\mu}{2})\Delta_t,
$$

which leads to

$$
\Delta_t \leq (1 - \frac{\gamma\mu}{2})^t\Delta_0 = \mathcal{O}\left((1 - \frac{\gamma\mu}{2})^t\right).
$$

To obtain $\Delta_T \leq \epsilon$, we have

$$
\Delta_t \leq (1 - \frac{\gamma\mu}{2})^t\Delta_0 \leq \exp(-\frac{\gamma\mu t}{2})\Delta_0. \tag{42}
$$

*To obtain $\Delta_t \leq \epsilon$, iteration complexity should satisfy*

$$T \geq \frac{2}{\gamma\mu}\log(\frac{\Delta_0}{\epsilon}) = \frac{2^{1-\beta/\rho}\cdot(8(L_0+L_1)+1)}{\mu\cdot(\mu\epsilon)^{\beta/\rho}}\cdot\log(\frac{\Delta_0}{\epsilon}) = \Omega((\frac{1}{\epsilon})^{\frac{\beta}{\rho}}\cdot\log(\frac{\Delta_0}{\epsilon})).$$

**Case III:** $1 \geq \beta > 2 - \rho$

*This is equivalent to $\frac{2-\beta}{\rho} < 1$.*

*For simplicity of notation, denote $\omega = \frac{\rho}{2-\beta}$ and $C = \frac{(2\mu)^{\frac{2-\beta}{\rho}}}{4}$. Then, equation 36 can be rewritten as*

$$\Delta_{t+1} \leq \Delta_t - C\gamma\Delta_t^{\frac{1}{\omega}}. \tag{43}$$

*When $\Delta_t$ is small enough, $\Delta_{t+1}^{1/\omega}$ will dominate $\Delta_{t+1}$ order-wisely since $1/\omega < 1$. This fact leads to*

$$C\gamma\Delta_{t+1}^{\frac{1}{\omega}} \overset{(i)}{\leq} \Delta_{t+1} + C\gamma\Delta_{t+1}^{\frac{1}{\omega}} \overset{(ii)}{\leq} \Delta_{t+1} + C\gamma\Delta_t^{\frac{1}{\omega}} \overset{(iii)}{\leq} \Delta_t,$$

*where (i) is because $\Delta_{t+1} \geq 0$; (ii) is because $\Delta_{t+1} \leq \Delta_t$; (iii) is because re-organization of equation 36. There exists a time $T_0 = \inf\{t \in \mathbf{N}|\Delta_t/(C\gamma)^{\omega/(\omega-1)} < 1\}$ such that for any $T_0 \leq t \leq T$, we have*

$$\begin{aligned}\Delta_{t+1} \leq (C\gamma)^{-\omega}\Delta_t^{\omega} &= (C\gamma)^{-\omega-\omega^2-\ldots-\omega^{t-T_0}}\Delta_{T_0}^{\omega^{t-T_0}}\\ &= (C\gamma)^{\frac{\omega(1-\omega^{t-T_0})}{\omega-1}}\Delta_{T_0}^{\omega^{t-T_0}}\\ &= (C\gamma)^{\omega/\omega-1}\big((C\gamma)^{\omega/\omega-1}\big)^{\omega^{-t-T_0}}\Delta_{T_0}^{\omega^{t-T_0}}.\end{aligned} \tag{44}$$

*To simplify analysis, denote all parameters associated with $\epsilon$ as $\hat{C} = (\frac{C(2\mu)^{\beta/\rho}}{8(L_0+L_1)+1})^{\omega/\omega-1}$. We have $(C\gamma)^{\omega/\omega-1} = \hat{C}\epsilon^{\beta/(\rho+\beta-2)} \leq \hat{C}$. This enables us to further reduce the recursion to*

$$\begin{aligned}\Delta_{t+1} &\leq (C\gamma)^{\omega/\omega-1}\big((C\gamma)^{\omega/\omega-1}\big)^{\omega^{-t-T_0}}\Delta_{T_0}^{\omega^{t-T_0}}\\ &\leq \hat{C}\big((C\gamma)^{\frac{\omega}{\omega-1}}\big)^{-\omega^{t-T_0}}\Delta_{T_0}^{\omega^{t-T_0}}\\ &= \mathcal{O}\bigg(\Big(\frac{\Delta_{T_0}}{\gamma^{\omega/\omega-1}}\Big)^{\omega^{t-T_0}}\bigg).\end{aligned} \tag{45}$$

*To obtain $\Delta_T \leq \epsilon$, above recursion should satisfy*

$$\hat{C}\big((C\gamma)^{\frac{\omega}{\omega-1}}\big)^{-\omega^{t-T_0}}\Delta_{T_0}^{\omega^{t-T_0}} = \epsilon.$$

*Taking logarithm on both sides and extracting $\epsilon^{\beta/(\rho+\beta-2)}$ from $(C\gamma)^{\omega/\omega-1}/\Delta_{T_0}$, we have*

$$\log(\hat{C}/\epsilon) = \omega^{t-T_0}\cdot\log\Big((C\gamma)^{\frac{\omega}{\omega-1}}/\Delta_{T_0}\Big) = \omega^{t-T_0}\cdot\log\Big((\hat{C}/\Delta_{T_0})\cdot\epsilon^{\frac{\beta}{\rho+\beta-2}}\Big) \leq \omega^{t-T_0}\big(\hat{C}/\Delta_{T_0}\big)\cdot\epsilon^{\frac{\beta}{\rho+\beta-2}}.$$

*Taking logarithm again on both sides of the above inequality, we have*

$$\begin{aligned}t - T_0 \geq &\frac{1}{\log\left(\frac{\rho}{2-\beta}\right)}\cdot\bigg[\log\Big(\frac{\beta}{\rho+\beta-2}\cdot\log\Big(\frac{1}{\epsilon}\Big)\Big) + \log\Big(\log\Big(\frac{(2\mu)^{2/(\rho+\beta-2)}}{(32(L_0+L_1)+4)^{\rho/(\rho+\beta-2)}\epsilon}\Big)\Big)\\ &\qquad\qquad + \log\Big(\frac{(32(L_0+L_1)+4)^{\rho/(\rho+\beta-2)}\Delta_{T_0}}{(2\mu)^{2/(\rho+\beta-2)}}\Big)\bigg]\\ \gtrsim &\Omega\Big(\log\Big(\frac{\beta}{\rho+\beta-2}\cdot\log\Big(\frac{1}{\epsilon}\Big) + \log\Big(\log\Big(\frac{(2\mu)^{2/(\rho+\beta-2)}}{(32(L_0+L_1)+4)^{\rho/(\rho+\beta-2)}\epsilon}\Big)\Big)\Big)\Big)\\ = &\Omega\Big(\log\Big((\frac{1}{\epsilon})^{\frac{\beta}{\rho+\beta-2}}\Big)\Big).\end{aligned}$$

# F   Proof of Proposition 1

**Proposition 1 (Convergence of GD)** *Let Assumptions 1 and 2 hold. Assume there exists a positive constant $G$ such that $\|\nabla f(x_t)\| \leq G, \forall t \in T$, When $\rho = \alpha = 1$ and setting $\gamma \leq \min\{\frac{1}{L_0}, \frac{1}{2L_1 G}\}$, gradient descent converges to an $\epsilon$-stationary point within $T \geq \Omega(\frac{G}{\epsilon})$ iterations.*

**Proof 5** *When $\alpha = 1$, putting the update rule of gradient descent $w_{t+1} = w_t - \gamma \nabla f(w_t)$ into descent lemma equation 1 yields*

$$
\begin{aligned}
&f(w_{t+1}) \\
&\overset{(i)}{\leq} f(w_t) - \gamma \|\nabla f(w_t)\|^2 + \frac{\gamma^2}{2}(L_0 + L_1\|\nabla f(w_t)\|)\|\nabla f(w_t)\|^2 \\
&= f(w_t) - (\gamma - \frac{L_0\gamma^2}{2})\|\nabla f(w_t)\|^2 + \frac{L_1\gamma^2}{2}\|\nabla f(w_t)\|^3 \\
&\overset{(ii)}{\leq} f(w_t) - \frac{\gamma}{2}\|\nabla f(w_t)\|^2 + \frac{L_1\gamma^2}{2}\|\nabla f(w_t)\|^3 \\
&\overset{(iii)}{\leq} f(w_t) - \frac{\gamma}{4}\|\nabla f(w_t)\|^2
\end{aligned}
\tag{46}
$$

*where (i) is due to descent lemma equation 1; (ii) and (iii) are due to the learning rate design $\gamma \leq \min\{\frac{1}{L_0}, \frac{1}{2L_1 G}\}$, which ensures $-(\gamma - \frac{L_0\gamma^2}{2}) \leq -\frac{\gamma}{2}$ and $\frac{L_1\gamma^2}{2}\|\nabla f(w_t)\|^3 \leq \frac{\gamma}{4}\|\nabla f(w_t)\|^2$.*

*When applying Assumption 2 with $\rho = 1$ and denote $f(w_t) - f^*$ as $\Delta_t$, we have*

$$
\Delta_{t+1} \leq \Delta_t - \gamma\mu^2\Delta_t^2
\tag{47}
$$

*The rest of proof is exactly the same as proof for Theorem 1 Case I, we omit discussion here. As a result, one can show equation 47 converges to a $\epsilon$-stationary point after $\mathcal{O}(\frac{G}{\epsilon})$ iterations.*

# G  Proof of Lemma 2

**Lemma 2** *Let Assumptions 3 and 4 hold. The gradient $\|\nabla F(w_t)\|$ and stochastic gradient $\nabla f_\xi(w_t)$ satisfy*

$$\|\nabla F(w_t)\| \leq \frac{1}{1-\tau_1}\|\nabla f_\xi(w_t)\| + \frac{\tau_2}{1-\tau_1}, \, and$$
$$\|\nabla f_\xi(w_t)\| \leq (1+\tau_1)\|\nabla F(w_t)\| + \tau_2. \tag{16}$$

**Proof 6** *Based on Assumption 4, we have*

$$\|\nabla F(w_t) - \nabla f_{\xi'}(w_t)\| \leq \tau_1\|\nabla F(w_t)\| + \tau_2,$$

*which, by triangle inequality, further implies that*

$$\left|\|\nabla F(w_t)\| - \|\nabla f_{\xi'}(w_t)\|\right| \leq \|\nabla F(w_t) - \nabla f_{\xi'}(w_t)\| \leq \tau_1\|\nabla F(w_t)\| + \tau_2.$$

*Rearranging the above inequality yields that*

$$\|\nabla F(w_t)\| \leq \frac{1}{1-\tau_1}\|\nabla f_{\xi'}(w_t)\| + \frac{\tau_2}{1-\tau_1}.$$

*For the upper bound of $\|\nabla f_{\xi'}(w_t)\|$, it directly follows from Assumption 4.*

# H  Lemma 5 and Proof

**Lemma 5** *For $\frac{1}{2}\gamma^2 \frac{(L_0 + L_1\|\nabla F(w_t)\|^\alpha)}{h_t^{2\beta}} \cdot 4\tau_2^2$, we have the following upper bound*

$$\frac{1}{2}\gamma^2 \frac{(L_0 + L_1\|\nabla F(w_t)\|^\alpha)}{h_t^{2\beta}} \cdot 4\tau_2^2 \leq \frac{1}{2}\gamma^2(L_0 + L_1)(1 + 4\tau_2^2)^2 + \frac{\gamma}{4h_t^\beta}\|\nabla F(w_t)\|^2. \tag{48}$$

**Proof 7** *When $\|\nabla F(w_t)\| \leq \sqrt{1 + 4\tau_2^2/(2\tau_1^2 + 1)}$, we have $\|\nabla F(w_t)\|^\alpha \leq (1 + 4\tau_2^2/(2\tau_1^2 + 1))^{\frac{\alpha}{2}}$ for any $\alpha > 0$. Then, we have*

$$\frac{\gamma^2}{2} \frac{(L_0 + L_1\|\nabla F(w_t)\|^\alpha)}{h_t^{2\beta}} 4\tau_2^2$$
$$\leq \frac{1}{2}\gamma^2(1 + \frac{4\tau_2^2}{2\tau_1^2 + 1})^{\frac{\alpha}{2}}\frac{(L_0 + L_1)}{h_t^{2\beta}}4\tau_2^2$$
$$\leq \frac{1}{2}\gamma^2(L_0 + L_1)(1 + 4\tau_2^2)^{\frac{\alpha}{2}}(1 + 4\tau_2^2)$$
$$\leq \frac{1}{2}\gamma^2(L_0 + L_1)(1 + 4\tau_2^2)^2, \tag{49}$$

*where the first inequality uses $\|\nabla F(w_t)\|^\alpha \leq (1 + 4\tau_2^2/(2\tau_1^2 + 1))^{\alpha/2}$ and $(1 + 4\tau_2^2/(2\tau_1^2 + 1))^{\alpha/2} \geq 1$; the second inequality uses the fact that $\frac{1}{h_t} \leq 1$ (so does $\frac{1}{h_t^{2\beta}}$) and upper bound $4\tau_2^2$ by $1 + 4\tau_2^2$; the last inequality uses the fact that thus $(1 + 4\tau_2^2)^{1+\alpha/2} \leq (1 + 4\tau_2^2)^2$.*

*When $\left\|\nabla F(w_t)\right\| > \sqrt{1 + 4\tau_2^2/(2\tau_1^2 + 1)}$, we have*

$$\frac{\gamma^2}{2}\frac{L_1\left\|\nabla F(w_t)\right\|^{\alpha}}{h_t^{2\beta}} \cdot 4\tau_2^2$$

$$=\frac{\gamma^2}{2h_t^{\beta}}\frac{L_1\left\|\nabla F(w_t)\right\|^{\alpha}}{h_t^{\beta}} \cdot 4\tau_2^2$$

$$\overset{(i)}{\leq}\frac{\gamma^2}{2h_t^{\beta}}\frac{L_1\left\|\nabla F(w_t)\right\|^{\alpha}}{4L_1\gamma(2\tau_1^2+1)(\frac{1}{1-\tau_1}\|\nabla f_{\xi'}(w_t)\| + \frac{\tau_2}{1-\tau_1})^{\beta}} \cdot 4\tau_2^2$$

$$=\frac{\gamma^2}{2h_t^{\beta}}\frac{L_1\left\|\nabla F(w_t)\right\|^{\alpha}}{4L_1\gamma(\frac{1}{1-\tau_1}\|\nabla f_{\xi'}(w_t)\| + \frac{\tau_2}{1-\tau_1})^{\beta}} \cdot \frac{4\tau_2^2}{(2\tau_1^2+1)}$$

$$\overset{(ii)}{<}\frac{\gamma^2}{2h_t^{\beta}}\frac{L_1\left\|\nabla F(w_t)\right\|^{\alpha}}{(4L_1\gamma)(\frac{1}{1-\tau_1}\|\nabla f_{\xi'}(w_t)\| + \frac{\tau_2}{1-\tau_1})^{\beta}} \cdot \left\|\nabla F(w_t)\right\|^2$$

$$\overset{(iii)}{\leq}\frac{\gamma^2}{2h_t^{\beta}}\frac{L_1\left\|\nabla F(w_t)\right\|^{\alpha}}{(4L_1\gamma)\left\|\nabla F(w_t)\right\|^{\beta}}\left\|\nabla F(w_t)\right\|^2$$

$$=\frac{\gamma^2}{2h_t^{\beta}}\frac{L_1}{4L_1\gamma\left\|\nabla F(w_t)\right\|^{\beta-\alpha}}\left\|\nabla F(w_t)\right\|^2$$

$$\overset{(iv)}{<}\frac{\gamma^2}{2h_t^{\beta}}\frac{L_1}{4L_1\gamma}\left\|\nabla F(w_t)\right\|^2$$

$$=\frac{\gamma}{8h_t^{\beta}}\left\|\nabla F(w_t)\right\|^2, \tag{50}$$

*where (i) uses the fact $\frac{1}{h_t^{\beta}} \leq \frac{1}{4L_1\gamma(2\tau_1^2+1)(\frac{1}{1-\tau_1}\|\nabla f_{\xi'}(w_t)\|+\frac{\tau_2}{1-\tau_1})^{\beta}}$; (ii) uses the fact that now $\|\nabla F(w_t)\|^2 > 1 + 4\tau_2^2/(2\tau_1^2 + 1) \geq 4\tau_2^2/(2\tau_1^2 + 1)$; (iii) uses the fact $\|\nabla F(w_t)\| \leq \frac{1}{1-\tau_1}\|\nabla f_{\xi'}(w_t)\| + \frac{\tau_2}{1-\tau_1}$; (iv) uses the fact $\|\nabla F(w_t)\| > \sqrt{1 + 4\tau_2^2/(2\tau_1^2 + 1)} \geq 1$ and thus $\frac{1}{\|\nabla F(w_t)\|^{\beta-\alpha}} < 1$.*

*Similarly, when $\left\|\nabla F(w_t)\right\| > \sqrt{1 + 4\tau_2^2/(2\tau_1^2 + 1)}$, for $\frac{1}{2h_t^{2\beta}}\gamma^2 L_0 \cdot 4\tau_2^2$, we have*

$$\frac{1}{2h_t^{2\beta}}\gamma^2 L_0 \cdot 4\tau_2^2$$

$$=\frac{\gamma}{2h_t^{2\beta}}(\gamma L_0) \cdot 4\tau_2^2$$

$$\leq\frac{\gamma}{2h_t^{\beta}}(\gamma L_0) \cdot 4\tau_2^2$$

$$\leq\frac{\gamma}{2h_t^{\beta}}\frac{1}{4 \cdot (2\tau_1^2 + 1)}4\tau_2^2$$

$$=\frac{\gamma}{8h_t^{\beta}}\frac{4\tau_2^2}{2\tau_1^2 + 1}$$

$$\leq\frac{\gamma}{8h_t^{\beta}}\left(1 + \frac{4\tau_2^2}{2\tau_1^2 + 1}\right)$$

$$<\frac{\gamma}{8h_t^{\beta}}\left\|\nabla F(w_t)\right\|^2, \tag{51}$$

*where the first inequality uses the fact that $\frac{1}{h_t^{\beta}} \leq 1$; the second inequality uses the fact $\gamma \leq \frac{1}{4L_0(2\tau_1^2+1)}$; the third inequality uses the fact $\frac{4\tau_2^2}{2\tau_1^2+1} \leq 1 + \frac{4\tau_2^2}{2\tau_1^2+1}$; the fourth inequality uses the fact that $\left\|\nabla F(w_t)\right\|^2 > 1 + 4\tau_2^2/(2\tau_1^2 + 1)$. Combining equation 49, equation 50, equation 51 gives us the desired result.*

# I  Proof of Theorem 2

**Theorem 2 (Convergence of IAN-SGD)** *Let Assumptions 1, 3 and 4 hold. For the IAN-SGD algorithm, choose learning rate* $\gamma = \min\{\frac{1}{4L_0(2\tau_1^2+1)}, \frac{1}{4L_1(2\tau_1^2+1)}, \frac{1}{\sqrt{T}}, \frac{1}{8L_1(2\tau_1^2+1)(2\tau_2/(1-\tau_1))^\beta}\}$, *and* $A = \frac{1}{1-\tau_1}$
$\delta = \frac{\tau_2}{1-\tau_1}$, $\Gamma = (4L_1\gamma(2\tau_1^2+1))^{\frac{1}{\beta}}$. *Denote* $\Lambda := F(w_0) - F^* + \frac{1}{2}(L_0 + L_1)(1+4\tau_2^2)^2$. *Then, with probability at least* $\frac{1}{2}$, *IAN-SGD produces a sequence satisfying* $\min_{t \le T} \|\nabla F(w_t)\| \le \epsilon$ *if the total number of iteration* $T$ *satisfies*

$$T \ge \Lambda \max\left\{\frac{256\Lambda}{\epsilon^4}, \frac{64L_1(2\tau_1^2+1)(2+2\tau_1)^\beta}{(1-\tau_1)^\beta\epsilon^{2-\beta}}, (2\tau_1^2+1) \cdot \frac{64(L_0+L_1)+128L_1(2\tau_2/(1-\tau_1))^\beta}{\epsilon^2}\right\}. \quad (17)$$

**Proof 8** *By the descent lemma in equation 3 and the update rule of IAN-SGD in equation 14, we have*

$$
\begin{aligned}
&F(w_{t+1}) - F(w_t)\\
&\le \nabla F(w_t)^\top(w_{t+1} - w_t) + \frac{1}{2}(L_0 + L_1\|\nabla F(w_t)\|^\alpha)\|w_{t+1} - w_t\|^2\\
&= -\gamma\frac{\nabla F(w_t)^\top\nabla f_\xi(w_t)}{h_t^\beta} + \frac{1}{2}\gamma^2(L_0 + L_1\|\nabla F(w_t)\|^\alpha)\frac{\|\nabla f_\xi(w_t)\|^2}{h_t^{2\beta}}.
\end{aligned}
\quad (52)
$$

*Taking expectation over* $\xi, \xi'$ *and* $w_t$, *and by the independence between* $\xi$ *and* $\xi'$, *we have*

$$
\begin{aligned}
&\mathbb{E}_{w_t}\left[\left[\mathbb{E}_{\xi,\xi'}\left[F(w_{t+1}) - F(w_t)|w_t\right]\right]\right]\\
&\le \mathbb{E}_{w_t,\xi'}\left[\frac{-\gamma\|\nabla F(w_t)\|^2}{h_t^\beta} + \frac{1}{2}\gamma^2(L_0 + L_1\|\nabla F(w_t)\|^\alpha)\frac{\mathbb{E}_\xi\left[\|\nabla f_\xi(w_t)\|^2|w_t\right]}{h_t^{2\beta}}\right].
\end{aligned}
$$

*From Assumption 4, we have*

$$
\begin{aligned}
\mathbb{E}_\xi\left[\|\nabla f_\xi(w_t)\|^2|w_t\right] &= \mathbb{E}_\xi\left[\|\nabla f_\xi(w_t) - \nabla F(w_t) + \nabla F(w_t)\|^2|w_t\right]\\
&\overset{(i)}{\le} 2\mathbb{E}_\xi\left[\|\nabla f_\xi(w_t) - \nabla F(w_t)\|^2|w_t\right] + 2\|\nabla F(w_t)\|^2\\
&\overset{(ii)}{\le} 2(\tau_1\|\nabla F(w_t)\| + \tau_2)^2 + 2\|\nabla F(w_t)\|^2\\
&\overset{(iii)}{\le} (4\tau_1^2 + 2)\|\nabla F(w_t)\|^2 + 4\tau_2^2,
\end{aligned}
\quad (53)
$$

*where inequalities (i) and (iii) are direct applications of* $(a+b)^2 \le 2a^2 + 2b^2$, *and (ii) follows from Assumption 4. Substituting equation 53 into the above descent lemma, we have*

$$
\begin{aligned}
&\mathbb{E}_{w_t,\xi'}\left[\mathbb{E}_\xi\left[F(w_{t+1}) - F(w_t)|w_t\right]\right]\\
&\le \mathbb{E}_{w_t,\xi'}\left[-\gamma\frac{\|\nabla F(w_t)\|^2}{h_t^\beta} + \frac{1}{2}\gamma^2(L_0 + L_1\|\nabla F(w_t)\|^\alpha)\frac{\mathbb{E}_\xi\left[\|\nabla f_\xi(w_t)\|^2|w_t\right]}{h_t^{2\beta}}\right]\\
&\le \mathbb{E}_{w_t,\xi'}\left[-\gamma\frac{\|\nabla F(w_t)\|^2}{h_t^\beta} + \frac{1}{2}\gamma^2(L_0 + L_1\|\nabla F(w_t)\|^\alpha)\frac{(4\tau_1^2 + 2)\|\nabla F(w_t)\|^2 + 4\tau_2^2}{h_t^{2\beta}}\right]\\
&= \mathbb{E}_{w_t,\xi'}\left[\left(\frac{\gamma}{h_t^\beta}\left(-1 + \gamma(2\tau_1^2 + 1)\cdot\frac{L_0 + L_1\|\nabla F(w_t)\|^\alpha}{h_t^\beta}\right)\right)\|\nabla F(w_t)\|^2 + \frac{1}{2}\gamma^2\frac{L_0 + L_1\|\nabla F(w_t)\|^\alpha}{h_t^{2\beta}}4\tau_2^2\right]. \quad (54)
\end{aligned}
$$

*Next, we provide an upper bound of the term* $\gamma(2\tau_1^2 + 1)\cdot\frac{L_0 + L_1\|\nabla F(w_t)\|^\alpha}{h_t^\beta}$ *in the above equation. By the clipping structure and step size rule, we have* $\frac{1}{h_t^\beta} = \min\left\{1, \frac{1}{4L_1\gamma(2\tau_1^2+1)(\frac{1}{1-\tau_1}\|\nabla f_{\xi'}(w_t)\| + \frac{\tau_2}{1-\tau_1})^\beta}\right\} \le 1$ *and* $\gamma \le \frac{1}{4L_0(2\tau_1^2+1)}$, *which imply that*

$$(2\tau_1^2 + 1)\cdot\frac{\gamma L_0}{h_t^\beta} < (2\tau_1^2 + 1)\cdot\gamma L_0 \le \frac{1}{4}, \quad (55)$$

*On the other hand, we have*

$$
\begin{aligned}
\frac{1}{4}h_t^\beta &\overset{(i)}{\geq} \frac{1}{4}h_t^\alpha \\
&= \frac{1}{4}(h_t^\beta)^{\frac{\alpha}{\beta}} \\
&\overset{(ii)}{\geq} \frac{1}{4}(4\gamma L_1(2\tau_1^2+1))^{\frac{\alpha}{\beta}}\Big(\frac{1}{1-\tau_1}\big\|\nabla f_{\xi'}(w_t)\big\| + \frac{\tau_2}{1-\tau_1}\Big)^\alpha \\
&\overset{(iii)}{\geq} \frac{1}{4}(4\gamma L_1(2\tau_1^2+1))\Big(\frac{1}{1-\tau_1}\big\|\nabla f_{\xi'}(w_t)\big\| + \frac{\tau_2}{1-\tau_1}\Big)^\alpha \\
&\overset{(iv)}{\geq} \gamma L_1(2\tau_1^2+1)\big\|\nabla F(w_t)\big\|^\alpha,
\end{aligned}
\tag{56}
$$

*where (i) utilizes the fact $h_t \geq 1$ and $\beta \geq \alpha$; (ii) utilizes the fact that $h_t^\beta \geq 4L_1\gamma(2\tau_1^2+1)(\frac{1}{1-\tau_1}\big\|\nabla f_{\xi'}(w_t)\big\| + \frac{\tau_2}{1-\tau_1})^\beta$; (iii) utilizes the fact that $\gamma \leq \frac{1}{4L_1(2\tau_1^2+1)}$, thus $(4\gamma L_1(2\tau_1^2+1)) \leq (4\gamma L_1(2\tau_1^2+1))^{\alpha/\beta}$ since $\beta \in [\alpha,1]$; (iv) follows from Lemma 2.*

*Combining equation 55 and equation 56, we conclude that $\gamma(2\tau_1^2+1)\cdot\frac{L_0+L_1\|\nabla F(w_t)\|^\alpha}{h_t^\beta} \leq \frac{1}{2}$, and equation 54 further reduces to*

$$
\begin{aligned}
\mathbb{E}_{w_t}&\Big[\mathbb{E}_{\xi,\xi'}\big[F(w_{t+1})-F(w_t)|w_t\big]\Big] \\
&\leq \mathbb{E}_{w_t,\xi'}\Big[-\frac{\gamma}{2h_t^\beta}\big\|\nabla F(w_t)\big\|^2 + \frac{1}{2}\gamma^2\frac{L_0+L_1\big\|\nabla F(w_t)\big\|^\alpha}{h_t^{2\beta}}4\tau_2^2\Big] \\
&\overset{(i)}{\leq} \mathbb{E}_{w_t,\xi'}\Big[-\frac{\gamma}{2h_t^\beta}\big\|\nabla F(w_t)\big\|^2 + \frac{1}{2}\gamma^2(L_0+L_1)(1+4\tau_2^2)^2 + \frac{\gamma}{4h_t^\beta}\big\|\nabla F(w_t)\big\|^2\Big] \\
&= \mathbb{E}_{w_t,\xi'}\Big[-\frac{\gamma}{4h_t^\beta}\big\|\nabla F(w_t)\big\|^2 + \frac{1}{2}\gamma^2(L_0+L_1)(1+4\tau_2^2)^2\Big],
\end{aligned}
\tag{57}
$$

*where (i) follows from equation 48 in the auxiliary Lemma 5. Rearranging the above inequality yields that*

$$
\mathbb{E}_{w_t,\xi'}\Big[\frac{\gamma}{4h_t^\beta}\big\|\nabla F(w_t)\big\|^2\Big] \leq \mathbb{E}_{w_t}\Big[\mathbb{E}_{\xi,\xi'}\big[F(w_t)-F(w_{t+1})|w_t\big]\Big] + \frac{1}{2}(L_0+L_1)\gamma^2(1+4\tau_2^2)^2. \quad \forall t \in [T]
\tag{58}
$$

*Next, we further lower bound $\frac{\|\nabla F(w_t)\|^2}{h_t^\beta}$ by $\|\nabla F(w_t)\|$ to eliminate the randomness in $h_t$. We have*

$$\gamma \frac{\left\|\nabla F(w_t)\right\|^2}{h_t^\beta}$$

$$\overset{(i)}{=} \gamma \min\left\{1, \frac{1}{4L_1(2\tau_1^2+1)\gamma(\frac{1}{1-\tau_1}\|\nabla f_{\xi'}(w_t)\| + \frac{\tau_2}{1-\tau_1})^\beta}\right\}\left\|\nabla F(w_t)\right\|^2$$

$$\overset{(ii)}{\geq} \gamma \min\left\{1, \frac{1}{4L_1\gamma(2\tau_1^2+1)\left(\underbrace{\frac{\tau_1+1}{1-\tau_1}}_{C_1}\|\nabla F(w_t)\| + \underbrace{\frac{2\tau_2}{1-\tau_1}}_{C_2}\right)^\beta}\right\}\left\|\nabla F(w_t)\right\|^2$$

$$\overset{(iii)}{\geq} \gamma \min\left\{1, \frac{1}{4L_1\gamma(2\tau_1^2+1)\left(2C_1\|\nabla F(w_t)\|\right)^\beta}, \frac{1}{4L_1\gamma(2\tau_1^2+1)\left(2C_2\right)^\beta}\right\}\left\|\nabla F(w_t)\right\|^2$$

$$\overset{(iv)}{=} \min\left\{\gamma, \frac{1}{4L_1(2\tau_1^2+1)\left(2C_1\|\nabla F(w_t)\|\right)^\beta}, \frac{1}{4L_1(2\tau_1^2+1)(2C_2)^\beta}\right\}\left\|\nabla F(w_t)\right\|^2$$

$$\overset{(v)}{=} \min\left\{\gamma, \frac{1}{4L_1(2\tau_1^2+1)\left(2C_1\|\nabla F(w_t)\|\right)^\beta}\right\}\left\|\nabla F(w_t)\right\|^2$$

$$= \min\left\{\gamma\left\|\nabla F(w_t)\right\|^2, \frac{\left\|\nabla F(w_t)\right\|^{2-\beta}}{4(2C_1)^\beta L_1(2\tau_1^2+1)}\right\}, \tag{59}$$

*where (i) expands the expression of $\frac{1}{h_t^\beta}$; (ii) utilizes the second inequality of Lemma 2 to upper bound $\|\nabla f_{\xi'}(w_t)\|$ by $(\tau_1+1)\|\nabla F(w_t)\| + \tau_2$; (iii) utilizes the fact $\frac{1}{(a+b)^\beta} \geq \min\{\frac{1}{(2a)^\beta}, \frac{1}{(2b)^\beta}\}$ by setting $a = \frac{\tau_1+1}{1-\tau_1}\|\nabla F(w_t)\|$, $b = \frac{2\tau_2}{1-\tau_1}$, $\beta \geq 0$; (iv) substitute $\gamma$ inside the minimum operator. Moreover, (v) follows from the step size rule $\gamma = \frac{1}{8L_1(2\tau_1^2+1)(2\tau_2/(1-\tau_1))^\beta} \leq \frac{1}{4L_1(2\tau_1^2+1)(4\tau_2/(1-\tau_1))^\beta}$. Substituting equation 59 into equation 58 and summing over t from 0 to $T-1$, we obtain that*

$$\sum_{t=0}^{T-1}\mathbb{E}_{w_t}\left[\min\left\{\frac{\gamma}{4}\left\|\nabla F(w_t)\right\|^2, \frac{\left\|\nabla F(w_t)\right\|^{2-\beta}}{16L_1(2C_1)^\beta(2\tau_1^2+1)}\right\}\right]$$

$$\leq \sum_{t=0}^{T-1}\mathbb{E}_{w_t,\xi'}\left[\frac{\gamma}{4h_t^\beta}\left\|\nabla F(w_t)\right\|^2\right]$$

$$\leq \sum_{t=0}^{T-1}\mathbb{E}_{w_t}\left[\mathbb{E}_{\xi,\xi'}\left[F(w_t) - F(w_{t+1})|w_t\right]\right] + T\frac{1}{2}\gamma^2(L_0+L_1)(1+4\tau_2^2)^2.$$

*By step size rule $\gamma \leq \frac{1}{\sqrt{T}}$, the above inequality becomes*

$$\sum_{t=0}^{T-1}\mathbb{E}_{w_t}\left[\min\left\{\frac{\gamma}{4}\left\|\nabla F(w_t)\right\|^2, \frac{\left\|\nabla F(w_t)\right\|^{2-\beta}}{16L_1(2C_1)^\beta(2\tau_1^2+1)}\right\}\right] \leq F(w_0) - F^* + \frac{1}{2}(L_0+L_1)(1+4\tau_2^2)^2.$$

*Denote $K = \{t|t \in [T]$ such that $\gamma\left\|\nabla F(w_t)\right\|^2 \leq \frac{\|\nabla F(w_t)\|^{2-\beta}}{4L_1(2C_1)^\beta(2\tau_1^2+1)}\}$, then the above inequality implies that*

$$\sum_{t \in K}\mathbb{E}_{w_t}\left[\frac{\gamma}{4}\left\|\nabla F(w_t)\right\|^2\right] \leq F(w_0) - F^* + \frac{1}{2}(L_0+L_1)(1+4\tau_2^2)^2, \tag{60}$$

*and*

$$\sum_{t \in K^c}\mathbb{E}_{w_t}\left[\frac{\left\|\nabla F(w_t)\right\|^{2-\beta}}{16L_1(2C_1)^\beta(2\tau_1^2+1)}\right] \leq F(w_0) - F^* + \frac{1}{2}(L_0+L_1)(1+4\tau_2^2)^2. \tag{61}$$

*Now denote RHS as $\Lambda := F(w_0) - F^* + \frac{1}{2}(L_0 + L_1)(1 + 4\tau_2^2)^2$, then we have*

$$\mathbb{E}_{w_t}\Big[\min_{t\in T}\|\nabla F(w_t)\|\Big]$$

$$\leq \mathbb{E}_{w_t}\Big[\min\Big\{\frac{1}{|K|}\sum_{t\in K}\|\nabla F(w_t)\|, \frac{1}{|K^c|}\sum_{t\in K^c}\|\nabla F(w_t)\|\Big\}\Big]$$

$$\overset{(i)}{\leq} \mathbb{E}_{w_t}\Big[\min\Big\{\sqrt{\frac{1}{|K|}\sum_{t\in K}\|\nabla F(w_t)\|^2}, \Big(\frac{1}{|K^c|}\sum_{t\in K^c}\|\nabla F(w_t)\|^{2-\beta}\Big)^{\frac{1}{2-\beta}}\Big\}\Big]$$

$$\overset{(ii)}{\leq} \max\Big\{\mathbb{E}_{w_t}\sqrt{\frac{1}{|K|}\sum_{t\in K}\|\nabla F(w_t)\|^2}, \mathbb{E}_{w_t}\Big(\frac{1}{|K^c|}\sum_{t\in K^c}\|\nabla F(w_t)\|^{2-\beta}\Big)^{\frac{1}{2-\beta}}\Big\}$$

$$\overset{(iii)}{\leq} \max\Big\{\sqrt{\frac{1}{|K|}\sum_{t\in K}\mathbb{E}_{w_t}\|\nabla F(w_t)\|^2}, \Big(\frac{1}{|K^c|}\sum_{t\in K^c}\mathbb{E}_{w_t}\|\nabla F(w_t)\|^{2-\beta}\Big)^{\frac{1}{2-\beta}}\Big\}$$

$$\overset{(iv)}{\leq} \max\Big\{\sqrt{(4\Lambda)\frac{4(L_0+L_1)(2\tau_1^2+1) + \sqrt{T} + 8L_1(2\tau_1^2+1)((2\tau_2)/(1-\tau_1))^\beta}{T}},$$

$$\Big(\Lambda\frac{32L_1(2\tau_1^2+1)((2\tau_1+2)/1-\tau_1)^\beta}{T}\Big)^{\frac{1}{2-\beta}}\Big\},$$

*where (i) uses the concavity $\psi(\cdot) = y^{\frac{1}{2}}$ and $\psi(\cdot) = y^{\frac{1}{2-\beta}}$ and Jensen's inequality $\psi(\frac{\sum_i x_i}{n}) \geq \frac{\sum_i \psi(x_i)}{n}$ applied on $\|\nabla F(w_t)\|$; (ii) uses the fact $\mathbb{E}[\min\{A,B\}] \leq \mathbb{E}[A] \leq \max\{\mathbb{E}[A], \mathbb{E}[B]\}$; (iii) applies Jensen's inequality over concave function $\Psi(\cdot) = y^{\frac{1}{2}}$ and $\Psi(\cdot) = y^{\frac{1}{2-\beta}}$; (iv) is due to re-organize equation 60, equation 61 and set $K = T$, $K^c = \frac{T}{2}$ (Since we require either $K$ or $K^c$ should be larger than $\frac{T}{2}$ to guarantee that $K + K_c = T$).*

*Thus, in order to find a point satisfies*

$$Pr(\min_{t\in[T]}\|\nabla F(w_t)\| \geq \epsilon) \leq \frac{1}{2},$$

*We must have $\mathbb{E}_{w_t}[\min_{t\in[T]}\|\nabla F(w_t)\|] \leq \frac{\epsilon}{2}$.*

*This indicates that the RHS of the above inequality should be smaller than $\frac{\epsilon}{2}$, this implies $T$ satisfies*

$$T \geq \Lambda\max\Big\{\frac{256\Lambda}{\epsilon^4}, \frac{64L_1(2\tau_1^2+1)(2+2\tau_1)^\beta}{(1-\tau_1)^\beta\epsilon^{2-\beta}}, (2\tau_1^2+1)\cdot\frac{64(L_0+L_1) + 128L_1((2\tau_2)/(1-\tau_1))^\beta}{\epsilon^2}\Big\}. \quad (62)$$

## J   IAN-SGD under generalized PŁ condition

**Lemma 3 (Descent inequality of IAN-SGD under generalized PŁ condition)** *Let Assumptions 1, 2, 3 and 4 hold. For the IAN-SGD algorithm, choose target error $0 < \epsilon \leq \min\{1, 1/2\mu\}$ and learning rate $\gamma = (2\mu\epsilon)^{\frac{4-2\beta}{\rho}} \cdot \min\{\frac{1}{4L_0(2\tau_1^2+1)}, \frac{1}{4L_1((2\tau_1+2)/(1-\tau_1))(2\tau_1^2+1)}, \frac{1}{8L_1(2\tau_1^2+1)(2\tau_2/(1-\tau_1))^\beta}, \frac{L_1(2\tau_1^2+1)}{16((L_0+L_1)(1+4\tau_2^2)+L_1(\tau_1^2+1/2))^2}\}$, and $A = \frac{1}{1-\tau_1}$ $\delta = \frac{\tau_2}{1-\tau_1}$, $\Gamma = (4L_1\gamma(2\tau_1^2+1))^{\frac{1}{\beta}}$. Then, we have the following descent inequality*

$$\mathbb{E}_{w_{t+1}}[\Delta_{t+1}] \leq \mathbb{E}_{w_t}[\Delta_t - \frac{\gamma^{3/2}(L_1(2\tau_1^2+1))^{1/2}(2\mu\Delta_t)^{(2-\beta)/\rho}}{4} + \frac{\gamma^{3/2}(L_1(2\tau_1^2+1))^{1/2}(2\mu\epsilon)^{(2-\beta)/\rho}}{8}]. \quad (20)$$

**Proof 9** *When chosen target error $0 < \epsilon \leq \min\{1, 1/2\mu\}$ and learning rate satisfy*

$$\gamma \leq (2\mu\epsilon)^{\frac{4-2\beta}{\rho}} \cdot \frac{1}{4L_0(2\tau_1^2+1)} \leq \frac{1}{4L_0(2\tau_1^2+1)}$$

$$\gamma \leq (2\mu\epsilon)^{\frac{4-2\beta}{\rho}} \cdot \frac{1}{4L_1((2\tau_1+2)/(1-\tau_1))(2\tau_1^2+1)} \leq \frac{1}{4L_1(2\tau_1^2+1)}$$

$$\gamma \leq (2\mu\epsilon)^{\frac{4-2\beta}{\rho}} \cdot \frac{1}{8L_1(2\tau_1^2+1)(2\tau_2/(1-\tau_1))^\beta} \leq \frac{1}{8L_1(2\tau_1^2+1)(2\tau_2/(1-\tau_1))^\beta}, \quad (63)$$

*the derivations from equation 52 to equation 58 stated in Proof G are still valid, we omit discussions of detailed derivation here.*

*Next, we lower bound $\frac{\|\nabla F(w_t)\|^2}{h_t^\beta}$ by $\|\nabla F(w_t)\|$ to eliminate the randomness in $h_t$. For any $\beta > \alpha > 0$, we have*

$$\gamma \frac{\left\|\nabla F(w_t)\right\|^2}{h_t^\beta}$$

$$\overset{(i)}{=} \gamma \min\left\{1, \frac{1}{4L_1(2\tau_1^2+1)\gamma(\frac{1}{1-\tau_1}\left\|\nabla f_{\xi'}(w_t)\right\| + \frac{\tau_2}{1-\tau_1})^\beta}\right\}\left\|\nabla F(w_t)\right\|^2$$

$$\overset{(ii)}{\geq} \gamma \min\left\{1, \frac{1}{4L_1\gamma(2\tau_1^2+1)\big(\underbrace{\frac{\tau_1+1}{1-\tau_1}}_{C_1}\left\|\nabla F(w_t)\right\| + \underbrace{\frac{2\tau_2}{1-\tau_1}}_{C_2}\big)^\beta}\right\}\left\|\nabla F(w_t)\right\|^2$$

$$\overset{(iii)}{\geq} \gamma \min\left\{1, \frac{1}{4L_1\gamma(2\tau_1^2+1)\big(2C_1\left\|\nabla F(w_t)\right\|\big)^\beta}, \frac{1}{4L_1\gamma(2\tau_1^2+1)\big(2C_2\big)^\beta}\right\}\left\|\nabla F(w_t)\right\|^2$$

$$\overset{(iv)}{=} \min\left\{\gamma, \frac{1}{4L_1(2\tau_1^2+1)\big(2C_1\left\|\nabla F(w_t)\right\|\big)^\beta}, \frac{1}{4L_1(2\tau_1^2+1)(2C_2)^\beta}\right\}\left\|\nabla F(w_t)\right\|^2$$

$$\overset{(v)}{=} \min\left\{\gamma, \frac{1}{4L_1(2\tau_1^2+1)\big(2C_1\left\|\nabla F(w_t)\right\|\big)^\beta}\right\}\left\|\nabla F(w_t)\right\|^2$$

$$= \min\left\{\gamma\left\|\nabla F(w_t)\right\|^2, \frac{\left\|\nabla F(w_t)\right\|^{2-\beta}}{4(2C_1)^\beta L_1(2\tau_1^2+1)}\right\}$$

$$\overset{(vi)}{\geq} \min\left\{\gamma(L_1\gamma(2\tau_1^2+1))^{\beta/2}\|\nabla F(w_t)\|^{2-\beta} - \gamma^2 L_1(2\tau_1^2+1), \frac{\left\|\nabla F(w_t)\right\|^{2-\beta}}{4(2C_1)^\beta L_1(2\tau_1^2+1)}\right\}$$

$$\overset{(vii)}{\geq} \min\left\{\gamma^{3/2}(L_1(2\tau_1^2+1))^{1/2}\|\nabla F(w_t)\|^{2-\beta} - \gamma^2 L_1(2\tau_1^2+1), \frac{\left\|\nabla F(w_t)\right\|^{2-\beta}}{4(2C_1)^\beta L_1(2\tau_1^2+1)}\right\}$$

$$\overset{(viii)}{\geq} \min\left\{\gamma^{3/2}(L_1(2\tau_1^2+1))^{1/2}\|\nabla F(w_t)\|^{2-\beta}, \frac{\left\|\nabla F(w_t)\right\|^{2-\beta}}{4(2C_1)^\beta L_1(2\tau_1^2+1)}\right\} - \gamma^2 L_1(2\tau_1^2+1)$$

$$\overset{(x)}{\geq} \gamma^{3/2}(L_1(2\tau_1^2+1))^{1/2}\|\nabla F(w_t)\|^{2-\beta} - \gamma^2 L_1(2\tau_1^2+1)$$

$$\overset{(ix)}{\geq} \gamma^{3/2}(L_1(2\tau_1^2+1))^{1/2}(2\mu\Delta_t)^{(2-\beta)/\rho} - \gamma^2 L_1(2\tau_1^2+1) \tag{64}$$

*where (i)-(v) follows the same arguments stated in Proof G. (vi) applies equation 29 by setting $C = (L_1\gamma(2\tau_1^2+1)^{\beta/2})$ (since $L_1\gamma(2\tau_1^2+1) < 1$), $x = \|\nabla F(w_t)\|$, $\Delta = \beta$, $\omega' = 2, \omega = 2 - \beta$; (vii) follows from $(L_1\gamma(2\tau_1^2+1))^{\beta/2} \geq (L_1\gamma(2\tau_1^2+1))^{1/2}$ as $(L_1\gamma(2\tau_1^2+1)) < 1$; (viii) follows from fact for any $b \geq 0$, $\min\{a-b,c\} \geq \min\{a,c\} - b$; (x) follows from fact $\gamma \leq \frac{1}{4L_1((2\tau_1+2)/(1-\tau_1))(2\tau_1^2+1)} \leq \frac{1}{4L_1(2C_1)^\beta(2\tau_1^2+1)}$. Thus, $L_1\gamma(2\tau_1^2+1) < 1$ and $\gamma^{3/2}(L_1(2\tau_1^2+1))^{1/2} \leq \gamma \leq \frac{1}{4(2C_1)^\beta(2\tau_1^2+1)}$ always holds; (ix) substitutes assumption 2 to replace $\|\nabla F(w_t)\|$ by $(2\mu\Delta_t)^{1/\rho}$.*

*When $\alpha = 0, \beta > 0$, above argument still applies, next, we consider the case when $\alpha = \beta = 0$, we have*

$$\gamma\|\nabla F(w_t)\|^2 \geq \gamma(L_1\gamma(2\tau_1^2+1))^{1/2}\|\nabla F(w_t)\|^2$$

$$\geq \gamma(L_1\gamma(2\tau_1^2+1))^{1/2}\|\nabla F(w_t)\|^2 - \gamma^2 L_1(2\tau_1^2+1)$$

$$\geq \gamma(L_1\gamma(2\tau_1^2+1))^{1/2}(2\mu\Delta_t)^{2/\rho} - \gamma^2 L_1(2\tau_1^2+1)$$

$$= \gamma^{3/2}(L_1(2\tau_1^2+1))(2\mu\Delta_t)^{2/\rho} - \gamma^2 L_1(2\tau_1^2+1), \tag{65}$$

*where the first inequality is due to $\gamma \leq \frac{1}{L_1(2\tau_1^2+1)}$ and the last inequality substitutes assumption 2 to replace $\|\nabla F(w_t)\|$ by $(2\mu\Delta_t)^{1/\rho}$.*

*Combining two cases, we conclude that for any $\beta \geq 0$, we always have*

$$\gamma \frac{\left\|\nabla F(w_t)\right\|^2}{h_t^\beta} \geq \gamma^{3/2}(L_1(2\tau_1^2 + 1))(2\mu\Delta_t)^{(2-\beta)/\rho} - \gamma^2 L_1(2\tau_1^2 + 1) \tag{66}$$

*Substituting equation 66 into equation 58 and re-arranging terms, we have*

$$\mathbb{E}_{w_t}\left[\frac{\gamma^{3/2}(L_1(2\tau_1^2 + 1))^{1/2}(2\mu\Delta_t)^{(2-\beta)/\rho}}{4}\right]$$
$$\leq \mathbb{E}_{w_t}\left[\left[F(w_t) - F(w_{t+1})\right]\right] + \frac{1}{2}(L_0 + L_1)\gamma^2(1 + 4\tau_2^2)^2 + \frac{\gamma^2 L_1(2\tau_1^2 + 1)}{4}.$$
$$= \mathbb{E}_{w_t}\left[\left[F(w_t) - F(w_{t+1})\right]\right] + \frac{1}{2}\gamma^2\left((L_0 + L_1)(1 + 4\tau_2^2)^2 + \frac{L_1(2\tau_1^2 + 1)}{2}\right) \tag{67}$$

*Substituting $F^*$ on both sides and re-arranging terms yields that*

$$\mathbb{E}_{w_{t+1}}[\Delta_{t+1}] \leq \mathbb{E}_{w_t}[\Delta_t - \frac{\gamma^{3/2}(L_1(2\tau_1^2 + 1))^{1/2}(2\mu\Delta_t)^{(2-\beta)/\rho}}{4} + \frac{1}{2}\gamma^2\left((L_0 + L_1)(1 + 4\tau_2^2)^2 + L_1(\tau_1^2 + 1/2)\right)].$$

*Since $\gamma < \frac{L_1(2\tau_1^2+1)(2\mu\epsilon)^{(4-2\beta)/\rho}}{16((L_0+L_1)(1+4\tau_2^2)+L_1(\tau_1^2+1/2))^2}$, the above inequality further implies*

$$\mathbb{E}_{w_{t+1}}[\Delta_{t+1}] \leq \mathbb{E}_{w_t}[\Delta_t - \frac{\gamma^{3/2}(L_1(2\tau_1^2 + 1))^{1/2}(2\mu\Delta_t)^{(2-\beta)/\rho}}{4} + \frac{\gamma^{3/2}(L_1(2\tau_1^2 + 1))^{1/2}(2\mu\epsilon)^{(2-\beta)/\rho}}{8}], \tag{68}$$

*which gives the desired result.*

