# OpenReview forum: "Adaptive Gradient Normalization and Independent Sampling for (Stochastic) Generalized-Smooth Optimization"
_TMLR — Accepted by TMLR_

### Review · Reviewer_qBxj · 2025-03-04

**Summary Of Contributions:**

This work analyzes the convergence rates of optimization algorithms that leverage ‘adaptive gradient normalization’ under an objective function satisfying generalized Lipschitz smoothness condition.

- With deterministic gradient oracles, it studies the (sub-)linear convergence of ‘adaptively normalized gradient descent (AN-GD)’ under an additional assumption of generalized PŁ condition. The authors claim that their analysis provides an intuition about the impact of the normalization parameter $\beta$ to the convergence rate.
- Under stochastic (unbiased) gradient oracles, it proposes a novel algorithm called ‘independent-adaptively normalized stochastic gradient descent (IAN-SGD)’ and studies its $\mathcal{O}(\epsilon^{-4})$ convergence rate under an additional assumption about moderate gradient noises.

The authors test the efficacy of their proposed algorithm numerically.

**Audience:**

Yes

**Broader Impact Concerns:**

Since this work focuses on the theoretical aspects of optimization algorithms, the Broader Impact Statement section seems unnecessary.

**Claims And Evidence:**

No

**Requested Changes:**

**Required Changes**

- Every point I listed above in the weaknesses should be resolved.

**Recommended Changes & Some Questions expected to be answered**

- In the abstract, there seems to be a couple of nearly duplicate sentences: “We analyze the overall affects of adaptive normalization and function geometry on convergence rate. Our results provide a comprehensive understanding of the interplay between adaptive gradient normalization and function geometry.” Can these be more concise?
- In the second sentence of Section 1, the words “$L$-smoothness fails to characterize the global geometry” are not very clear to me. What does “to fail to characterize the geometry of functions” mean?
- The “notation” paragraph/section will strengthen the rigor of the paper. It never specifies which type of norm is used in it explicitly, which seems to be a standard Euclidean $\ell_2$ norm.
- The motivation of introducing generalized PŁ geometry is weak. Why does it resolve the theory-practice gap, as the authors mentioned? Do the examples provided in the experiments satisfy the generalized PŁ geometry?
- The main text should contain the pointers to the proof for each theorem, proposition, and lemma.
- The symbol $\nabla f_{\xi_B}(w_t)$ is a bit weird. How about $\nabla f_{B_t} (w_t)$, where $B_t = \\{\xi \text{ sampled at time }t\\}$?
- Discussion about choosing a good normalization parameter $\beta$ will strengthen the paper.
- It seems nice to compare proof techniques with those of Zhang et al. (2019). But isn’t there any difference in the convergence rate? The comparison regarding the convergence rate will definitely strengthen the paper.
- Typos:
    - Put a hyphen: “Kuradyka-Łojasiewicz”.
    - In Theorem 1, be consistent with $t$ or $T$. recommend to write like “in order to achieve $\Delta_T\le \epsilon$, the total number of iteration satisfies $T=\Omega(…)$”. Also, For the case III, the expression for $\Delta_t$ needs an additional parantheses: $\mathcal{O}\left(\left(…\right)^{\left(\frac{\rho}{2-\beta}\right)^{t-T_0}}\right)$.
    - Section D, Lemma 3: the original lemma is actually Lemma E.1 of the arXiv version of Chen et al. (2023), which is Lemma 5 of the ICML version of the same work.
    - Section D, Lemma 4: in the long series of (in)equalities at the beginning of the proof, a plus symbol (’+’) is missing in the third line.
    - Section F, Proposition 3, Equation (38): I guess it must be corrected as $\Delta_{t+1} \le \Delta_t - \gamma \mu^2 \Delta_t^2$, or no?
    - Section G: below the equation (45), there are a couple of duplicate words: “equation equation 53”.
    - Section H, Remark 3: a missing square. “… $\le \frac1B(2\tau_1^2\lVert \nabla F(w) \rVert^2 + 2\tau_2^2)$”.

**Strengths And Weaknesses:**

**Strengths**

- S1. The analysis under generalized Lipschitz smoothness is well motivated. Remarks 1 and 2 clearly show the relationship between other related definitions of generalized smoothness. Propositions 1 and 2 provide clear examples of machine learning problems satisfying the generalized smoothness condition.
- S2. This work provides a fine-grained convergence analysis of AN-GD under generalized smoothness & generalized PŁ condition.
- S3. The idea behind the proposed algorithm (IAN-SGD) is interesting. Also, the algorithm has a convergence guarantee under generalized smoothness.

**Weaknesses**

- **W1. Invalid Comparison Between AN-GD and vanilla GD**
    - Proposition 3 is used to claim the superiority of AN-GD over GD, thereby arguing the benefits of adaptive normalization. However, the proposition only guarantees the convergence upper bound of GD. Suppose the authors wanted to claim that AN-GD is a better algorithm than GD under generalized smoothness + generalized PL condition. In that case, they should have compared the **lower** convergence bound for GD with their convergence upper bound for AN-GD in Theorem 1.
    - Even if the comparison between Theorem 1 and Proposition 3 were valid, the authors’ claim that GD requires a very small learning rate might not be true. The main reason why they claim that GD is worse than AN-GD is that the constant $G$, an upper bound of gradient norms along the optimization trajectory, can be large. Then when is it a problem? Recall that, when $\rho=\alpha=\beta=1$, the sample complexity until $\epsilon$-convergence for AN-GD (GD, resp.) is $\mathcal{O}(\tfrac1\epsilon\log(\tfrac1\epsilon))$ ($\mathcal{O}(\tfrac{G}\epsilon)$, resp.). Then GD converges slower than AN-GD when $G\gtrsim \log(\tfrac1\epsilon)$. Suppose our target error is $\epsilon=e^{-r}$ for some $r>1$. Let’s say $G=\Omega(r)$. Then in Proposition 3, the learning rate of GD is $\gamma  = \mathcal{O} (r^{-1})$. On the other hand, to guarantee the AN-GD’s convergence rate in Theorem 1, we need to set $\gamma = \mathcal{O}(\epsilon^{\beta/\rho})=\mathcal{O}(\epsilon)=\mathcal{O}(e^{-r})$: the AN-GD uses an exponentially small learning rate, which can be much smaller than the learning rate choice of GD.
    - By the way, I cannot understand why the generalized PŁ condition is introduced only in Section 4. Does this condition have any relationship with generalized smoothness? Also, the range $\rho\le 2$ should be specified in Assumption 2, or not?
- **W2. Issues in Theorem 1.**
    - I read the whole proof of Theorem 1 and I found some errors and major mistakes. I omit minor typos from here, though there are many.
    - In almost the last lines of the proof of Case I ($\rho < 2-\beta$), I cannot understand why we get a complexity $T=\mathcal{O}\left(\left(\tfrac1\epsilon\right)^{\tfrac{2-\rho-\beta}{\rho}} + \left(\tfrac1\epsilon\right)^{\tfrac{\beta}{\rho}}\right)$. Isn’t the “taking logarithm we have …” part just giving us $T=\mathcal{O}\left(\left(\tfrac1\epsilon\right)^{\tfrac{2-\rho}{\rho}}\right)$, which is a strictly worse rate than that described in the theorem?
    - In order to take advantage of Lemma 4, Theorem 1 must state that the target error (NOT a target “accuracy!”) should satisfy $0<\epsilon \le \min\\{1, 1/(2\mu)\\}$.
    - The Case III ($1\ge \beta > 2-\rho$) in the main theorem statement should contain the fact that $T_0$ is chosen so that $\Delta_{T_0} < \gamma^{\frac{\rho}{\rho+\beta-2}}$; otherwise the expression in Equation (9) may not converge to zero.
    - The proof of the Case III begins with the fact that $\{\Delta_t\}$ converges to zero. But why does the proof consider the convergence in terms of $\epsilon\downarrow 0$, instead of $t\to\infty$? I guess the latter (in terms of $t$) is correct.
- **W3. Regarding the Implementation Detail of IAN-SGD (Equation (14))**
    - Why is the normalization factor $h_t$ defined as a form of $\max\\{1,…\\}$? According to this expression, $h_t=1$ when $\lVert \nabla f_{\xi_{B’}}(w_t) \rVert$ is small enough, which can happen in the later phase of the training (near-convergence). In this case, doesn’t the later behavior of IAN-SGD become nearly identical to vanilla SGD? If so, the $\mathcal{O}(\epsilon^{-4})$ rate in Theorem 2 is not very surprising.
    - There is no description about $\delta$. Is a term for maintaining numerical stability? What does it mean by “…IAN-SGD requires estimating the value of $\delta$ …”?
- **W4. Regarding Assumption 4 and the batch size choices in Theorem 2**
    - The authors claim that Assumption 4 is “strictly weaker” than the bounded gradient noise assumption ($\tau_1=0$). However, the assumption only allows $\tau_1, \tau_2>0$. Moreover, Lemma 2, a key lemma to prove Theorem 2, considers $\tau_1 > 0$. Then, is the claim valid?
    - Even in Section 1.1, the authors claim that they require only a “constant-level” batch size. In my opinion, however, this is misleading and can make the readers misunderstand the result. In fact, the batch size must satisfy $\Omega(\tau_1^2)$, which is proportional to the variance of the stochastic gradients. Thus, the proved convergence guarantee requires the batch size that depends on the problem instance, which cannot be claimed as “constant-level”. I have checked the proof of Theorem 2 and Lemma 2; I found they break down unless $B=\Omega(\tau_1^2)$.
    - Lastly, the sentence “By normalizing during data pre-processing, the value of $\tau_1$ can be approximately controlled as $\mathcal{O}(1)$ in practice” is not very straightforward to me.

---

> ### Author Response · Authors · 2025-04-21
> **Response to reviewer qBxj**
>
> **Q1** Proposition 3 (Proposition 1 in the revision) only guarantees the convergence upper bound of GD. Suppose the authors wanted to claim that AN-GD is a better algorithm than GD under generalized smoothness + generalized PL condition. In that case, they should compare the lower convergence bound for GD with their convergence upper bound for AN-GD in Theorem 1.
>
> **A**: Good point. We agree that Proposition 3 (Proposition 1 in the revision) only presents the $\Omega(\frac{G}{\epsilon})$ complexity of GD in the special setting of $\rho=\alpha=1$. We want to clarify that our AN-GD obtains a comparable complexity order (from Theorem 1) and we did not claim a lower complexity. Our key point is that the analysis of AN-GD does not rely on the gradient norm upper bound $G$, which is needed in the analysis of GD to handle the generalized-smoothness, and this $G$ can be a large numerical constant that limits the choice of learning rate and affects the practical performance. On the other hand, in the existing work (Chen et al., 2023), it has been shown in their Theorem 2 that both GD and AN-GD obtains the complexity lower bound of generalized-smooth nonconvex optimization, and hence both are optimal deterministic methods. In the revision, we have added the above clarification to the paragraph after the new Proposition 1.
>
> **Q2:** Counter-example claim, the AN-GD uses an exponentially small learning rate, which is much smaller than the learning rate choice of GD.
>
> **A:** Good question. The counter-example provided by the reviewer considers a small target error, i.e., $\epsilon = \exp(-r)$ for some $r>0$. Then, it is argued that if $G\lesssim \log(\frac{1}{\epsilon})=r$, the learning rate choice of AN-GD is more restrictive than GD.
> We notice that the key condition in the above argument is $G\lesssim r$, i.e., $G$ is bounded by a small numerical number, which is unlikely the case in generalized-smooth optimization. Specifically, if we set $x=x_0$, $x'=x^*$, where $x^*$ is a stationary point, then we have $||\nabla f(x_0) ||\leq (L_0)^{\alpha}||x_0-x^* ||$. This implies that $G$ is at least $\Omega(||x_0-x^* ||)$, which is a large number when the initialization is far from $x^*$ (typically holds in practice). Of course, if $r$ is large (i.e., $\epsilon$ is extremely small), then AN-GD's learning rate can be conservative due to normalization around the stationary points.
>
> **Q3:** The time complexity for theorem 1 should be $T=\mathcal{O}\big(\big(\frac{1}{\epsilon}\big)^{\frac{2-\rho}{\rho}}\big)$ instead of
> $T=\mathcal{O}\big(\big(\frac{1}{\epsilon}\big)^{\frac{2-\rho-\beta}{\rho}}+\big(\frac{1}{\epsilon}\big)^{\frac{\beta}{\rho}}\big)$, which is a strictly worse rate than that described in the theorem?
>
> **A:** We apologize for this calculation error, and thanks for pointing this out. We found that there is a calculation error in the last step of case I's proof. In the revision, we update statements of Theorem 1 and the correct rates should be $T\geq \max (\frac{8\rho(8(L_0+L_1)+1)}{(2-\beta-\rho)(2\mu)^{2/\rho}\epsilon^{(2-\rho)/{\rho}}}, \frac{1}{(2^{(2-\beta-\rho)/(2-\beta)}-1)(\Delta_0\epsilon)^{(2-\beta-\rho)/{\rho}}} ) = \Omega((\frac{1}{\epsilon})^{\frac{2-\rho}{\rho}})$ when $\beta<2-\rho$. This result shows that when $\rho+\beta<2$, the effect of adaptive normalization can be marginal.
>
> **Q4:** Why is the normalization factor defined as a clipping form of $\max \{(4L_1\gamma)^{\frac{1}{\beta}}(2||\nabla f(w_t)||+\delta),1 \}$? According to this expression,  $h_t=1$ when $||\nabla f(w_t) ||$ is small enough, which can happen in the later phase of the training (near-convergence). In this case, doesn’t the later behavior of IAN-SGD become nearly identical to vanilla SGD? If so, the $\mathcal{O}(\epsilon^{-4})$ rate in Theorem 2 is not very surprising.
>
> **A:** Great question. Yes, intuitively, when gradient becomes small in the later optimization phase, the term $h_t=1$ and IAN-SGD reduces to SGD. This is consistent with the fact that the generalized-smooth condition reduces to the standard smooth condition when gradient approaches zero. On the other hand, from a technical point of view, the gradient clipping structure ensures a lower bound on the stochastic gradient in Lemma 2, which is essential for establishing the convergence proof.
> We do agree that the sample complexity $\mathcal{O}(\epsilon^{-4})$ is not surprising, and our goal is to utilize independent sampling and adaptive gradient normalization to design a variant of clipped SGD that can converge under the relaxed noise Assumption 4 using small batch size. In this revision, we established new Lemma 2 and Theorem 2 with convergence guarantee under batch size equals to 1.

---

> > ### Author Response · Authors · 2025-04-21
> > **Response to reviewer qBxj (Continued)**
> >
> > **Q5**:The authors claim that Assumption 4 is “strictly weaker” than the bounded gradient noise assumption ($\tau_1=0$). However, the assumption only allows $\tau_1, \tau_2>0$. Moreover, Lemma 2, a key lemma to prove Theorem 2, considers $\tau_1\ge 0$. Then, is the claim valid?
> >
> > **A**: Great question! In this revision, we have updated both Lemma 2 and Theorem 2 to fix this issue. To elaborate, the updated Assumption 4 considers $0\le \tau_1 <1$ and $\tau_2 >0$. Then, we can establish the new Lemma 2 and Theorem 2. These new results as well as their proof are similar to their previous versions, with slightly different numerical constants. Following the Theorem 2, we add a remark showing that when $\tau_1 = 0$, our results recovers sample complexity in Zhang et al (2019) up to differences with constant coefficients. Through this way, our analysis is strictly weaker than Zhang et al (2019) since the sample complexity can be generalized under relaxed noise assumption by allowing $\tau_1>0$.
> >
> > References: Zhang, Jingzhao, et al. "Why gradient clipping accelerates training: A theoretical justification for adaptivity." arXiv preprint arXiv:1905.11881 (2019).
> >
> > **Q6**:The batch size must satisfy $\Omega(\tau_1^2)$, which is proportional to the variance of the stochastic gradients. Thus, the proved convergence guarantee requires the batch size that depends on the problem instance, which cannot be claimed as “constant-level”.
> >
> > **A:** Good point, and we agree with the reviewer's comment. In the revision, we have established the new Lemma 2 and Theorem 2 with batch size $B=1$ to address this issue (we realize that using $B=\Omega(\tau_1^2)$ is redundant). In particular, the proof of the new Lemma 2 directly follows from Assumption 4 and triangle inequality, which is different from the proof of the previous mini-batch version.
> >
> > **Minor Comments**
> >
> > **Q7**: Why generalized PL condition is introduced only in Section 4. Does this condition have any relationship with generalized smoothness? Also, the range $\rho\leq 2$ should be specified in Assumption 2, or not?
> >
> > **A**:  We note that in the stochastic setting, a similar recursion can be established for IAN-SGD under the same generalized PL condition. In the revision, we have included the details in the Lemma 5 and Appendix J.
> > There is no explicit relationship between generalized-smooth condition and generalized PL condition. By introducing the generalized PL condition in Section 4, we aim to quantify the effects of adaptive normalization (controlled by $\beta$) under various geometric conditions characterized by the generalized PL condition and generalized smoothness.
> > We have specified the condition $\rho\leq 2$ in the assumption in the revision.
> >
> > **Q8**: Theorem 1 must state that the target error (NOT a target “accuracy!”) should satisfy $\epsilon \leq \min (1,\frac{1}{2\mu})$
> >
> > **A**: We have updated the description of $\epsilon$ in theorem 1 and added the requirements of $0<\epsilon \leq \min (1, 1/2\mu)$ in the statement.
> >
> > **Q9**:The Case III in the main theorem statement should contain the fact that $T_0$ is chosen so that $\Delta_{T_0}\leq \mathcal{O}(\gamma^{\frac{\rho}{\rho+\beta-2}})$; otherwise the expression in Equation (9) may not converge to zero.
> >
> > **A**: We have updated the statement of theorem 1 to include the condition on $T_0$.
> >
> > **Q10**:The proof of the Case III begins with the fact that  converges to zero. But why does the proof consider the convergence in terms of t, instead of $\epsilon$?
> >
> > **A**: We apologize for the misleading proof logic. We have rewritten the proof of Theorem 1 (see Appendix E) with clear logic, and we do not require the target error converges to zero.
> >
> > **Q11**: There is no description about $\delta$. Is it a term for maintaining numerical stability? What does it mean by “…IAN-SGD requires estimating the value of  …”?
> >
> > **A**: Yes. We apologize for the misleading sentence. We have deleted this sentence in the revision. As we have shown in the ablation study, the choice of $\delta$ is quite flexible -- the algorithm converges at comparable speeds with $\delta=\{ 1e^{-8}, 1e^{-3}, 1e^{-1}, 1, 10\}$. Thus, there is no need to estimate the exact value of $\delta$ accurately in practice.
> >
> > **Q12**: the sentence “By normalizing during data pre-processing, the value of $\tau_1$ can be approximately controlled as $\mathcal{O}(1)$ in practice” is not very straightforward to me.
> >
> > **A**: We apologize for the confusion. This sentence has been removed in the revised version. In our experiments, we demonstrate that with standard data pre-processing and proper model initialization, IAN-SGD can converge in practice without requiring large batches of independent samples. In this revision, we also add ablation study on verifying IAN-SGD can converge using extreme small batch size, i.e., batch size $B=B'=\{1,2,4 \}$. More details can be found at Appendix A.4.

---

> ### Author Response · Authors · 2025-04-21
> **Response to reviewer qBxj (Continued)**
>
> **Q13**:In the abstract, there seems to be a couple of nearly duplicate sentences: “We analyze the overall affects of adaptive normalization and function geometry on convergence rate. Our results provide a comprehensive understanding of the interplay between adaptive gradient normalization and function geometry.” Can these be more concise?
>
> **A**: Thanks for your suggestion, in the revision, we changed our abstract sentence to " We analyze the convergence behavior of using adaptive normalization under function geometries characterized by generalized smoothness and generalized PL conditions. Our results provide theoretical insights into adaptive normalization across various scenarios.", which corresponds to the statements of Theorem 1 and its remarks which list some special choices of adaptive normalization $\beta$ under certain geometry characterized by $\rho$.
>
> **Q14**:In the second sentence of Section 1, the words “L-smoothness fails to characterize the global geometry” are not very clear to me. What does “to fail to characterize the geometry of functions” mean?
>
> **A**: We change this sentence to "it has been shown recently that the $L$-smoothness condition fails to hold in many nonconvex machine learning problems...". We want to emphasis that $L$-smooth condition may not hold in modern ML problems. Thus, it's induced function geometry characterized by descent lemma may not hold in practice.
>
> **Q15**: The “notation” paragraph/section will strengthen the rigor of the paper. It never specifies which type of norm is used in it explicitly, which seems to be a standard Euclidean  $\ell_2$ norm.
>
> **A**: Thanks for the suggestion, we have added a notation section that summarizes the main notations used in this paper and specify the norm to be $\ell_2$-norm over Euclidean space.
>
> **Q16**: The motivation of introducing generalized PL geometry is weak. Why does it resolve the theory-practice gap, as the authors mentioned? Do the examples provided in the experiments satisfy the generalized PL geometry?
>
> **A**: Thanks for pointing it out. We have references to explain the motivation of inducing generalized PL condition.
> According to Zhou et al.(2016), Phase retrieval satisfies  PL condition with $\rho=2$ under mild assumptions. For DRO's experiments, we choose $f^*(t)=\frac{1}{4}(t+2)^2_{+}-1$, which is the conjugate dual of $\chi$-square divergence according to Jin et al. (2021)). This function is strongly convex in positive domain. In experiments, we restrict ourselves in regression task where the loss is regularized quadratic function. As a result, the DRO objective in our experiment satisfies PL condition under mild assumptions.
> Regarding on PL condition in deep neural networks, Liu et al.(2022), Scaman et al.(2022) showed that
> PL conditions and its variations hold in deep neural networks under mild assumptions. We have included these clarifications in our experiments section
>
> References:
>
> Zhou, Yi, Huishuai Zhang, and Yingbin Liang. "Geometrical properties and accelerated gradient solvers of non-convex phase retrieval." 2016 54th Annual Allerton Conference on Communication, Control, and Computing (Allerton). IEEE, 2016.
>
> Jin, Jikai, et al. "Non-convex distributionally robust optimization: Non-asymptotic analysis." Advances in Neural Information Processing Systems 34 (2021): 2771-2782.
>
> Chaoyue Liu, Libin Zhu, and Mikhail Belkin. Loss landscapes and optimization in over-parameterized non-
> linear systems and neural networks. Applied and Computational Harmonic Analysis, 59:85–116, 2022a.
>
> Kevin Scaman, Cedric Malherbe, and Ludovic Dos Santos. Convergence rates of non-convex stochastic
> gradient descent under a generic lojasiewicz condition and local smoothness. In Proceedings of the 39th
> International Conference on Machine Learning, Proceedings of Machine Learning Research. PMLR, 17–23
> Jul 2022.
>
> **Q17**:The main text should contain the pointers to the proof for each theorem, proposition, and lemma.
>
> **A**: We have added the pointers of proof in the revision.
>
> **Q18**: The symbol $\nabla f_{\xi_B}(w_t)$ is a bit weird. How about $\nabla f_{B_t}(w_t)$?
>
> **A**: Great Suggestion. We have followed your advice and changed the notation. Since now Theorem 2 is established under two independent samples, we now use $\xi$ and $\xi'$ to denote them.
>
> **Q19**: Discussion about choosing a good normalization parameter $\beta$ will strengthen the paper.
>
> **A**: Thanks for suggestion. We have added two paragraphs at the end of experiment section to summarize the usage of adaptive normalization $\beta$, learning rate, stabilization term $\delta$ and independent samples' batch size based on our observations from phase retrieval, DRO, deep network training and ablation studies.
>
> References: Zhang, Jingzhao, et al. "Why gradient clipping accelerates training: A theoretical justification for adaptivity." arXiv preprint arXiv:1905.11881 (2019)

---

> ### Author Response · Authors · 2025-04-21
> **Response to reviewer qBxj (Continued)**
>
> **Q20**: It seems nice to compare proof techniques with those of Zhang et al. (2019). But isn’t there any difference in the convergence rate?
>
> **A**: We have added a remark under Theorem 2 to compare our results with theirs. To summarize, our time complexity result is established under the more general affine bounded noise. In the special case of $\tau_1=0, \beta=1$, our results match (zhang et al.2019) up to constant factors.
>
> References: Zhang, Jingzhao, et al. "Why gradient clipping accelerates training: A theoretical justification for adaptivity." arXiv preprint arXiv:1905.11881 (2019)
>
> **For typos pointed out by reviewer qBxj, we have corrected them accordingly.** At last, we would like to thank reviewer qBxj for the efforts in reviewing our manuscript and providing valuable feedbacks. Please let us know if further clarifications are needed.

---

### Review · Reviewer_4DJr · 2025-03-20

**Summary Of Contributions:**

The paper studies normalized/clipped SGD under the generalized smoothness assumption where smoothness can grow with the gradient norm. The main novelty of the paper is proposing to use two independent batches, one for the gradient and the other one for its norm, doing this simplifies the analysis of normalized/clipped SGD.

**Audience:**

Yes

**Claims And Evidence:**

No

**Requested Changes:**

- Change Th1 to only work with the precision $\varepsilon$ to stay consistent with Th2.

- Discuss why you did not consider PL in the stochastic case. Again, you should be consistent throughout the paper.

- Include more prior work using the generalized smoothness assumption (I already provided two) and most importantly discuss what do you bring new compared to what is in the previous works (especially the ones I cited concerning the noise assumption and the dependence on $L_1$).

- Go through the proofs and make sure everything is okay.

**Strengths And Weaknesses:**

**Strenghts**
- Studying AN-GD under the generalized PL condition is nice, I am not sure if it has been done before, but it is nice.

- The noise assumption 4, although it needs to be satisfied almost surely, is interesting and has been considered in lots of prior work for SGD with noise bounded in expectation, it is indeed an interesting relaxation in this setting. The problem of doing this analysis with noise bounded in expectation was already done in prior work like [Koloskova et al. (2023)](https://openreview.net/pdf?id=C3DXiFTrve), other works such as [Zhang et al. (2020)](https://jkjin.com/publication/example/example.pdf) (you already cite this one) also claim that they did not do it only for simplicity.

- The writing is good, ignore minor typos and grammatical errors (the authors should go through the paper and correct the obvious typos).

**Weaknesses**

- The novelty of the paper, in my humble opinion (based on familiarity with the setting of the paper), is very limited.

- The overall organization (not the of the paper) is not good. I did not like how the PL condition was just ignored in the stochastic case. Also, theorem 1 works with both the precision $\varepsilon$ and the number of iterations $t$, the authors should either present their results in terms of the complexity of the algorithm as a function of the precision or the error in terms of the number of iterations, mixing both makes it slightly difficult to understand the result.

- The dominating term in Th2 is $\mathcal{O}(\frac{\Delta + L_0 + L_1}{\varepsilon^4})$, might (the assumption are slightly different) be worse than what was obtained in previous works such as [Zhang et al. (2020)](https://jkjin.com/publication/example/example.pdf)  for which this term does not depend on $L_1$, anyways, this needs to be discussed.

- Proof 7 is wrong (it is the first one I checked). You claim that the convexity of squared $L_2$ norm gives something like $\|x + y\|^2 \leq \|x\|^2 + \|y\|^2$ is wrong, you should use the expectation.

---

> ### Author Response · Authors · 2025-04-21
> **Response to reviewer 4DJr**
>
> Thank you very much for the effort into reviewing our manuscript and providing valuable feedback. Below is a response to the review questions. We have revised the manuscript accordingly and uploaded a revised manuscript. Please let us know if further clarifications are needed.
>
> **Q1**: Proof 7 is wrong. Go through the proofs and make sure everything is correct.
>
> **A**: We apologize for the misuse of Jensen's inequality in Proof 7 and thanks a lot for pointing it out. In the revision, we have double-checked and polished all the proofs throughout the paper.  Regarding the Proof 7, we realize that it is unnecessary to use the batchsize $B=O(\tau_1^2)$, and we fixed this technical issue by establishing the new Lemma 2. To elaborate, in the revision, we adopt the constant batch size $B=1$ and establish the new Lemma 2 based on triangle inequality. This Lemma 2 provides upper and lower bounds of the stochastic gradient norm, based on which we further establish the new Theorem 2 following the same proof logic as the previous version, with slightly different numerical constants.
>
> **Q2**: Change Theorem 1 to only work with the precision $\epsilon$ to stay consistent with Theorem 2.
>
> **A**: Thanks for your suggestion. In the revision, we have rewritten the statements of Theorem 1 in terms of $\epsilon$.
>
> **Q3**: Discuss why you did not consider PL in the stochastic case. Again, you should be consistent throughout the paper.
>
> **A**: Thanks for your suggestion. Yes, similar convergence results can be obtained in the stochastic case. In the revision, we followed your suggestion and established a key descent inequality (see Appendix J) of IAN-SGD under the generalized PL condition. In particular, the obtained descent inequality is highly similar to the Lemma 3 obtained in the deterministic case for AN-GD, and hence the same convergence analysis follows.
>
> **Q4**: Include more prior work using the generalized smoothness assumption Koloskova et al (2023), Zhang et al (2020) and most importantly discuss what do you bring new compared to what is in the previous works (especially the ones I cited concerning the noise assumption and the dependence on ).
>
> **A**: Thanks for pointing out the related references. We have added one paragraph to discuss  **Algorithms for generalized smooth optimization** in the related work section. Moreover, to compare our results with the recent works such as Koloskova et al (2023), Zhang et al (2020) studying gradient clipping algorithms, we create the Table 1 in related work summarizing the assumptions, algorithms, convergence and complexity.
>
> In this work, our aim to advance the theoretical understanding of clipped SGD in generalized-smooth optimization under small batch size and relaxed noise assumptions. By leveraging adaptive gradient normalization and independent sampling, we obtain $\mathcal{O}(\epsilon^{-4})$ complexity with $B=B'=1$ and affine bounded noise. Compared with the previous work zhang et al (2019), our almost sure affine bounded noise is more relaxed than almost sure bounded noise. Notably, when $\tau_1=0$, our results recover the complexity bound obtained in zhang et al (2019) up to difference in constant coefficient.
>
> References:
>
> Koloskova, Anastasia, Hadrien Hendrikx, and Sebastian U. Stich. "Revisiting gradient clipping: Stochastic bias and tight convergence guarantees." International Conference on Machine Learning. PMLR, 2023.
>
> Zhang, Bohang, et al. "Improved analysis of clipping algorithms for non-convex optimization." Advances in Neural Information Processing Systems 33 (2020): 15511-15521.
>
> Zhang, Jingzhao, et al. "Why gradient clipping accelerates training: A
> theoretical justification for adaptivity." arXiv preprint arXiv:1905.11881 (2019)

---

> ### Author Response · Authors · 2025-04-21
> **Response to reviewer 4DJr(continued)**
>
> **Q5**: the dominating term in Thm2 is $\mathcal{O}(\frac{(F(w_0)-F^*+(L_0+L_1)(1+4\tau_2^2)^2)^2}{\epsilon^{4}})$, might (the assumption are slightly different) be worse than what was obtained in previous works such as Zhang et al. (2020) for which this term does not depend on, anyways, this needs to be discussed.
>
> **A**: Thanks for your suggestion. We have added comparisons with Zhang et al (2020) in Table 1 and remark after Theorem 2. To elaborate, they analyze the general clip framework by leveraging momentum acceleration and obtains the complexity $\mathcal{O}(\frac{L_0(F(w_0)-F^*)\tau_2^2}{\epsilon^4})$ if $\frac{\gamma}{c}=\Theta(\tau_2)$, where $\gamma/c$ represents the ratio of learning rate and clipping threshold. We agree their result  has better dependence on the function value gap. We notice that the worse dependence in our result is due to the step $\mathbb{E}\big[\frac{1}{K}\sum_{t\in K}|| \nabla F(w_t)||\big]\leq \sqrt{\frac{1}{K}\sum_{t\in K}\mathbb{E}\big[||\nabla F(w_t) ||^2\big]} $. This step is necessary as our recursion is established over $||\nabla F(w_t) ||^2$ at each time $t\in K$. However, in Zhang et al (2020), by leveraging momentum and mixed clipping update $w_{t+1} = w_t - ( \nu \min(c, \frac{\gamma}{||m_t||} ) m_t + (1 - \nu) \min (c, \frac{\gamma}{||\nabla f_{\xi}(w_t)||} ) \nabla f_{\xi}(w_t) )$, they are able to conduct analysis under a general framework and construct recursion in terms of $\mathbb{E}\big[\frac{1}{T}\sum_{t=0}^{T-1}||\nabla F(w_t) ||]$, which tights the upper bound by eliminating the square root scaling. Despite the worse dependence, our complexity is established under a more relaxed noise assumption than Zhang et al (2020) by allowing $\tau_1>0$.
>
> References
>
> Zhang, Bohang, et al. "Improved analysis of clipping algorithms for
> non-convex optimization." Advances in Neural Information Processing Systems
> 33 (2020): 15511-15521.

---

### Review · Reviewer_ijCU · 2025-04-08

**Summary Of Contributions:**

I thank the authors for the submission. I am pleased to review this interesting paper.

The authors work with Adaptive Normalized Gradient Descent (ANGD) which achieve faster convergence by adapting to problem geometry. Firstly, the authors propose and theoretically justify a new fine-grained approach to adjust deterministic ANGD to generalized Polyak-Łojasiewicz functions (for instance, they extend ANGD theory to strongly convex functions). Secondly, for stochastic optimization, the authors develop a new modification of ANGD (IAN-SGD) to deal with bias of normalized stochastic gradient without large batches. It is based on independent sampling of norm to divide by. Beside optimal theoretical convergence, the authors also demonstrate superior performance of IAN-SGD in relevant real-life setups.  The experiments and ablation study are thorough and representative. They consist of generalized-smooth  Phase Retrieval problem, Distributionally-Robust Optimization and ResNet training on CIFAR10.

**Audience:**

Yes

**Broader Impact Concerns:**

No ethical concerns - theoretical paper

**Claims And Evidence:**

Yes

**Requested Changes:**

__Verdict__

The paper proposes interesting and novel ideas for theoretical analysis and practical use of ANGD. These ideas are also supported by superior performance in relevant experiments. However, there are serious issues that undermine my confidence (see Strengths And Weaknesses).

1) From a practical point of view, the authors do not check how IAN-SGD copes with various scaling errors and small batchsizes (for which it was designed).

2) From a theoretical point of view, the proofs are given under almost surely error assumption 4 and asymmetric generalized smoothness, which are quite restrictive and have been relaxed in other papers.

3) For better writing, I would recommend to fix typos from Minor Issues, mention all related works and add more details in discussions after Theorems 1 and 2.

If the authors address these concerns and polish the writing (fixing typos, expanding discussions, and citing relevant work), I would gladly recommend acceptance.

**Strengths And Weaknesses:**

__Strong points__

0) See Summary Of Contributions

1) The paper is coherent, easy to follow and well-written (aside from some typos noted below).

2) It addresses an important problem of optimizing generalized-smooth functions which has broad practical applications.

3) I found no major mistakes in the proofs.


__Major issues__

1) __Could the authors compare their results from Theorem 1 for $\rho = 2$ with existing results for strongly convex functions, e.g., from Gorbunov et al. (2024), Li et al. (2024)?__ The authors also mention in Related Works results for strongly convex functions from Vankov et al. (2024b), however, I could not locate these.

In addition, how do the rates behave if $L_0 \equiv 0?$

2) Throughout the paper, the authors say that they use weaker  almost surely scaling assumption 4 in comparison with (Zhang et al., 2019; 2020; Liu et al., 2022b). However, they do not mention modern works dedicated to __in expectation__ assumptions. For instance, there are Reisizadeh et al. (2023), Hübler et al. (2024), Jiang et al. (2024) for bounded errors or Chen et al. (2023), Zhang et al. (2024b) for scaling errors.

I think that almost surely assumption 4 is very restrictive, since even additive normal noise does not fit it. Moreover, modern large neural networks tend to have rather heavy-tailed errors [1], making assumption 4 even less applicable.

__Is there any way to prove the theorems under in expectation scaling assumption?__

[1] Zhang, Jingzhao, et al. "Why are adaptive methods good for attention models?." Advances in Neural Information Processing Systems 33 (2020): 15383-15393.


3) In Theorem 2, the authors prove that their IAN-SGD can work with small batchsizes $B,B'$. In ablation study and experiments, they verify it for small $B'.$ However, they never take small $B$ and run all methods in experiments with the same large batchsize.

__Could the authors test and compare IAN-SGD and competitors with small $B$?__

4) In Ablation A.1.2, the authors want to show that IAN-SGD is robust to gradient outliers in assumption 4. However, they do not alter noise parameters $\tau_1, \tau_2$, but vary hyperparameter $\delta$ for the same problems. I do not think that it is effective. Firstly, the outliners remain the same for all $\delta$. Secondly, in sum $2||\nabla f_{\xi_{B'}}(w_t)|| + \delta$, gradient norm may dominate small $\delta,$ resulting in the same IAN-SGD steps and final results.

__I would suggest to conduct experiments with various noise parameters $\tau_1, \tau_2$  and  IAN-SGD with $\delta = \tau_2/\tau_1.$__


__Minor issues__

1) I think it is worth mentioning  the related work dedicated to convergence of SignSGD with momentum under generalized smoothness:

Crawshaw, M., Liu, M., Orabona, F., Zhang, W., \& Zhuang, Z. (2022). Robustness to unbounded smoothness of generalized signsgd.

2) In Related works, the authors say "establishes convergence rate $O(\varepsilon^{-4})$ and corresponding lower bound" and right after "SPIDER algorithm (Fang et al., 2018) can reach the optimal $O(\varepsilon^{-3})$". It is better to mention that SPIDER is designed for sum-type functions.

3) The discussion after Theorem 2 should be written more carefully. For example, RMSProp, NSGD and SignSGD with momentum do not require large batchsizes too, and RMSProp even works under scaling assumption.

4) In \textbf{Algorithms for Generalized-Smooth Optimization} on 3rd page, reference Cutkosky \& Mehta (2020) is misused.

5) After \textbf{Machine Learning Applications} on 1st page, there must be G instead of g.

6) 16th page, "Figure 4: advantage of using" typo.

7) 18th page, "But When" typo.


__More questions__

1) Do results still hold under weaker symmetric generalized smoothness Assumption from Chen et al. (2023)?

2) Could the authors provide examples of Generalized Polyak-Łojasiewicz functions for $\rho < 2?$

---

> ### Author Response · Authors · 2025-04-21
> **Response to reviewer ijCU**
>
> **Q1**: Could the authors compare their results from Theorem 1 for $\rho=2$ with existing results for strongly convex functions, e.g., from Gorbunov et al. (2024), Li et al. (2024) Vankov et al (2024)?
>
> **A**: Thanks for the suggestion. In the revision, we discuss and compare these works after Theorem 1. We summarize the comparison below.
>
> For Gorbunov et al (2024), they introduced strongly convex function, but the main results they obtained for adaptive gradient methods are under convex condition, as opposed to our generalized PL condition. To elaborate, under the choices of learning rate $\gamma = \frac{\eta}{L_0+L_1||\nabla f(w_t) ||}$ and $\gamma = \frac{f(w_t)-f^*}{||\nabla f(w_t) ||}$, they obtained a similar two-phase convergence as our Theorem 1 when $1\geq\beta>2-\rho$, and they showed $\mathcal{O}(\frac{1}{\epsilon})$ local convergence rate. Moreover, by leveraging triangles method and Adaptive Gradient Descent (Malitsky, et al (2020)), they obtained the optimal $\mathcal{O}(\frac{1}{\epsilon})$ complexity under convex and generalized-smooth condition.
>
> For Vankov et al. (2024b), they also studied convergence of normalized and clipped gradient descent methods for non-convex, convex generalized-smooth optimization. Notably, by setting $\gamma=\frac{1}{\sqrt{T}||\nabla f(w_t)||}$, their proposed framework is equivalent to AN-GD with $\beta=1$, and they prove the optimal $\mathcal{O}(\frac{1}{\epsilon})$ complexity under convex and generalized-smooth condition.
>
> For Li et al. (2024), they established the optimal $\mathcal{O}(\log(\frac{1}{\epsilon}))$ complexity under strong convexity and the additional assumption that $||\nabla f(w_t) ||\leq G$. As a comparison, in our work, under PL condition, when $\alpha=\beta=0$, our theorem 1 recovers the linear convergence under $L$-smooth condition; When $\beta>0$, our theorem establishes local linear-convergence in terms of function value gap $F(w_t)-F^*$.
>
> References:
>
> Gorbunov et al (2024) Methods for convex ($L_0$,$L_1$)-smooth optimization: Clipping, acceleration, and adaptivity
>
> Li et al. (2024) Convex and non-convex optimiza- tion under generalized smoothness.
>
>
> Vankov et al. (2024b) Optimizing ($L_0$, $L_1$)-Smooth Functions by Gradient Methods
>
> Malitsky et al (2020). Adaptive gradient descent without descent
>
> **Q2**: How do the rates behave if $L_0=0$?
>
> **A**: In Theorem 1, our learning rate choice is $\gamma = \frac{(2\mu\epsilon)^{\beta/\rho}}{8(L_0+L_1)+1}$. When $L_0=0$, it will not change the order of the learning rate as well as the convergence rates.
>
> **Q3**: The authors also mention in Related Works results for strongly convex functions from Vankov et al. (2024b), however, I could not locate these.
>
> **A**: We apologize for the inconsistency in the related work. Vankov et al (2024) focus on normalized learning rate design of gradient methods for non-convex and convex functions under the generalized-smooth condition. We have corrected the statement accordingly in the related work section.
>
> References: Vankov et al. (2024b) Optimizing ($L_0$, $L_1$)-Smooth Functions by Gradient Methods
>
> **Q4**:  Throughout the paper, the authors say that they use weaker almost surely scaling assumption 4 in comparison with (Zhang et al., 2019; 2020; Liu et al., 2022b). However, they do not mention modern works dedicated to in expectation assumptions. For instance, there are Reisizadeh et al. (2023), Hübler et al. (2024), Jiang et al. (2024) for bounded errors or Chen et al. (2023), Zhang et al. (2024b) for scaling errors.
>
> **A**: We agree that our assumption 2 is not the weakest assumption among all the existing works. In the revision, we summarize and compare the noise assumptions in the related work section, where we mark them with red color. To further clarify the noise assumptions and convergence of gradient clipping algorithms, in the revision, we first established the new Theorem 2 with batch size $B=1$ (we realize that using $B=\Omega(\tau_1^2)$ is redundant). Then we create a table (See Table 1 in related work section) to summarize their smoothness assumptions, noise assumptions, algorithms and complexity result. As a comparison, our results relax the almost-sure bounded noise used in Zhang et al (2019), Zhang et al (2020), and achieves $\mathcal{O}(\epsilon^{-4})$ complexity using small batch size $B=B'=1$ compared with Koloskova et al (2023), Reisizadeh et al (2023).
>
> References:
>
> Reisizadeh et al. (2023) Variance-reduced Clipping for Non-convex Optimization
>
> Hübler et al (2024) Parameter-Agnostic Optimization under Relaxed Smoothness
>
> Jiang et al(2024) Efficient sign-based optimization: Accelerating convergence via variance reduction.
>
> Chen et al(2024). Generalized-smooth nonconvex optimization is as efficient as smooth nonconvex optimization.
>
> Zhang et al (2024b) Convergence guarantees for rmsprop and adam in generalized-smooth non-convex optimization with affine noise variance.

---

> ### Author Response · Authors · 2025-04-21
> **Response to reviewer ijCU (Continued)**
>
> **Q5**: Is there any way to prove the theorems under in expectation scaling assumption like zhang et al (2020)?
>
> **A**: Great question! Yes, we think similar results can be obtained under the in-expectation scaling assumption by using a large batch size. If batch size satisfies $B=\mathcal{O}(\epsilon^{-2})$, the stochastic gradient noise can be controlled as $\mathbb{E}||\nabla f_{B}(w_t)-\nabla F(w_t) ||^2\leq \mathcal{O}(\epsilon^2)$.
> Through this way, by setting $\gamma = \mathcal{O}(\epsilon)$, we can obtain $\frac{1}{T}\sum_{t=0}^{T-1}\mathbb{E}||\nabla f_{B}(w_t) ||\leq \mathcal{O}(\frac{1}{\epsilon T})+\mathcal{O}(\epsilon)$. When $T=\mathcal{O}(\epsilon^{-2})$, the above equation further implies that $\mathbb{E}||\nabla f_{B}(w_{\tilde{t}}) ||\leq \mathcal{O}(\epsilon)$, and moreover $\mathbb{E}||\nabla F(w_{\tilde{t}}) ||\leq \mathcal{O}(\epsilon)$, where $\tilde{t}$ is an index sampled uniformly through all time slots. These arguments lead to $\mathcal{O}(\epsilon^{-4})$ sample complexity. In Reisizadeh et al (2023), they leveraged this strategy to establish the convergence of clipped SGD with batch size $B=\mathcal{O}(\epsilon^{-2})$ under the expected noise assumptions.
>
> We note that this work focuses on the more practical small batch size setting, and hence requires the stronger almost-sure affine-bounded noise assumption. On the other hand, the recent work [2] proves that when assuming $\mathbb{E}[||\nabla f_{\xi}(w_t)-\nabla F(w_t) ||]\leq \sigma^2$, clipped SGD suffers from an unavoidable constant convergence error $\min( \sigma, \frac{\sigma^2}{c})$ under the generalized-smooth condition using small batch size. This reveals the fundamental challenge of proving convergence under expected scaling noise and small batch size in generalized-smooth optimization.
>
> Regarding Zhang et al (2020), which obtains $\mathcal{O}(\epsilon^{-4})$ complexity with $\Omega(1)$ batch size under the **$L$-smooth condition** and expected noise assumption, we found that their analysis cannot be directly followed under the generalized-smooth condition. To elaborate, their result relies on Lemma 11 in their Appendix, which establishes the inequality $\langle \frac{\nabla f_{\xi}(w_t)}{\| \nabla f_{\xi}(w_t)\|},\nabla F(w)\rangle\geq \frac{||\nabla F(w_t) ||}{3}-\frac{||\nabla f_{\xi}(w_t)-\nabla F(w_t) ||}{8}$. This inequality allows them to first decouple $\frac{\nabla f_{\xi}(w_t)}{||\nabla f_{\xi}(w_t) ||}$ and $\nabla F(w_t)$, then take expectation on both sides and further bound $\mathbb{E}||\nabla f_{\xi}(w_t)-\nabla F(w) ||$ using the expected scaling assumption.
>
> Under the generalized-smooth condition, their proof logic cannot be followed due to two main reasons. First, the clipping term $h_t$ includes gradient norm $||\nabla f_{\xi'}(w_t)||$ sampled independently from numerator $\nabla f_{\xi}(w_t)$, which breaks the structure $\langle \frac{\nabla f_{\xi}(w_t)}{||\nabla f_{\xi}(w_t) ||}, \nabla F(w_t)\rangle$. Second, the generalized-smooth condition induces higher order term $\frac{\gamma(L_0+L_1||\nabla F(w_t) ||^{\alpha})||\nabla f_{\xi}(w_t)||^2}{2h_t^{2\beta}}$, which cannot be simply upper bounded by summations of $\mathcal{O}(||\nabla f_{\xi}(w_t)-\nabla F(w_t)||^2)$ and $\mathcal{O}(||\nabla F(w_t)||^2)$ without additional assumptions on $\gamma$. These limitations motivate us to leverage $\frac{||\nabla F(w_t) ||}{h_t}$ using the almost-sure noise assumption, where we can obtain such structure by taking expectation over $\xi$ first, then extracting $||\nabla F(w) ||^2$ from $-\gamma \langle \frac{\mathbb{E}(f_{\xi}(w_t))}{h_t^{\beta}}, \nabla F(w_t)\rangle $ and $\frac{\gamma (4\tau_1^2+2)(L_0+L_1||\nabla F(w_t) ||^{\alpha}) ||\nabla F(w_t)  ||^{2}}{2h_t^{2\beta}}$.
>
> The other listed papers, such as Reisizadeh et al (2023), Huber et al (2023), Jiang et al (2024) and Chen et al(2023), Zhang et al (2024) are dedicated to variance reduction and momentum acceleration in stochastic algorithm design. Such structure is more powerful than gradient clipping in terms of variance control.
>
> References:
>
> [1] Zhang, Jingzhao, et al. "Why are adaptive methods good for attention models?." Advances in Neural Information Processing Systems 33 (2020): 15383-15393.
>
> [2] Koloskova, Anastasia, Hadrien Hendrikx, and Sebastian U. Stich. "Re-
> visiting gradient clipping: Stochastic bias and tight convergence guarantees."
> International Conference on Machine Learning. PMLR, 2023
>
> [3] Reisizadeh et al. (2023) Variance-reduced Clipping for Non-convex Optimization

---

> ### Author Response · Authors · 2025-04-21
> **Response to reviewer ijCU (Continued)**
>
> **Q6**: Could the authors test and compare IAN-SGD and competitors with small $B$?
>
> **A**: Thanks for pointing out these weaknesses. We realized that the requirement $B, B'=\Omega(\tau_1^2)$ of our theorem is unnecessary. In the revision, we have established the new Lemma 2 and Theorem 2 with batch size
> $B,B'= 1$ to remove the dependence of batch size on the stochastic noise assumption.
>
> For the experiments, we have added additional experiments (See Appendix A.4) to show the convergence of IAN-SGD for phase retrieval and DRO under small batch size such as $B=1,2,4$. Our results indicate that IAN-SGD can converge using small batch size. For phase retrieval and DRO, we compare IAN-SGD with other baselines by unifying $B=2$. And for deep learning, we compare IAN-SGD with other baselines by unifying $B=32$. Results indicate that IAN-SGD outer-performs other baseline methods. (See Figure 9, 10 in Appendix A.4)
>
> We also updated our ablation study on $\delta$ by varying $\tau_1,\tau_2$. Details can be found at Appendix A.3.1. For phase retrieval and DRO, we control the data noise by setting batch size $B=16$, we choose various $\tau_1, \tau_2$ leading to different choices of $\delta$. To observe the difference, we unify the learning rate during training. Results indicate that an increasing $\tau_1$ slows down the convergence, whereas the choice of $\tau_2$ is critical to convergence stability but has a marginal effect on convergence speed.
>
> **Minor Issues**
>
> **Q7**: Add related work dedicated to convergence of SignSGD with momentum under generalized smoothness.
>
> **A**: In the revision, we include prior work Crawshaw et al (2022), Sun et al. (2023) which studied variations of sign-SGD under generalized smooth condition and mild noise assumptions. We are glad to include more if you know more works focusing on sign-SGD under generalized smooth condition.
>
> Crawshaw, Michael, et al. "Robustness to unbounded smoothness of generalized signsgd." Advances in neural information processing systems 35 (2022): 9955-9968.
>
> Sun, Tao, et al. "Rethinking sign training: Provable nonconvex acceleration without first-and second-order gradient lipschitz." arXiv preprint arXiv:2310.14616 (2023).
>
> **Q8**: In Related works, the authors say "establishes convergence rate $\mathcal{O}(\epsilon^{-4})$ and corresponding lower bound" and right after "SPIDER algorithm (Fang et al., 2018) can reach $\mathcal{O}(\epsilon^{-3})$ the optimal ". It is better to mention that SPIDER is designed for sum-type functions.
>
> **A**: Thanks for pointing it out. We have updated our related work section to include condition "sum-type functions" when we introduce SPIDER algorithm.
>
> **Q9**: The discussion after Theorem 2 should be written more carefully. For example, RMSProp, NSGD and SignSGD with momentum do not require large batch sizes too, and RMSProp even works under scaling assumption.
>
> **A**: Thank you for pointing it out, we will limit our discussions and comparisons of Theorem 2 to algorithms which have clipping structure. The comparison of recent clipping algorithms and its complexity can also be found at Table 1.
>
> **Q10**: There are several typos in the article.
> 1. In **Algorithms for Generalized-Smooth Optimization** on 3rd page, reference Cutkosky \& Mehta (2020) is misused.
> 2. After **Machine Learning Applications** on 1st page, there must be G instead of g.
> 3. 16th page, "Figure 4: advantage of using" typo.
> 18th page, "But When" typo.
>
> **A**: Thanks for pointing it out. We have fixed these typos.
>
> **Others**
>
> **Q11**: Do results still hold under weaker symmetric generalized smoothness Assumption from Chen et al. (2023)?
>
> **A**: Good question. In the deterministic case, the results of AN-GD can be easily generalized to symmetric generalized smooth condition by setting the step size $\gamma= \frac{(2\mu\epsilon)^{\frac{\beta}{\rho}}}{12(K_0+K_1+K_2)+1}$ without changing the proof logic. For stochastic settings, the inequality $\mathbb{E}|| \nabla f_{\xi}(w') - \nabla f_{\xi}(w) ||^2 \leq || w' - w ||^2 \cdot ( K_0 + K_1 \mathbb{E}|| \nabla f_{\xi}(w) ||^{\alpha} + K_2 || w' - w ||^{\frac{\alpha}{1 - \alpha}} )^2$ induced by symmetric generalized smooth makes components of IAN-SGD, such as gradient clipping and independent sampling hard to analyze. To elaborate, Chen et al (2023) utilizes SPIDER update rule $w_{t+1} = w_t-\gamma \frac{v_t}{||v_t ||}$ to simplify $||w'-w ||$ by $\gamma$, whereas gradient clipping makes $||w'-w ||^{\frac{\alpha}{1-\alpha}}$ difficult to unify with others.
>
> **Q12**: Could the authors provide examples of Generalized Polyak-Lojasiewicz functions for $\rho<2$
>
> **A**: Sure, For function $f(w)=\frac{1}{2}w^2\exp(w^2)$, its gradient is $f'(w)=w\exp(w^2)(1+w^2)$, and $f^*(w)$ is 0 (when $w=0$). Then it satisfies $\frac{\|f'(w)\|}{f(w)}\geq 2$, which corresponds to generalized P{\L} condition with $\rho=1$, $\mu=1$.

---

> > ### Author Response · Authors · 2025-04-21
> > **NA**
> >
> > We thank reviewer ijCU very much for the effort into reviewing our manuscript and providing valuable feedback. Please let us know if further clarifications are needed.

---

> > > ### Comment · Reviewer_ijCU · 2025-05-30
> > >
> > > I thank the authors for their detailed and thoughtful response to my questions, which have been fully addressed. In addition, I would like to highlight two more points for consideration in the revision:
> > >
> > > 1) The remark regarding the convergence of IAN-SGD under the expected noise assumption is particularly insightful. Including this discussion in the manuscript would provide a more complete understanding of the method's behavior and the role of noise in $(L_0, L_1)$-smooth optimization.
> > >
> > > 2) I believe it would be more accurate to state that ClipSGD with small batches (as in [1]) can achieve an $\mathcal{O}(\varepsilon^{-5})$ complexity and $\varepsilon$-stationary convergence by setting $c = \sigma/\varepsilon$ and $\eta = \varepsilon^2/(\sigma^2(L_0 + cL_1))$. This adjustment ensures a fair and precise comparison between the methods.
> > >
> > > [1] Anastasia Koloskova, Hadrien Hendrikx, and Sebastian U Stich. Revisiting gradient clipping:
> > > Stochastic bias and tight convergence guarantees. In International Conference on Machine
> > > Learning, pages 17343–17363. PMLR, 2023.

---

### Author Response · Authors · 2025-07-09
**Response to required minor changes**

Dear Action Editor and Reviewers,
We sincerely thank you for your constructive feedback and valuable suggestions, which have significantly improved the quality of our manuscript.

In response to the revision request from reviewer ijCU, we have made the following updates:

Regarding the **first point**, we have expanded our discussion comparing the noise assumption adopted in Assumption 4 with those used in prior works, including the almost sure bounded noise assumption in [1,2,3] and the expected noise assumption in [4,5]. We clarified our motivation for using a relaxed noise condition, which aims to strike a balance between achieving the standard $\mathcal{O}(\epsilon^{-4})$ convergence rate and maintaining an $\Omega(1)$-level batch size. Additionally, we added Remark 3, briefly discuss the insight behind how a large batch size of $\Omega(\epsilon^{-2})$ can help improve the convergence rate from $\mathcal{O}(\epsilon^{-5})$ to $\mathcal{O}(\epsilon^{-4})$ under expected bounded noise assumption, following the formal proof logic of [5].


Regarding the **second point**, we have updated our discussion of [4] accordingly. We revised the description of [4] in Table 1, related work and main section to ensure a consistent and fair comparison across methods.

Finally, we have further refined the manuscript by correcting several misleading mathematical notations and grammatical errors throughout the text.

best regards

TMLR 4210 Authors

References:

[1] Jingzhao Zhang, Tianxing He, Suvrit Sra, and Ali Jadbabaie. Why gradient clipping accelerates training: A
theoretical justification for adaptivity. In International Conference on Learning Representations, 2019.

[2] Bohang Zhang, Jikai Jin, Cong Fang, and Liwei Wang. Improved analysis of clipping algorithms for nonconvex
optimization. In Advances in Neural Information Processing Systems, 2020.

[3] Mingrui Liu, Zhenxun Zhuang, Yunwen Lei, and Chunyang Liao. A communication-efficient distributed gradient clipping algorithm for training deep neural networks. In Advances in Neural Information Processing
Systems, 2022.

[4] Anastasia Koloskova, Hadrien Hendrikx, and Sebastian U Stich. Revisiting gradient clipping: Stochastic bias
and tight convergence guarantees. In International Conference on Machine Learning, pp. 17343–17363.
PMLR, 2023.

[5] Amirhossein Reisizadeh, Haochuan Li, Subhro Das, and Ali Jadbabaie. Variance-reduced clipping for nonconvex
optimization. arXiv preprint arXiv:2303.00883, 2023.

---

### Decision · Action_Editor_tq57 · 2025-06-11

**Recommendation:** Accept with minor revision

**Additional Comments:**

This paper studies the convergence of normalized (clipped) gradient descent under a generalized smoothness condition.
For the deterministic setting, the algorithm's convergence is established under the additional assumption of the generalized PŁ condition.
Under stochastic gradient oracles, a new algorithm is proposed, which uses two samples to estimate direction and normalization independently, and its convergence rate is analyzed under additional gradient noise assumptions.

The reviews of this submission were particularly detailed. The authors fixed many issues in the original submission in their revision and clarified the reviewers' concerns. Reviewer ijCU recommends including some of the insightful discussion from the rebuttal to the manuscript (see the minor revision request by reviewer ijCU: https://openreview.net/forum?id=KKSQQMlEfw&noteId=NWS6YLDH94). We kindly ask the authors to consider this comment when preparing the final version.

**Audience:**

Yes

**Audience Explanation:**

Normalized and clipped gradient descent methods are used for neural network training, particularly in combination with privacy-preserving mechanisms or to enhance robustness. Therefore, studying and analyzing the convergence of such methods is of interest to the TMLR audience.

**Claims And Evidence:**

Yes

**Claims Explanation:**

The assumptions are clearly stated, and the proofs for all the theorems are included in the submission. The revision addressed some major concerns, and the reviewers now find the evidence sufficiently accurate and clear.